# Multiple sclerosis: Exploring the limits and implications of genetic and environmental susceptibility

**Douglas S. Goodin** [1]*, **Pouya Khankhanian**[2], **Pierre-Antoine Gourraud**[3], **Nicolas Vince**[4]

**1** Department of Neurology, San Francisco & the San Francisco VA Medical Center, University of California, San Francisco, San Francisco, California, United States of Ameirca, **2** Kaiser Permanente, Walnut Creek Medical Center, Dublin, California, United States of Ameirca, **3** Center for Neuro-Engineering and Therapeutics, University of Pennsylvania, Philadelphia, Pennsylvania, United States of Ameirca, **4** INSERM, Center for Research in Transplantation and Translational Immunology, UMR 1064, Nantes Université, Nantes, France

\* douglas.goodin@ucsf.edu

## Abstract

### Objective

To explore and describe the basis and implications of genetic and environmental susceptibility to multiple sclerosis (MS) using the Canadian population-based data.

### Background

Certain parameters of MS-epidemiology are directly observable (e.g., the recurrence-risk of MS in siblings and twins, the proportion of *women* among MS patients, the population-prevalence of MS, and the time-dependent changes in the *sex-ratio*). By contrast, other parameters can only be inferred from the observed parameters (e.g., the proportion of the population that is "genetically susceptible", the proportion of *women* among susceptible individuals, the probability that a susceptible individual will experience an environment "*sufficient*" to cause MS, and if they do, the probability that they will develop the disease).

### Design/methods

The "genetically susceptible" subset ($G$) of the population ($Z$) is defined to include everyone with <u>any</u> non-zero life-time chance of developing MS under <u>some</u> environmental conditions. The value for each observed and non-observed epidemiological parameter is assigned a "plausible" range. Using both a *Cross-sectional Model* and a *Longitudinal Model*, together with established parameter relationships, we explore, iteratively, trillions of potential parameter combinations and determine those combinations (i.e., solutions) that fall within the acceptable range for both the observed and non-observed parameters.

### Results

Both *Models* and all analyses intersect and converge to demonstrate that probability of genetic-susceptibilty, $P(G)$, is limited to only a fraction of the population {i.e., $P(G) \leq 0.52$}

**Data Availability Statement:** All of the data underlying the reported results of the present manuscript are from papers in the published litterature. These references are publicly available in

the published papers and are referenced in the manuscript. I do not (and did not) have access to any of the original data underlying these publications.

**Funding:** The authors received no specific funding for this work.

**Competing interests:** The authors have declared that no competing interests exist.

and an even smaller fraction of *women* {i.e., $P(G|F) < 0.32$)}. Consequently, most individuals (particularly *women*) have no chance whatsoever of developing MS, regardless of their environmental exposure. However, for any susceptible individual to develop MS, <u>requires</u> that they also experience a "*sufficient*" environment. We use the Canadian data to derive, separately, the exponential response-curves for *men* and *women* that relate the increasing likelihood of developing MS to an increasing probability that a susceptible individual experiences an environment "*sufficient*" to cause MS. As the probability of a "*sufficient*" exposure increases, we define, separately, the limiting probability of developing MS in *men* (**c**) and *women* (**d**). These Canadian data strongly suggest that: ($c < d \leq 1$). If so, this observation establishes both that there must be a "truly" random factor involved in MS pathogenesis and that it is this difference, rather than any difference in genetic or environmental factors, which primarily accounts for the *penetrance* difference between *women* and *men*.

## Conclusions

The development of MS (in an individual) requires both that they have an appropriate genotype (which is uncommon in the population) and that they have an environmental exposure "*sufficient*" to cause MS given their genotype. Nevertheless, the two principal findings of this study are that: $P(G) \leq 0.52$)} and: ($c < d \leq 1$). Threfore, even when the necessary genetic and environmental factors, "*sufficient*" for MS pathogenesis, co-occur for an individual, they still may or may not develop MS. Consequently, disease pathogenesis, even in this circumstance, seems to involve an important element of chance. Moreover, the conclusion that the macroscopic process of disease development for MS includes a "truly" random element, if replicated (either for MS or for other complex diseases), provides empiric evidence that our universe is non-deterministic.

## Introduction

Susceptibility to multiple sclerosis (MS) is known to be complex, involving the critical interplay between both environmental events and genetic factors [1–3]. Our previously published analysis regarding the nature of this susceptibility [3] was based on a few basic, well-established, epidemiological parameters of MS, which have been repeatedly observed in populations across Europe and North America. These parameters include the prevalence of MS in a population, the recurrence-risk for MS in siblings and twins of individuals with MS, the proportion of *women* among MS patients, and the time-dependent changes in both the female-to-male (*F: M*) *sex-ratio* and the disease prevalence, which have taken place over the last several decades [3]. For this analysis, we defined a "genetically susceptible" subset (*G*) of the general population (*Z*) to include everyone who has any non-zero chance of developing MS over the course their lifetime. We concluded that genetic susceptibility, so defined, is limited to only a small proportion of these northern populations (<7.3%) and, thus, that most individuals in these populations have no chance whatsoever of developing MS, regardless of any environmental conditions that they may experience during their lifetimes [3]. Nevertheless, despite this critical dependence of susceptibility to MS upon the genotype of an individual, we also concluded that certain environmental events were also necessary for MS to develop and that, consequently, both essential genetic factors and essential environmental events are in the causal

pathway leading to MS [3]. If either of these are missing, MS cannot develop. Finally, we concluded that, seemingly, even when the "*sufficient*" genetic and environmental determinants were present, the actual development of MS depended, in part, upon an element of chance [3].

What this analysis did not undertake, however, was to explicitly explore the limits of these conclusions based upon the statistical uncertainties, which surround each of the various epidemiological observations that have been made. It also did not explore the potential limitations of, and the implications for, our conclusion that chance plays an important role in disease pathogenesis. It is the purpose of this study, therefore, to undertake these explorations using both the confidence intervals (*CIs*) and "plausible" ranges for the different basic epidemiological parameters and by incorporating these uncertainties into the governing equations relating these parameters both to each other and to the underlying susceptibility to MS that exists within in a population.

For this analysis, we have used, primarily, the data reported from the Canadian Collaborative Project on Genetic Susceptibility to Multiple Sclerosis [4, 5]. The reason for this choice is three-fold. First, this Canadian dataset is a *population-based* sample with an initial cohort of 29, 478 MS patients who were born between the years 1891 and 1993 [4–7]. This cohort consists of all MS patients seen in 15 MS Centers scattered throughout the Canadian Provinces [5]. The cohort did not specifically include patients from the Northern Territories [5] although, likely, many of these patients were referred for 2$^{nd}$ opinions to the provincial centers. This study endeavored to include most (or all) of the MS patients in Canada at the time and, indeed, the authors estimate (from their twin studies) that their ascertainment scheme captured 65–83% of all Canadian MS patients [7]. The total population of Canada in 2010 was 34 million people [8]. Therefore, depending upon the number of patients in the cohort who were still alive at the time of ascertainment, this translates to a prevalence of MS in this region of approximately 105–134 persons per 100, 000 population. For the purposes of our analysis, this cohort is assumed to represent a large random sample of the symptomatic Canadian MS population at the time. Second, this dataset provides, from the same population, estimates for the recurrence-risk in monozygotic (*MZ*) twins, in dizygotic (*DZ*) twins, in non-twin siblings (*S*), and for changes in the (*F:M*) *sex-ratio* over time [4–7]. Consequently, this Canadian dataset is likely among the most complete and the most reliable in the world. And third, these data come from a single geographic region of similar latitude, which is critical when considering a disease, for which disease prevalence has a marked latitudinal gradient in different parts of the world [9].

## Methods

### 1. General methods

**A. General model specifications and definitions for genetic susceptibility to MS.** We consider a general population (*Z*), which is composed of (*N*) individuals (*k* = 1, 2, . . ., *N*) who are living under the prevailing environmental conditions during some specific *Time Period* (*T*)–conditions that are designated, generically, as (*E_T*). In Table 1, we define the different parameters used in our analysis and, in Table 2, we provide a set of parameter abbreviations, which are used for the purposes of notational simplicity. The subset (*MS*) is defined to include <u>all</u> individuals within (*Z*) who either have or will develop MS over the course of their lifetime. The occurrence of (*MS*) represents the event that an individual, randomly selected from (*Z*), belongs to this (*MS*) subset and the term $P(MS|E_T)$ represents the probability of this event, given the prevailing environmental conditions of (*E_T*)—i.e., $P(MS|E_T) = P(MS|Z, E_T)$. This probability is referred to as the "*penetrance*" of MS for the population (*Z*) during the *Time Period* (*E_T*). The occurrence of (*G*) is defined as the event that an individual, randomly selected

**Table 1. Definitions for the groups and epidemiological parameters used in the analysis.**

| Parameter | Definition |
|---|---|
| $(Z)$ | Set of all $(N)$ individuals (i.e., unique genotypes) in the population |
| $(G)$, $(G^c)$ | Subsets of individuals (genotypes) in $(Z)$ who have _any_ non-zero chance $(G)$ or no chance $(G^c)$ of developing MS |
| $(G_{w^s})$, $(G_{dw^s})$ | Subset of susceptible _women_: $(G_{w^s}) = (F, G)$; $(G_{dw^s}) =$ the $d^{th}$ susceptible _woman_ in $(G)$ |
| $(G_i)$, $(G_{ia})$, $(G_{is})$, $(G_{it})$ | Genotypes or subsets either of the $i^{th}$, or an _i-type_, individual in $(G)$: $(G_i) =$ full unique genotype; $(G_{ia}) =$ genotype of only autosomal _MS_-related factors; $(G_{is}) =$ genotype of all _MS_-related factors; $(G_{it}) =$ all genotypes who share the same $\{E_i\}$ family of _sufficient_ exposures |
| $\{G_s\}$, $\{G_a\}$ | Families of sets: – $\{G_s\}$ includes all subsets $(G_{it})$ within $(G)$; and $\{G_a\}$ includes all subsets $(G_{ia})$ within $(G)$ |
| $(M)$, $(F)$ | Subsets of _men_ $(M)$ and _women_ $(F)$ in $(Z)$—$P(M) + P(F) = 1$ |
| $(G1)$, $(G2)$ † | "_High_" $(G1)$ and "_low_" $(G2)$ _penetrance_" subsets in $(G)$–_see Text_ |
| $(G1')$, $(G2')$ † | "_High_" $(G1')$ and "_low_" $(G2')$ _penetrance_" subsets in $(F, G)$ or $(M, G)$ |
| $(MZ)$, $(DZ)$, $(S)$ | Subsets _MZ_-twins, _DZ_-twins, or non-twin siblings $(S)$ in $(Z)$ |
| $(E_T)$ † | The prevailing environmental conditions during the _Time Period_ $(T)$ |
| $\{E_i\}$ † | Family of sets of environmental exposures, each of which is _sufficient_, by itself, to cause MS in the $i^{th}$ individual in $(G)$–_see Text_ |
| $\{E_{iw}\}$ † | Family of sets of more "_intense_" environmental exposures, within $\{E_i\}$, required by an _i-type_ _woman_–$(E_{iw}) \subset (E_i)$–_see Text_ |
| $(E)$ † | Union of all disjoint events $(\{E_i\}, G_i)$–_see Text_ |
| $(E_w)$ † | Union of all disjoint events $(\{E_{iw}\}, G_i)$–$(E_w) \subset (E)$–_see Text_ |
| $(E_{pop})$, $(E_{sib})$, $(E_{twn})$ | Distinct parts of a _sufficient_ environmental exposure, equally likely to be shared by anyone $(E_{pop})$, more or less likely to be shared by siblings $(E_{sib})$ and more or less likely to be shared by twins $(E_{twn})$–_see Text_ |
| $(MS)$ † | Subset of individuals in $(Z)$ who either have, or will develop, _MS_ |
| $(MZ_{MS})$, $(DZ_{MS})$, $(S_{MS})$ | Subsets of _MZ_ co-twins $(MZ_{MS})$, _DZ_ co-twins $(DZ_{MS})$, or non-twin co-siblings $(S_{MS})$ who either have, or will develop, _MS_–_see Text_ |
| $P(MS)$ † | Life-time probability of developing _MS_ for a member of $(Z)$: |
| $P(MS \mid MZ_{MS})$ † $P(MS \mid DZ_{MS})$ † $P(MS \mid S_{MS})$ † | $P(MS)$ for a proband, randomly selected from $(Z)$, who has a co-twin or co-sibling in the $(MZ_{MS})$, $(DZ_{MS})$, or $(S_{MS})$ subsets–_see Text_ |
| $P(MS \mid IG_{MS})$ † | $P(MS \mid MZ_{MS})$ adjusted for the shared environment of _MZ_-twins |
| $P(MZ_{MS})$ † $P(IG_{MS})$ † | $P(MS)$ for the co-twin from an _MZ_ twin-pair, without considering the proband's circumstances. $P(MZ_{MS}) = P(IG_{MS}) = P(MS)$–_See Text_ |

† Parameters that vary with environmental conditions $(E_T)$, which is indicated in different manners in the _Text_, depending upon context. For example, during: $(E_T) =$ _Time Period #2_: $P(MS) = P(MS \mid Z, E_T) = P(MS \mid E_T) = P(MS \mid Z)_2 = P(MS)_2$

The subscript "MS" indicates that the non-proband relative preceding the subscript has or will develop MS

from $(Z)$, is a member of the $(G)$ subset. The term $P(G \mid E_T)$ represents the probability of this event given the prevailing environmental conditions of $(E_T)$. In turn, the $(G)$ subset is defined to include <u>all</u> individuals (genotypes) within $(Z)$ who have <u>any</u> non-zero chance of developing MS under <u>some</u> (unspecified and not necessarily realized) environmental conditions, regardless of how small that chance might be or how rarely the appropriate environmental conditions might occur. Consequently, <u>everyone</u> who actually develops MS <u>must</u> belong to the $(G)$ subset. Moreover, we assume that a person's genotype is independent of the environmental conditions that prevail during $(E_T)$. Therefore:

$$P(G \mid Z, E_T) = P(G \mid E_T) = P(G \mid Z) = P(G)$$

**Table 2. Principal parameter abbreviations[†].**

| Parameter | Definition |
|---|---|
| $(F, w)$ & $(M, m)$ | Alternate designations for: female/*women* $(F, w)$ and: male/*men* $(M, m)$ |
| *Subscripts* (1, 2) | Indicators for either *Time Period #1* or the "*current*" *Time Period #2*—except for the parameters: $x_1, x_2, x_1', x_2', y_1',$ *and* $y_2'$—see below |
| $x$ | *Penetrance* of MS for the $(G)$ subset—$x = P(MS \mid G)$ |
| $x_i$ | *Penetrance* of the $(i^{th})$ genotype in $(G)$ — $\forall G_i \in G$: $P(MS \mid G_i) = x_i$ |
| $X$ | Set of all individual *penetrance* values in $(G)$—$X = \{x_i\}$; $i = 1, 2, \ldots, m$ |
| $x'$ | Adjusted *MZ*-twin concordance for MS—$x' = P(MS \mid IG_{MS})$ |
| $\sigma_X^2$ | Variance of the set $(X)$—$Var(X)$ |
| $x_1$ | MS-*penetrance* of $(G1)$ subset (or *women*)—$x_1 = P(MS \mid G1)$; *see Text* |
| $x_2$ | MS-*penetrance* of $(G2)$ subset (or *men*)—$x_2 = P(MS \mid G2)$; *see Text* |
| $Zp$ | Failure probability in *all*: $Zp = P(MS, E \mid G)$; *see Text* |
| $Zw$ | Failure probability in *women*: $Zw = P(MS, E_w \mid F, G) = x_1$; *see Text* |
| $Zm$ | Failure probability in *men*: $Zm = P(MS, E \mid M, G) = x_2$; *see Text* |
| $x_1'$ | Adjusted *MZ*-twin concordance for $(G1)$ subset—$x_1' = P(MS \mid G1, IG_{MS})$ |
| $x_2'$ | Adjusted *MZ*-twin concordance for $(G2)$ subset—$x_2' = P(MS \mid G2, IG_{MS})$ |
| $y_1'$ | *MZ*-twin concordance for the $(G1)$ subset—$y_1' = P(MS \mid G1, MZ_{MS})$ |
| $y_2'$ | *MZ*-twin concordance for the $(G2)$ subset—$y_2' = P(MS \mid G2, MZ_{MS})$ |
| $u$ | Odds of a *sufficient* environmental exposure—$u = P(E) / [1 - P(E)]$ |
| $q_m^{min}, q_w^{min}$ | "*Minimum*" exposure level change in *men* $(q_m^{min})$ and *women* $(q_w^{min})$ |
| $q_p, q_m, q_w$ | "*Actual*" exposure level change in *all* $(q_p)$, *men* $(q_m)$ and *women* $(q_w)$ |
| $b(u), h(u), k(u)$ | Hazard functions for *all*–$b(u)$; for *men*–$h(u)$; and for *women*–$k(u)$ |
| $R$ | "*Actual*" proportionality factor (*if proportional*)—$k(u) = R * h(u)$ |
| $R_i, R_{dw^s}$ | "*Actual*" *value of R* in *i*-type $(R_i)$ and individual $(R_{dw^s})$ women |
| $R^{app}$ | "*Apparent*" value of $R$—$R^{app} = q_w^{min}/q_m^{min}$ |
| $\lambda_w, \lambda_m$ | Environmental threshold to develop MS in *women* $(\lambda_w)$ and *men* $(\lambda_m)$ |
| $\lambda, \lambda_i$ | Threshold differences, generally: $\lambda = (\lambda_w - \lambda_m)$; and for "*i-types*" $(\lambda_i)$ |
| $C$ | Ratio of $P(MS)$ at *Time #1* to that at *Time #2*—$C = P(MS)_1/P(MS)_2$ |
| $C_F, C_M$ | The ratio $(C)$ for *women* $(C_F)$; and for *men* $(C_M)$ |
| $p, p'$ | Proportion of *women* in subset $(G) - p$; and in subset $(MS, G) - p'$ |
| $r, s$ | Enrichment of genotypes – *women*: $r = (x_1'/x_1)$; men: $s = (x_2'/x_2)$ |
| **$b, c, d$** | Exponential response curve limits in *all* (**$b$**); *men* (**$c$**); and *women* (**$d$**) |
| $s_a$ | Factor to adjust $P(MS \mid MZ_{MS})$ because of the shared twin environment $P(MS \mid IG_{MS}) = P(MS \mid MZ_{MS})/s_a$; $s_a \geq 1$—see Text |

[†] Each of these parameters (except $h(u), k(u), R, R_i, R_{w_j}, \lambda, \lambda_w, \lambda_m, p,$ **$b, c,$** & **$d$**) vary with different environmental conditions $(E_T)$, which is indicated in various ways in the *Text*–see Table 1

In this circumstance, each of the $(m \leq N)$ individuals in the $(G)$ subset $(i = 1, 2, \ldots, m)$ has a unique genotype $(G_i)$. The occurrence of $(G_i)$ represents the event that an individual, randomly selected from $(Z)$, belongs to the $(G_i)$ subset–a subset consisting of only a single individual (i.e., the so-called "$i^{th}$ susceptible individual" or "$i^{th}$ individual")–and the term $\{P(G_i) = 1/N\}$ represents the probability of this event. Therefore, it follows from the definition of the $(G)$ subset that, if *every* relevant environmental condition–*see below*–is possible during

some *Time Period* ($E_T$), then, during this *Time Period*:

$$\forall\, G_i \in G : P(MS\,|\,G_i, E_T) = x_i > 0$$

The conditional probabilities: $\{x_i = P(MS\,|\,G_i, E_T)\}$ and: $\{x = P(MS\,|\,G, E_T)\}$, are referred to as the "*penetrance*" of MS, during ($E_T$), for $i^{th}$ susceptible individual and for the ($G$) subset, respectively. Clearly, the *penetrance* of MS, both for the individual and for the group, will vary depending upon the likelihood of different environmental conditions during different *Time Periods*. If the environmental conditions during some *Time Periods* were such that certain members of the ($G$) subset have no possibility of ever developing MS, then, for these individuals, during these *Time Periods*:

$$P(MS\,|\,G_i, E_T) = 0$$

Because, *currently*, some individuals can (and do) develop MS, it must be that, during our "*current*" *Time Period*:

$$P(MS\,|\,G, E_T) > 0$$

However, if, at some other time, the environmental conditions were such that no member of ($G$) could ever develop MS then, during these *Time Periods*:

$$P\big(MS\,|\,G, E_T\big) = P\big(MS\,|\,E_T\big) = 0$$

We also define a subset ($G_{w^s}$), which includes of <u>all female</u> members of the ($G$) subset {i.e., ($G_{w^s}$) = ($F, G$)}, and we define the proportion of *women* in ($G$) as: $p = P(F\,|\,G)$. In this circumstance, each of the ($m * p$) *women* in the ($G_{w^s}$) subset ($d = 1, 2, \ldots, mp$) has a unique genotype ($G_{dw^s}$). The occurrence of ($G_{dw^s}$) represents the event that an individual, randomly selected from ($Z$), belongs to the ($G_{dw^s}$) subset–a subset consisting of only the $d^{th}$ susceptible *woman*– and the term $\{P(G_{dw^s}) = 1/N\}$ represents the probability of this event. Also, $\{P(G_{w^s}) = mp/N\}$ represents the probability of the event that an individual, randomly selected from ($Z$), belongs to the ($G_{w^s}$) subset.

Individuals, who do not belong to the ($G$) subset, belong to the mutually exclusive (complimentary) subset ($G^c$), which consists of <u>all</u> individuals who have <u>no</u> chance, whatsoever, of developing MS, regardless of any environmental experiences that they either have had or could have had. The occurrence of ($G^c$) is defined as the event that an individual, randomly selected from the population ($Z$), is a member of the ($G^c$) subset. The term $P(G^c\,|\,E_T)$ represents the probability of this event, given the environmental conditions of ($E_T$). Consequently, each of the ($m^c = N - m$) "non-susceptible" individuals in the ($G^c$) subset ($j = 1, 2, \ldots, m^c$) has a unique genotype ($G_j$). As *above*, the occurrence of ($G_j$) represents the event that an individual, randomly selected from ($Z$), belongs to the ($G_j$) subset–a subset also consisting of a single individual. The term $\{P(G_j) = 1/N\}$ represents the probability of this event. Thus, under <u>any</u> environmental conditions, during <u>any</u> *Time Period*:

$$\forall G_j \in G^c \,\&\, \forall(E_T) : P(MS\,|\,G_j, E_T) = P(MS\,|\,G_j) = 0$$

and thus,

$$\forall(E_T) : P(MS\,|\,G^c, E_T) = P(MS\,|\,G^c) = 0$$

Notably, *MZ*-twins, despite having nearly "identical" genotypes (*IG*), nevertheless, still have subtle genetic differences from each other. Thus, even if these subtle differences are irrelevant

to MS susceptibility (as seems likely, and which we assume to be true), these differences still exist. Consequently, every individual–i.e., each complete genotype $(G_i)$ and $(G_j)$–in the population is unique. Despite this uniqueness, however, we can also define a so-called "susceptibility-genotype" for the $i^{th}$ susceptible individual such that this genotype consists of <u>all</u> (and only) those genetic factors, which are related to MS susceptibility. Because the specification of such a susceptibility-genotype necessarily includes many fewer genetic factors than the $i^{th}$ individual's complete genotype, it is possible that one or more other individuals in the population share the same susceptibility-genotype with the $i^{th}$ individual. For example, in this conceptualization, *MZ*-twins would necessarily belong to the same susceptibility-genotype. We refer to the group of individuals, who belong to the $i^{th}$ susceptibility-genotype, as the $(G_{is})$ subset within $(Z)$. The occurrence of $(G_{is})$ represents the event that a person, randomly selected from $(Z)$, belongs to the $(G_{is})$ subset, which consists of a single susceptibility genotype. The term $P(G_{is})$ represents probability of this event. Some members of $(G)$ are *MZ*-twins and, thus, both twins are members of the same $(G_{is})$ subset. Therefore, the total number of these susceptibility-genotypes in the population $(m_{is})$ must be less than $(m)$–i.e., $(m_{is} < m)$.

Also, it is possible that two or more "susceptibility genotypes" may share an identical family of "*sufficient*" environmental exposures $\{E_i\}$ with the $i^{th}$ individual (*see* Methods *#2B; below*). Therefore, we define the "*i-type*" group $(G_{it})$ to include <u>all</u> "susceptibility genotypes" who share the same $\{E_i\}$ family. The probability $\{P(G_{it})\}$ represents the probability of the event that and individual, randomly selected from $(Z)$, belongs to the $(G_{it})$ group. Also, from *above*, the total number of "*i-type*" groups in the population $(m_{it})$ must be less than $(m)$–i.e., $(m_{it} \le m_{is} < m)$. In addition, we define the family $\{G_s\}$ to include <u>all</u> of the "*i-type*" groups $(G_{it})$ within $(Z)$ and define the event $\{G_s\}$ as representing the union of the disjoint $(G_{it})$ events such that:

$$\{G_s\} = (G_{1t}) \cup (G_{2t}) \cup \ldots \cup (G_{m_{it}t})$$

Because every susceptible person belongs to one, and only one, of these "*i-type*" groups, the probability of this event is expressed as:

$$P(\{G_s\}) = P(G) = m/N$$

We also define the set $(X)$ to be the set of *penetrance* values for members of the $(G)$ subset during some *Time Period*. Provided that the variance of $(X)$ is not equal to zero {i.e., $Var(X) = \sigma_X^2 \ne 0$}, the subset $(G)$ can be further partitioned into two mutually-exclusive subsets, $(G1)$ and $(G2)$, suitably defined, such that the *penetrance* of MS for the subset $(G1)$ during a certain *Time Period* is greater than that for $(G2)$. The terms $P(G1)$ and $P(G2)$, represent the probabilities of the events that an individual, randomly selected from $(Z)$, is a member of the subsets $(G1)$ and $(G2)$, respectively. Although many such partitions are possible, for the purposes of the present manuscript, $(G1)$ is generally considered interchangeable with the subset of susceptible *women*–i.e., $(G1) = (F \cap G) = (F, G) = (G_{w^s})$–and $(G2)$ is generally considered interchangeable with the subset of susceptible *men*–i.e., $(G2) = (M \cap G) = (M, G)$.

When considering the enrichment of more penetrant genotypes (*see Section 2a, 2b in* S1 File), the subsets $(F, G)$ and $(M, G)$ will each be further partitioned into high- and low-*penetrance* sub-subsets–i.e., $(G1')$ and $(G2')$, respectively–where the basis for this further partition into $(G1')$ and $(G2')$ sub-subsets is unspecified. The definitions of, and the probabilities for, these events mirrors that *above* for $(G1)$ and $(G2)$. Moreover, although the basis for this further partition must be something other than gender, it can be anything else that creates a partition, and it doesn't need to be the same basis for both genders.

*{NB*: *A note on terminology. When a claim refers to any partition of the* $(G)$ *subset, the probabilities of developing MS (i.e., the penetrance of MS) for members of the* $(G1)$ *and* $(G2)$ *subsets)*

*are designated, respectively, such that: $x_1 = P(MS \mid G1)$ and: $x_2 = P(MS \mid G2)$. When the partition is based specifically on gender, to provide clarity, and to avoid any confusion with our Time Period designations, the group of females/women are indicated, alternatively, either by an upper-case (F) or by a lower-case (w). Similarly, in these circumstances, males/men are indicated, alternatively, either by an upper-case (M) or by a lower-case (m)–see* Table 2. *In some circumstances, however, (when the meaning is clear), for purposes of notational simplicity, the designations of $(x_1)$ and $(x_2)$ continue to be used to designate the penetrance of MS for the subsets of susceptible women $(x_1)$ and men $(x_2)$. In other circumstances, however, greater clarity is provided by using the letter designations. For example, considering the partition of (G) into the subsets (F, G) and (M, G), the penetrance of MS for susceptible women and men are designated, respectively, as (Zw) and (Zm) such that: $x_1 = Zw = P(MS \mid F, G)$ and: $x_2 = Zm = P(MS \mid M, G)$–see* Table 2. *When the listing of individual women within the (F, G) subset is important to an argument, the designations $(G_{w^s})$ and $(G_{dw^s})$ are used.*

*Moreover, although this manuscript focuses on the gender partition for the disease MS, the Models developed pertain to any partition for any disease, which has data analogous to that found in Canada for the gender partition of MS* [6, 7].*}*

**B. General model specifications and definitions for environmental susceptibility to MS.** The term $\{E_i\}$ represents the family of specific sets of environmental exposures, each of which, by itself, is "*sufficient*" to cause MS to develop in the $i^{th}$ susceptible individual. Each set within the $\{E_i\}$ family must be distinct (in some respect) from every other set within this family but, otherwise, there can be any degree of overlap between the factors or events that comprise these sets. Also, there can be any number of sets within the $\{E_i\}$ family although, because ($\forall\ G_i \in G$: $x_i > 0$) under <u>some</u> environmental conditions, the family cannot be empty. Thus, at least one "*sufficient*" set of exposures <u>must</u> exist for <u>every</u> susceptible individual. If we assign $(v_i)$ to the number of sets of sufficient exposures for the $i^{th}$ (or an "*i-type*") susceptible individual, then $\{E_i\}$ represents the family of sets: $\{E_{i1}, E_{i2}, \ldots, E_{iv_i}\}$; and $P(\{E_i\} \mid E_T)$ represents the probability of the event that, at least, one of these sets of "*sufficient*" exposure occurs, given the prevailing environmental conditions of the time $(E_T)$. Moreover, if more than one individual belongs to a particular "*i-type*" group, each group-member, by definition, will have the same $\{E_i\}$ family of "*sufficient*" exposures as the $i^{th}$ individual.

Notably, also, the probability, $P(\{E_i\})$, depends entirely upon the actual environmental conditions that prevail during any *Time Period*–i.e., conditions that are fixed for any specific $(E_T)$. Thus:

$$\forall(i = 1, = 2, \ldots, m) : P(\{E_i\} \mid E_T) = A_i$$

where $(A_i)$ represents an unknown constant. This constant $(A_i)$ may be different for each $\{E_i\}$ and, also, it may be different during different *Time Periods*. Consequently, during any $(E_T)$, each $\{E_i\}$ represents a *population-wide* exposure–i.e., an exposure that is "*available*" to everyone. However, whether anyone, in particular the $i^{th}$ susceptible individual, experiences that exposure, is a different matter (*see below*).

Also, for MS to develop in the $i^{th}$ susceptible individual, the events $\{E_i\}$ and $(G_i)$ must occur jointly–i.e., the individual $(G_i)$ must experience at least one of the $\{E_i\}$ environments. This joint occurrence is represented by the subset $(\{E_i\}, G_i)$. The occurrence of $(\{E_i\}, G_i)$ represents the event that an individual, randomly selected from $(Z)$, is both in the $(G_i)$ subset (*described above*) and that they experience an environment "*sufficient*" to cause MS in them. The probability of this event, given that this person is a member of $(G)$ subset and given the environmental conditions of $(E_T)$, is represented as $P(\{E_i\}, G_i \mid G, E_T)$. If the event $(G_i)$ occurs without $\{E_i\}$, then whatever exposure does occur, it is insufficient, and the $i^{th}$ individual cannot develop MS.

However, the relationship between one individual's family of "*sufficient*" exposures to that of others may be complex. For example, every "*i-type*" group may have a family with sets unique to that group or, alternatively, the families for any two or more individuals (not in the same "*i-type*" group) may overlap to any degree, even to the point where their families are almost identical. However, if *every* susceptible individual has an *identical* family of "*sufficient*" environmental exposures, then: $\forall(i): P(\{E_i\}, G_i | G, E_T) = P(\{E_i\} | G, E_T)$; and everyone is a member of the same "*i-type*" group. If some individuals can develop MS under *any* environmental condition, then, for these individuals: $P(\{E_i\}, G_i | G, E_T) = P(G_i | G, E_T)$. And, finally, if there are $(s_e)$ specific sets of environmental exposure $(e = 1, 2, \ldots, s_e)$ that are "*sufficient*" to cause MS in *any* susceptible individual, then, for the family $\{E_e\}$ of these sets of environmental exposure:

$$\forall G_i \in G \; \& \; \forall(e) : P(\{E_e\}, G_i | G, E_T) = P(\{E_e\} | G, E_T)$$

*{NB: It may be that some of these $\{E_e\}$ environments, which are "sufficient" to cause MS in every susceptible individual, are so improbable (e.g., being inoculated with myelin basic protein together with complete Freund's adjuvant), that they never occur spontaneously. Even so, any individual who can only develop MS under these extreme environmental conditions, is still able to develop MS under some environmental conditions and, thus, every such individual will be a member of the (G) subset. If anyone can develop MS under these extreme conditions, then everyone is a member of the (G) subset.}*

*Definition of the exposure (E).* Although an individual (genotype) may experience more than one set of environmental exposures, which may be part of one, or more than one, $\{E_i\}$ family, each individual's total environmental experience is unique to them. Therefore, we will represent the exposure event of interest $(E)$ as the union of the disjoint events, which exhibit the pairing of susceptible individuals with "*sufficient*" environments, such that:

$$(E) = (\{E_1\}, G_1) \cup (\{E_2\}, G_2) \cup \ldots \cup (\{E_m\}, G_m)$$

in which case:

$$P(E | G, E_T) = \sum\nolimits_{i=1}^{m} P(\{E_i\}, G_i | G, E_T)$$

or:

$$P\left(E | G, E_T\right) = \sum\nolimits_{i=1}^{m} P(G_i | G, E_T) * P(\{E_i\} | G_i, G, E_T)$$

Because genotype is assumed to be independent of the prevailing environmental conditions $(E_T)$:

$$\forall G_i \in G : P\left(G_i | G, E_T\right) = P\left(G_i | G\right) = \frac{P(G_i)}{P(G)} = \frac{(1/N)}{(m/N)} = 1/m$$

so that:

$$P\left(E | G, E_T\right) = (1/m) * \sum\nolimits_{i=1}^{m} P(\{E_i\} | G_i, G, E_T)$$

Thus, the term $P(E | G, E_T)$ represents the probability of the event that a member of the $(G)$ subset, selected at random, will experience an environmental exposure "*sufficient*" to cause MS in them, given their unique genotype and given the prevailing environmental conditions of the time $(E_T)$. Furthermore, from the definition of $(E)$, this event can only occur when the event

(*G*) also occurs, so that:

$$P(E, G) = P(E)$$

Notably, many environmental factors or events, which are part of a set within the $\{E_i\}$ family, may (and likely do) represent a range of environmental experiences. For example, suppose that, for the $i^{th}$ susceptible individual to develop MS, for one set of exposures, they need to experience a vitamin D deficiency of some minimum severity, lasting for some minimum amount of time, and occurring during some "critical" age-window. In this case, the definition for the environmental event of a "*sufficient*" vitamin D deficiency for this individual, for this set, presumably, would also include deficiencies of the same (or greater) severity, lasting the same (or a longer) amount of time, and occurring during the same (or more restrictive) age-window. In this circumstance, we can define a "critical exposure *intensity*" level as that vitamin D level, at (or above) which, the deficiency becomes "*sufficient*" for the $i^{th}$ (or an "*i-type*") individual. An expanded discussion of this notion of exposure "*intensity*" is presented elsewhere (*see Sections 6g & 8a, 8b in* S1 File).

Importantly, as noted previously, each set of "*sufficient*" environmental exposures is unspecified as to: 1) how many events or factors are involved; 2) when, during the life of an individual, these events or factors need to occur; 3) what these events or factors are; and 4) whether these factors need to be present or absent. Notably, this specification of a "*sufficient*" sets of exposures is completely agnostic with respect to whether these factors or events increase or decrease risk. For example, if behaving in some manner, or having some experience, protects the $i^{th}$ person from getting MS, then one or more of the "*sufficient*" sets of exposure for this person will include <u>not</u> behaving in this manner or <u>not</u> having this experience. Nevertheless, regardless of any such complexities, each of these sets, of whatever they consist, simply needs to be "*sufficient*", by themselves, to cause MS to develop in the $i^{th}$ (or an "*i-type*") susceptible individual. Thus, our <u>*definition*</u> of a "*sufficient*" set of exposures includes <u>*every*</u> environmental condition (known, suspected, or unknown), which is required (i.e., necessary) for such "*sufficiency*".

*Partitioning the environmental exposure.* In addition, any set of environmental exposures, for any individual, can be partitioned conceptually into three mutually exclusive subsets, which we term: ($E_{pop}$, $E_{sib}$, and $E_{twn}$). The subset ($E_{pop}$) includes all those environmental experiences or events equally likely to be shared by the population generally (including siblings and twins). The occurrence of ($E_{pop}$) represents the event that a specific environmental event or factor, which an individual experiences, is a member of the ($E_{pop}$) subbset. The subset ($E_{sib}$) includes all those environmental experiences or events either more or less likely to be shared by siblings (including twins) compared to the general population. The occurrence of ($E_{sib}$) represents the event that a specific environmental event or factor, which an individual experiences, is a member of the ($E_{sib}$) subset. Presumably, the ($E_{sib}$) environmental experiences occur mostly (but not necessarily exclusively) during childhood. The subset ($E_{twn}$) includes all those environmental experiences or events more or less likely to be shared by *MZ*- and *DZ*-twins compared both to non-twin co-siblings and to the general population. The the occurrence of ($E_{twn}$) represents the event that a specific environmental event or factor, which an individual experiences, is a member of the ($E_{twn}$) subbset. Presumably, the ($E_{twn}$) environmental events occur mostly (but not necessarily exclusively) during the intrauterine and early post-natal periods. Importantly, creating this partition does not imply that any of these experiences are unique to twins or siblings–*everyone* experiences each environmental component. The difference is that twins and siblings are more or less likely to *share* certain experiences.

Each of the $(v_i)$ sets of "*sufficient*" environmental exposures within the $\{E_i\}$ family can be partitioned into these three mutually exclusive events. Thus, for $(j = 1, 2, \ldots, v_i)$, the event $(E_{ij})$ represents the occurrence of the $j^{th}$ "*suffiicient*" set of exposures within the $\{E_i\}$ family. The event $(E_{ij})$ can then be represented as the union of these three disjoint events such that:

$$\forall (j = 1, 2, \ldots, v_i) : E_{ij} = (E_{twn})_{ij} \cup (E_{sib})_{ij} \cup (E_{pop})_{ij}$$

In this circumstance, the probability of each event $(E_{ij})$ is the joint probability of these three independent component events such that:

$$P\left(E_{ij}\right) = P\{(E_{twn})_{ij}, (E_{sib})_{ij}, \left(E_{pop}\right)_{ij}\} = P(E_{twn})_{ij} * P(E_{sib})_{ij} * P\left(E_{pop}\right)_{ij}$$

and the event $\{E_i\}$ is represented as:

$$\{E_i\} = (E_{i1}) \cup (E_{i2}) \cup \ldots \cup (E_{iv_i})$$

The same applies to every $\{E_i\}$ family–i.e., $\forall (i): (i = 1, 2, \ldots, m)$.

*{NB: Most, if not all, environmental exposures are "population-wide" in the sense that the risk of these events is shared by everyone. For example, the amount of sunlight reaching the Earth's surface in a particular region can be considered a "population-wide" exposure in the sense that the same amount of sunlight is "available" to everyone in that region. Despite this, however, there may be certain individuals or certain subgroups within the population who experience less sun-exposure than others (e.g., if they disproportionatley use sun-screen, if they disproportionatley avoid the sun, or if they are otherwise disproportionatley protected from sun-exposure). Conversely, there may also be certain individuals or groups who experience more sun-exposurre than others. However, given the fact that a co-twin (or a non-twin co-sibling) experiences such an imbalance, unless their proband twin (or proband sibling) is either more or less likely to to experience a similar imbalance compared to others, then these exposures would still be part of the $(E_{pop})$ environment. Also, the $(E_{sib})$ environment may include experiences outside the childhood micro-environment if, for example, sharing the same biological mother made the intra-uterine environment more similar for siblings than that for the general population. In addition, if twins disproportionately shared certain childhood or adult experrriences more so than other siblings or the general population, then these experiences woud be part of the $(E_{twn})$ environment.*

*Although it is unspecified as to what experiences consitiute each subset, nevertheless, these three subsets of environmental exposure ($E_{twn}$, $E_{sib}$, and $E_{pop}$) are envisioned to be mutually exclusive and that, together, they comprise any idividual's unique environmental experience. Thus, as noted above, every individual experiences each of these components of enviornmental exposure, regardless of whether they are twins or non-twin siblings and regardless of whether they are members of the $(G)$ subset. For example, even though the <u>same</u> intrauterine environment is shared by twins, everyone experiences <u>some</u> intrauterine environment. Similarly, although both twins and non-twin co-siblings experience a <u>similar</u> childhood environment, everyone experiences <u>some</u> childhood environment. Nevertheless, in considering these components of environmental exposure as they relate to the suffucent sets as described above–i.e for $(i = 1, 2, \ldots, m)$ and for $(j = 1, 2, \ldots, v_i)$, we are here focused on the events $(E_{ij})$, for which, during any Time Period, it will be the case that:*

$$P(E_{ij}) = P(E_{twn})_{ij} * P(E_{sib})_{ij} * (E_{pop})_{ij}$$

*Notably, during any specific Time Period $(E_T)$, both $P(E_{ij})$ and its component parts are constants. In this conceptualization, however, each successive Time Period $(E_T)$ are envisioned to overlap with each other. For example, suppose that all of the relevant ($E_{twn}$, $E_{sib}$, and $E_{pop}$)*

*exposures need to take place before the age of 30 years. In this circumstance, for a person born in 1975, ($E_T$) will represent the Time Period from 1975 to 2005. By contrast, for a person born in 1980, ($E_T$) will represent the overlapping Time Period from 1980 to 2010.}*

*Impact of the ($E_{sib}$) environment.* Despite this conceptual framework, however, the observations from Canada in adopted individuals, in siblings and half-siblings raised together or apart, in conjugal couples, and in brothers and sisters of different birth order, have indicated that MS-risk is not affected by the familial micro-environment but suggest, rather, that the important environmental risks (not considering twins) result from exposures that are experienced *population-wide* [10–16]. Thus, these studies, collectively, provide compelling evidence for the absence of any ($E_{sib}$) environmental impact on MS.

*Relationships between, and limits relating to*: (*MS*), (*E*), *and* (*G*). It is clear from the definitions of environmental and genetic susceptibility (*above*) that, for the event of (*MS*) to occur, both the event (*G*) and the event (*E*) must also occur. If either of these events does not occur, the event (*MS*) cannot occur. Therefore:

$$P(MS, G, E \mid E_T) = P(MS, E \mid E_T) = P(MS, G \mid E_T) = P(MS \mid E_T)$$

Also, using the definitions in Table 2, and both from *Section 7b in* S1 File and from *Methods #1C & #1D* (*below*), it must be the case that, if, *currently*, {$P(E) \neq 1$}, then, also:

$$P(MS \mid E) > P(MS \mid MZ_{MS})_2$$

$$\boldsymbol{c} = P(MS \mid E, M) > P(MS \mid M, MZ_{MS})_2$$

and:

$$\boldsymbol{d} = P(MS \mid E, F) > P(MS \mid F, MZ_{MS})_2$$

**C. Circumstances relating to twins and siblings of individuals with MS.** The terms (*MZ*), (*DZ*), and (*S*) represent, respectively, the subsets of *MZ*-twins, *DZ*-twins, and non-twin sibships within (*Z*). The occurrence of (*MZ*), (*DZ*), or (*S*) represent the events that an individual, selected at random from (*Z*), belongs, respectively, to each of these subsets and the terms $P(MZ)$, $P(DZ)$, $P(S)$ represent the respective probabilities of these events. For clarity, the randomly selected individual is always referred to as the "proband twin" or the "proband sibling" depending upon the subset to which they belong. The other member (or members) of the twinship or sibship are always referred to as the "co-twin(s)" or the "co-sibling(s)".

*Circumstances for twins and siblings of selected probands.* Initially, we will consider two events for an *MZ* twin-pair. The first is the event that the proband, randomly selected from (*Z*), is a member of the (*MS*, *MZ*) subset and that their co-twin is a member of the (*MZ*) subset. The second is the event that the proband, randomly selected from (*Z*), is a member of the (*MZ*) subset and that their co-twin is a member of the (*MS*, *MZ*) subset. Clearly, the probability of these two events is the same. Therefore, to distinguish the circumstances of the proband from those of the co-twin, we will use the term ($MZ_{MS}$) to indicate, specifically, the status of the co-twin. Thus, during the *Time Period* ($E_T$):

$$P(MZ_{MS} \mid E_T) = P(MS \mid MZ, E_T)$$

where {$P(MS \mid MZ, E_T)$} represents the probability of the event (*MS*) in the proband twin during ($E_T$) and {$P(MZ_{MS} \mid E_T)$} represents the same probability for the event (*MS*) in the co-twin

during ($E_T$). Also, because any two *MZ*-twins have "identical" genotypes, therefore:

$$\forall G_i \in G : P(G_i, MZ_{MS} | E_T) = P(MS, G_i | MZ, E_T)$$

and:

$$\forall G_i \in G : P(MZ_{MS} | G_i, E_T) = P(MS | G_i, MZ, E_T)$$

In which case:

$$\forall G_i \in G : P\left(G_i | MZ_{MS}, E_T\right) = \frac{P\left(G_i, MZ_{MS} | E_T\right)}{P\left(MZ_{MS} | E_T\right)} = \frac{P(MS, G_i | MZ, E_T)}{P(MS | MZ, E_T)}$$

$$= P\left(G_i | MS, MZ, E_T\right)$$

Consequently, *every* proband who has an *MZ* co-twin in the (*MS*, *MZ*) = (*MS*, *MZ*, *G*) subset and who shares an "identical" genotype with their co-twin, *must* also be a member of the (*G*) subset. Therefore, summing over all susceptible individuals:

$$\sum_{i=1}^{m} P\left(G_i, MZ_{MS} | E_T\right) = P(G, MZ_{MS} | E_T) = P\left(MZ_{MS} | E_T\right)$$

and:

$$\sum_{i=1}^{m} P\left(G_i | MZ_{MS}, E_T\right) = P(G | MZ_{MS}, E_T) = P(G | MS, MZ, E_T) = 1$$

Similarly, the term $P(MZ_E)$ represents the probability of the event that the co-twin of an *MZ*-twin proband, randomly selected from (*Z*), is a member of the (*MZ*, *E*) subsets. Thus:

$$P(MZ_E | E_T) = P(E | MZ, E_T)$$

In an analogous manner, for *DZ*-twinships, the status of the co-twin is indicated by the subsets and the events of: ($DZ_{MS}$) and ($DZ_E$). And for non-twin sibships, the status of the co-sibling is indicated by the subsets and events of: ($S_{MS}$) and ($S_E$).

Thus, the two terms, $P(MS | MZ_{MS}, E_T)$ and $P(MS | DZ_{MS}, E_T)$ represent the conditional lifetime probability of the event that an individual (the proband), randomly selected from (*Z*), is a member of the either the (*MS*, *MZ*) or the (*MS*, *DZ*) subset, given the fact that their co-twin also belongs, respectively, to the (*MS*, *MZ*) or the (*MS*, *DZ*) subset, and given the prevailing environmental conditions of the time ($E_T$). These probabilities are estimated by the proband-wise concordance rate for either *MZ*- or *DZ*-twins [17]. This rate is calculated based on the number of concordant twin-pairs ($C_{TP}$) compared to the number of discordant twin-pairs ($D_{TP}$) and adjusted based upon the degree to which twins are "doubly ascertained". The term "doubly ascertained", in this context, represents the proportion of twin-pairs, for whom both twins were independently identified by the initial ascertainment scheme [17]. If all twin-pairs are "doubly ascertained" by this scheme, and if the sample from (*Z*), so ascertained, is random, then the formula for calculating the proband-wise concordance rate is:

$$MZ - Twin\ concordance\ rate = \frac{(2*C_{TP})}{(2*C_{TP} + D_{TP})}.$$

However, if the probability of "double ascertainment" is less than unity, then this formula requires some modification [17]. In the Canadian data [6] the double-ascertainment rate for concordant *MZ*-twins was 54.2% (13/24).

In a similar manner, the term $P(MS \mid S_{MS}, E_T)$ represents the conditional life-time probability of the event that an individual (the proband), randomly selected from ($Z$), is a member of the ($MS$, $S$) subset, given the fact that one or more of their non-twin co-siblings is a member of the ($MS$, $S$) subset and given prevailing environmental conditions of ($E_T$).

**D. Adjustments for the shared environment of twins.** Lastly, the term $P(MS \mid IG_{MS}, E_T)$, represents the proband-wise concordance rate for *MZ*-twins during ($E_T$)–i.e., $P(MS \mid MZ_{MS}, E_T)$–which has been adjusted for the fact that concordant *MZ*-twins, in addition to sharing their "identical" genotypes ($IG$), also disproportionately share their ($E_{twn}$) and ($E_{sib}$) environments with each other. Such an adjustment may be necessary because, if these disproportionately shared environmental experiences contribute to causing MS in the co-twin, they could also increase the likelihood of MS developing in the proband twin and such a circumstance could, potentially, alter any conclusions regarding the nature of genetic susceptibility in the population (*see Section 1a*, *1b in* S1 File, *for a discussion of why*, *and a development of how*, *this adjustment is made*).

**E. Characterizing genetic susceptibility to MS in a population.** From these *Model* specifications and definitions, we can use estimated values for observable population parameters to deduce the value of the non-observable parameter $P(MS \mid G, E_T)$, which represents the probability of the event that an individual, randomly selected from ($Z$), will develop MS over the course of their lifetime, given that they are a member of the ($G$) subset and given the prevailing environmental conditions of ($E_T$). From the definition of ($G$), as noted in *Section #1A* (*above*):

$$P(MS, G \mid E_T) = P(MS \mid E_T)$$

Therefore, because genotype is assumed to be independent of the prevailing environmental conditions of ($E_T$):

$$P(MS \mid G, E_T) = P(MS, G \mid E_T)/P(G \mid E_T) = P(MS \mid E_T)/P(G)$$

Rearrangement of this equation, yields:

$$P(G) = P(MS \mid E_T)/P(MS \mid G, E_T)$$

Consequently, the value of $P(G)$ can be estimated using the observed data from <u>any</u> specific *Time Period* ($E_T$)–including ours–during which: $P(MS \mid E_T) > 0$. Thus, the parameter $P(G)$ can be estimated regardless of whether <u>some</u> susceptible individuals have <u>no</u> chance of developing MS under the environmental conditions of ($E_T$). Therefore, considering only our "*current*" *Time Period*, this equation can be simplified to yield:

$$P(G) = P(MS)/P(MS \mid G)$$

Moreover, once the value of $P(G)$ is established, it can then be used to assess the nature of MS pathogenesis. For example, if: {$P(G) = 1$}–i.e., if the *penetrance* of MS for the population ($Z$) is the same as that for the subset ($G$)–then anyone can get MS under the appropriate environmental conditions. By contrast, if: {$P(G) < 1$}–i.e., if the *penetrance* of MS for the subset ($G$) is greater than that for the population ($Z$)–then only certain individuals in ($Z$) have any possibility of getting MS. Thus, a finding of: {$P(G) < 1$} would exclude any possibility that MS ever occurs in someone who lacks a genetic predisposition to getting the disease. In this sense (and in this case), MS must be considered a "genetic" disorder (i.e., unless a person has the appropriate genotype, they have no chance, whatsoever, of getting MS, regardless of their environmental exposure). Importantly, even if MS is "genetic" in this sense, this has no bearing

upon whether disease pathogenesis also requires the co-occurrence of specific environmental events.

In this analysis, two basic *Models* are used to estimate the values of various unknown epidemiological parameters of MS. The first *Model* takes a cross-sectional approach, in which deductions are made from the epidemiological data obtained during a single *Time Period* (i.e., the "*current*" *Time Period*). This will be referred to as the *Cross-sectional Model* (*see* Methods *#3; below; see also Section 3a, 3b in* S1 File). The second *Model* takes a longitudinal approach, in which deductions are made both from the "*current*" epidemiological data and from the observed changes in MS epidemiology that have taken place over the past 4–5 decades [3, 6]. This will be referred to as the *Longitudinal Model* (*see* Methods *#4; below; see also Section 4a–4c; in* S1 File). These two *Models* are independent of each other although both incorporate many of the same observed and non-observed epidemiological parameters, which are important for MS pathogenesis. The *Cross-sectional Model* derives theoretical relationships between different epidemiologic parameters, but it also makes two assumptions regarding *MZ*-twin data to establish these relationships. These two assumptions are also commonly made by other studies, which analyze *MZ*-twin data, and each has observational data to support them [18–20]. Nevertheless, for the derivations for Eqs 2a–2d (Methods *#3; below*), these conditions need to be assumed (*see Section 3a in* S1 File). By contrast, the *Longitudinal Model* does not make either of these assumptions to estimate possible ranges for the non-observed parameters and several possible conditions for this *Longitudinal Model* are depicted in Figs 1–4.

For both *Models*, the first step is to assign acceptable ranges for the value of certain "observed" parameters (e.g., twin and sibling concordance rates, the population prevalence of MS, or the proportion of *women* among MS patients). These ranges are assigned such that they always include their calculated 95% *CIs*. However, for certain parameters, the ranges considered plausible are expanded beyond the limits set by the *CIs*. The second step is to assign acceptable ranges for the "non-observed" parameters (e.g., the proportion of susceptible persons in the population or the proportion of *women* among susceptible individuals). These ranges are assigned such that they cover the entire "plausible" range for each such parameter. In both *Models*, a "substitution" analysis is undertaken to determine those parameter combinations (i.e., solutions), that fit within the acceptable ranges for both the observed and non-observed parameters. These solutions are then used to assess their implications about the basis of genetic and environmental susceptibility to MS in the population. For each *Model*, the total number of parameter combinations interrogated in this manner was $\sim 10^{11}$.

## 2. Establishing plausible ranges for parameter values

**A. Observed parameter values.** For notational simplicity, we sometimes use subscripts (*1*) and (*2*) to indicate the parameter values at *Time Period #1* and *Time Period #2* {e.g., $P(MS)_2 = P(MS \mid E_T)$ at *Time Period #2*}. For the purposes of this analysis, those parameter-values observed for persons born between 1976 and 1980 (i.e., *Time Period #2*) are always taken to be the "*current*" values. When only this *Time Period* is being considered, the terms ($E_T$) and the subscript (*2*) are generally omitted entirely to simplify the notation.

{*NB*: *In general, for individuals born during Time Period #2 (1976–1980), their MS status cannot be determined until 25–35 years later (i.e., 2001–2015). The estimates of other epidemiological parameters are from reports in the Time Period of (2001–2015), which is also when the Time Period #2 (F:M) sex ratio is reported [6, 7, 11–15]. For this reason, Time Period #2 is considered fixed as the "current" period. However, because the (F:M) sex ratio has increased between every previous 5-year epoch and Time Period #2 [6], the choice of any specific Time Period #1 is*

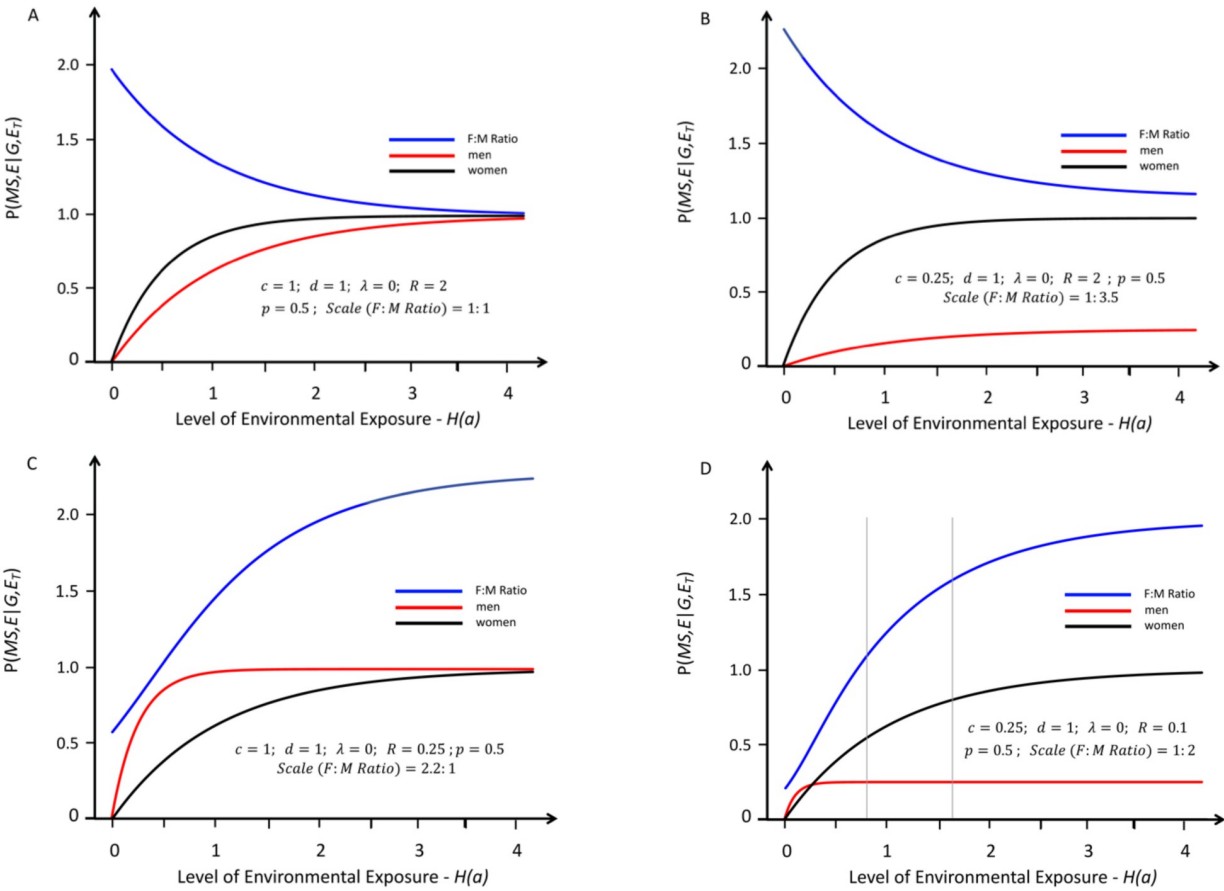

**Fig 1. Response curves representing the likelihood of developing MS in genetically susceptible *women* (black lines) and *men* (red lines) with an increasing probability of a "*sufficient*" environmental exposure–*see* Methods *#1B*.** The curves depicted are "strictly" proportional, meaning that the environmental threshold is the same for both *men* and *women*–i.e., under conditions in which: $(\lambda = 0)$–*see Text*. The blue lines represent the change in the (*F:M*) *sex ratio* (plotted at various scales, indicated in each *Figure*) with increasing exposure. The thin grey vertical lines represent the portion of the response curve that covers the change in the (*F:M*) *sex ratio* from 2.2 to 3.2 (i.e., the actual change observed in Canada [6] between *Time Periods #1 & #2*). The grey lines are omitted under circumstances either where these observed (*F:M*) *sex ratios* are not possible or where both $(Zw > Zm)$ and an increasing (*F:M*) *sex ratio* are not possible. Response curves *A* and *B* reflect conditions in which $(R > 1)$; whereas curves *C* and *D* reflect conditions in which $(R < 1)$. If $(R = 1)$, the blue line would be flat. Response curves *A* and *C* reflect conditions in which $(c = d = 1)$; whereas curves *B* and *D* reflect those conditions in which $(c < d = 1)$. Under the conditions for curves *A* and *B* $(R \geq 1)$, there is no possibility that the (*F:M*) *sex ratio* will be observed to increase with increasing exposure. Under the conditions of curve *C*–i.e., $(c = d = 1)$ and $(R < 1)$–at no exposure level is it possible that: $Zw = P(MS, E \mid G, F, E_T) > P(MS, E \mid G, M, E_T) = Zm$. Thus, the only "strictly" proportional model that could possibly account for an increasing (*F:M*) *sex ratio*, and for the fact that: $(Zw_2 > Zm_2)$, is a *Model* in which $(c < d \leq 1)$ and $(R < 1)$–i.e., curve *D*.

equivalent to any other. For our Time Period #1, we chose the 5-year epoch (1941–1945) because it was the earliest epoch with the narrowest CI [6].}

The 2010 Canadian census [8], reported that the proportion of *women* among the general Canadian population $(Z)$ is 50.4%. Thus, *men* and *women* comprise essentially equal proportions of this population and, therefore, the probabilities of the events that an individual, randomly selected from $(Z)$, is a *woman* or a *man*–$P(F)$ and $P(M)$, respectively–are each ~50%. Therefore, by definition:

$$P(F, G) = P(F \cap G) \leq P(F) = 0.5$$

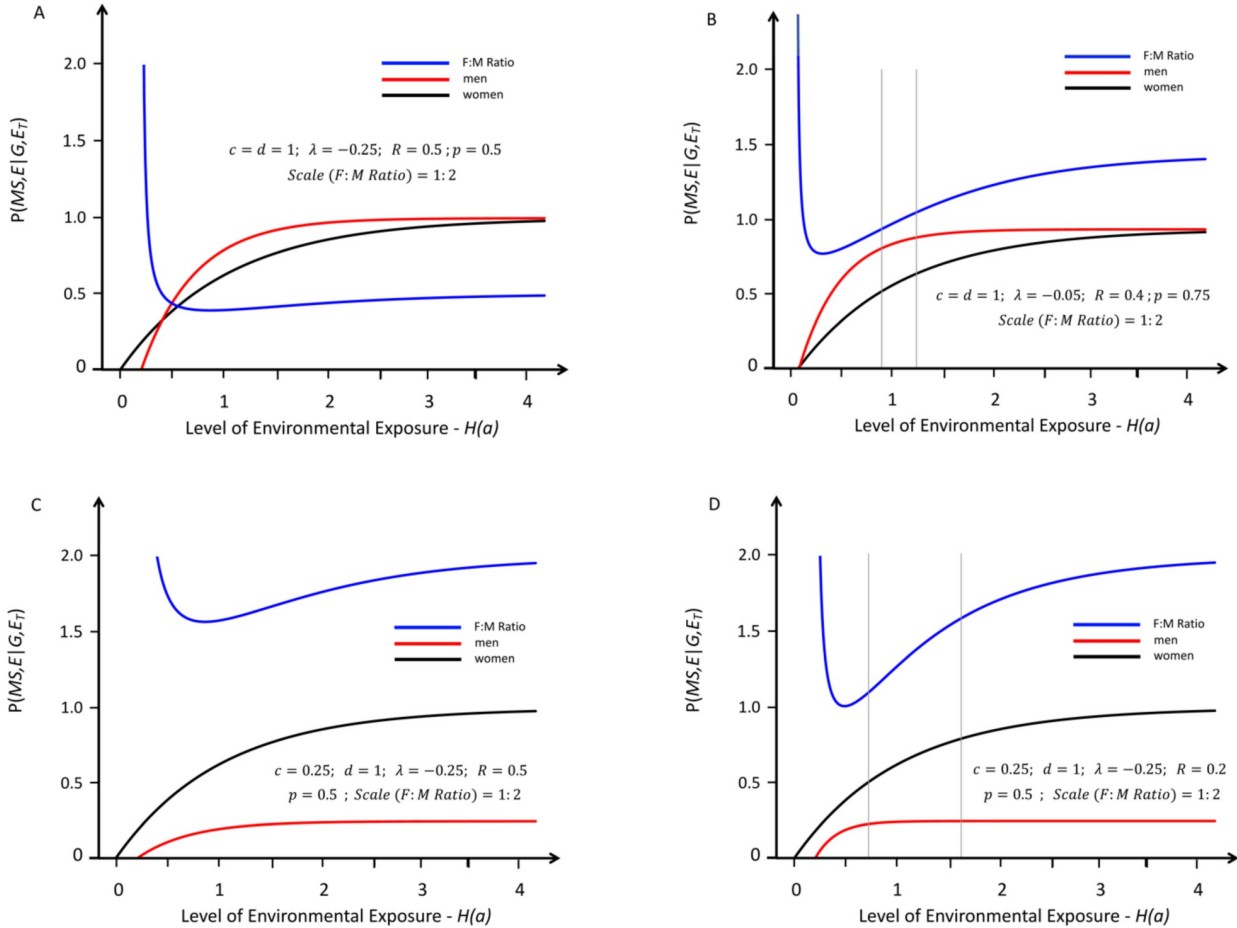

**Fig 2. Response curves for the likelihood of developing MS in genetically susceptible *women* (black lines) and *men* (red lines) with an increasing probability of a "*sufficient*" environmental exposure–*see* Methods #1B.** Like Fig 1, the curves depicted are also proportional although here the environmental threshold is greater for *men* than for *women*–i.e., under conditions in which: ($\lambda < 0$)–*see Text*. The blue lines represent the change in the (*F:M*) *sex ratio* (plotted at various scales, indicated in each *Figure*) with increasing exposure. The thin grey vertical lines represent the portion of the response curve that covers the change in the (*F:M*) *sex ratio* from 2.2 to 3.2 (i.e., the actual change observed in Canada [6] between *Time Periods #1 & #2*). The grey lines are omitted under circumstances where these observed (*F:M*) *sex ratios* are not possible. Response curves *A* reflects conditions in which ($c = d = 1$) & ($R > 1$); Response curves *B* reflects conditions in which ($c = d = 1$), ($R < 1$), & ($p \geq p'$); curves *C* reflect conditions in which ($c < d = 1$) and ($R < 1$) and curve *D* reflects those conditions in which ($c < d = 1$) and ($R < 0.5$). To account for the observed increase in the (*F:M*) *sex ratio*, curves *D* (compared to curves *C*) requires a small enough value of (*R*) so that the (*F:M*) *sex ratio* curve dips below 2.2 and, also, a small enough value of (*c*) so that the curve rises above 3.2. For all points in curves *A* after the intersection, and for all points in curves *B*, ($Zm > Zw$), which is not possible. Curves *C* never even approach the (*F:M*) *sex ratio* of 2.2. By contrast, for curves *D*, both an appropriate increase in the (*F:M*) *sex ratio* and ($Zw > Zm$), can be observed.

and:

$$P(M, G) = P(M \cap G) \leq P(M) = 0.5$$

The proband-wise concordance rate [7] for MS in *MZ*-twins, *currently* observed in Canada, is:

$$P(MS \mid MZ_{MS})_2 = 0.253$$

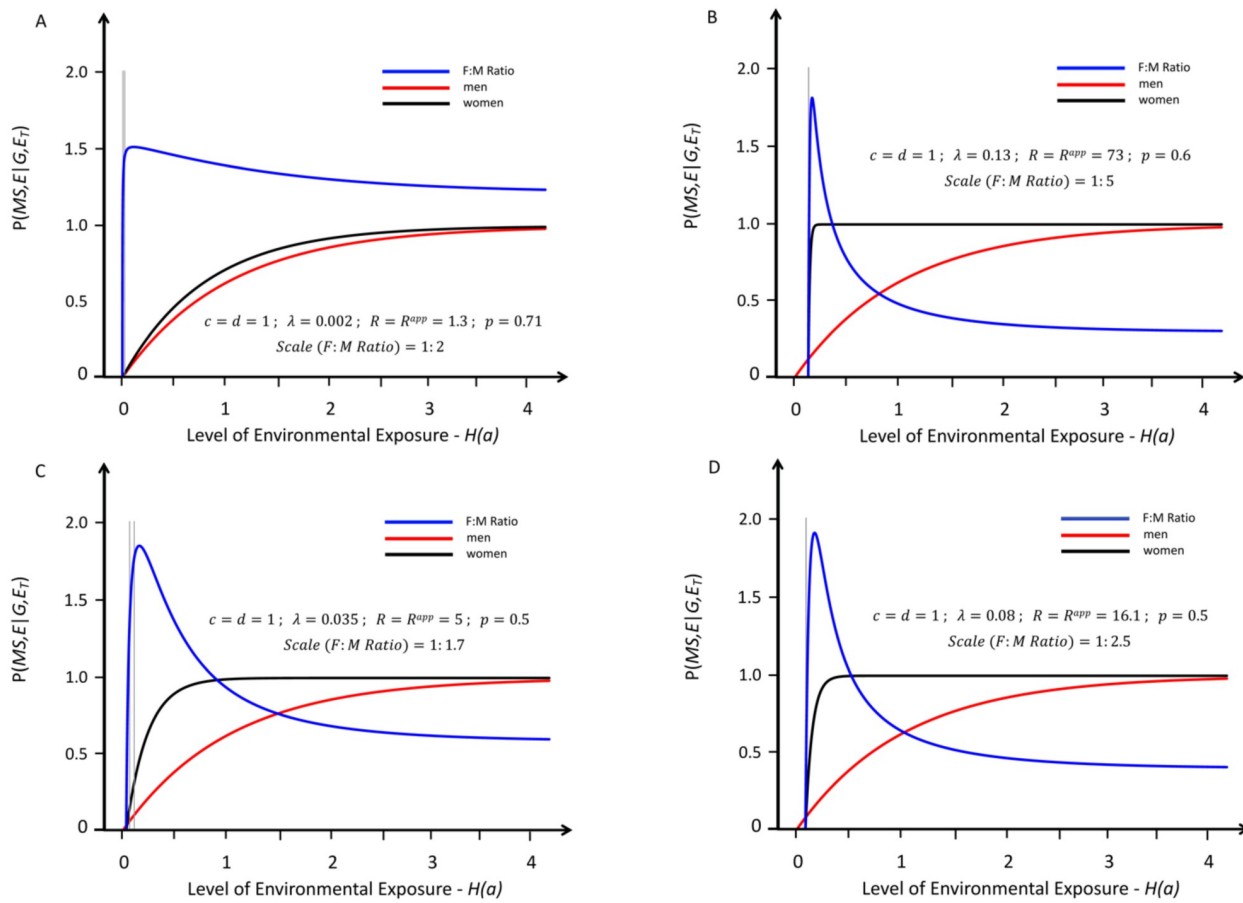

**Fig 3. Response curves for the likelihood of developing MS in genetically susceptible *women* (black lines) and *men* (red lines) with an increasing probability of a "*sufficient*" environmental exposure–*see* Methods #1B.** Like Fig 1, the curves depicted are also proportional ($R = R^{app}$), but, for these, the environmental threshold in *women* is greater than that it is in *men*–i.e., these are conditions in which: ($\lambda > 0$). Also, all these response curves represent actual solutions and reflect conditions in which ($c = d = 1$) and, as discussed in *Methods #4C*, are representative of all conditions in which ($c = d < 1$). Moreover, with increasing values from ($R^{app} \geq 1.3$), which is the minimum value of ($R^{app}$) for any solution–which is depicted in *Fig A*. The blue lines represent the change in the ($F$:$M$) *sex ratio* (plotted at various scales, indicated in each *Figure*) with increasing exposure. The thin grey vertical lines represent the portion of the response curve (for the depicted solution), which represents the actual change in the ($F$:$M$) *sex ratio* that occurred between *Time Periods #1 & #2*). To account for the observed increase in the ($F$:$M$) *sex ratio*, these curves require the Canadian observations [6] to have been made over a very small portion the response curve–i.e., for most of these response curve, the ($F$:$M$) *sex ratio* is decreasing. Also, for each of these response curves, including the maximum difference in the environmental threshold (i.e., $\lambda \leq 0.13$) under conditions of ($c = d = 1$), which is depicted in *Fig B*, the ascending portion of the curve (which reflects and increasing *F*:*M sex ratio*) is very steep–a circumstance indicating that the portion of the response curve available for fitting the Canadian data [6] is quite narrow. Also, the intersection of the response curves does not occur as early as seems to be implied by an extension of the conditions of *Panels C–B*. Also, such a rapid transition from an MS that is "*male-predominant*" to an MS, which is "*female-predominant*" would seem to fit poorly with the gradual transition, which has taken place over the past two centuries [3, 6, 22–30, 40, 77, 78, 88].

with ($n = 146$) twin-pairs included in this estimation [7]. Therefore, the 95% *CI* for this parameter, calculated from an exact binomial test [21], is:

$$CI = (0.18 - 0.33)$$

The estimates, from different studies, for the "*current*" proportion of *women* among the MS patients–i.e., $P(F|MS)_2$ –in North America ranges between 66% and 76% [3]. For the *Cross-sectional Model*, we expanded the "plausible" range beyond the 95% *CI* calculated from

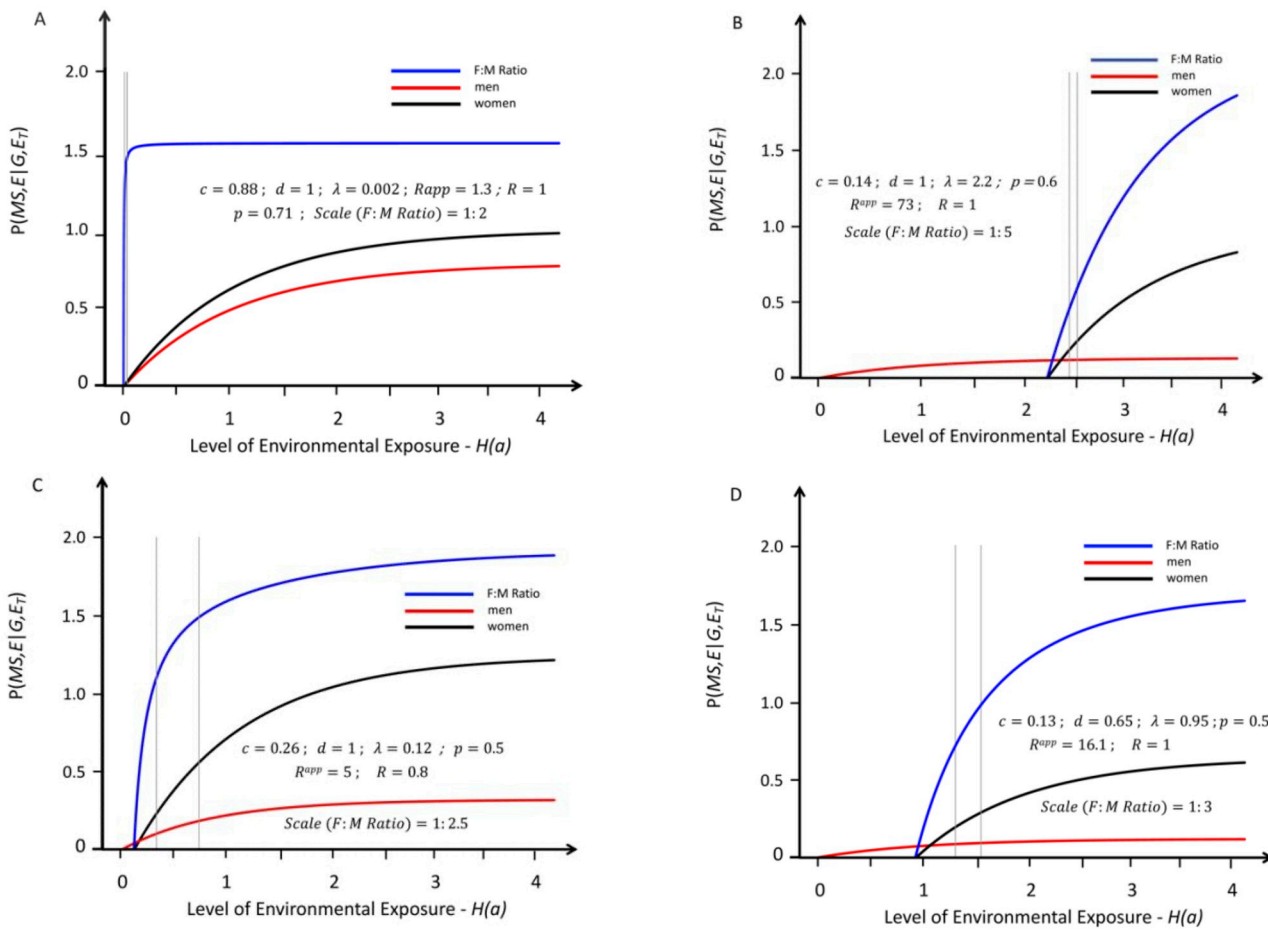

**Fig 4. Response curves for the likelihood of developing MS in genetically susceptible *women* (black lines) and *men* (red lines) with an increasing probability of a "*sufficient*" environmental exposure–*see* Methods *#1B*.** Like Fig 1, the curves depicted are also proportional ($R \leq 1$), but, for these, the environmental threshold in *women* is greater than that it is in *men*–i.e., these are conditions in which: ($\lambda > 0$). Also, these curves represent the same solutions as those depicted in Fig 3 except that these are for conditions in which ($c < d \leq 1$). The blue lines represent the change in the (*F:M*) *sex ratio* (plotted at various scales, indicated in each *Figure*) with increasing exposure. The thin grey vertical lines represent the portion of the response curve (for the depicted solution), which represents the actual change in the (*F:M*) *sex ratio* that occurred between *Time Periods #1 & #2*). Unlike the curves presented in Fig 3, however, an increase in the (*F:M*) *sex ratio* with increasing exposure is observed for any two-point interval along the entire response curves and, except for *Fig A*, the grey lines are clearly separated.

"*current*" Canadian data presented *below* [6]. Thus, for this *Model*, we considered the range:

$$0.66 \leq P(F \mid MS)_2 \leq 0.78$$

The reason for this is because the "*current*" estimated range from the Canadian study is quite narrow and some solutions, which fall within the range of different estimates from other locations in North America [3], might be excluded. This choice permits a wider range of possibilities to be considered as solutions for our *Cross-sectional Model*.

By contrast, for our *Longitudinal Model*, because we were interested specifically in how the parameter $P(F \mid MS)$ has changed for the Canadian population over time [6], we used the 95% CIs (from this single study) to estimate the ranges for this parameter value during each *Time Period*. For example, the proportion of *women* among the MS patients in Canada was 69% for patients born during *Time Period #1* (1941–1945) and this proportion was significantly less

($p < 10^{-6}$) than the 76% observed for patients born during the "*current*" *Time Period #2* (1976–1980) [6]. Although the authors of this study, do not report the actual numbers of individuals in each 5-year epoch, they do report that their 5-year samples averaged 2, 400 individuals per epoch [6]. Also, the authors graphically present the 95% *CIs* for the (*F:M*) *sex ratio* during each of these 5-year epochs in the *Figure* of their manuscript [6]. Estimating that the number of individuals in both *Time Periods #1* and *#2* is ~2, 000, and using an exact binomial test [21], for our *Longitudinal Model*, we estimate that:

$$P(F \,|\, MS)_1 : CI = (0.67 - 0.71)$$

$$P(F \,|\, MS)_2 : CI = (0.74 - 0.78)$$

Both of these ranges exceed those (based on the 95% *CIs*) presented in the *Figure* of the manuscript [6].

The "*current*" proband-wise concordance rates for MS in *female* and *male MZ*-twins, observed in Canada, are 34% and 6.5%, with the total number of *female* and *male* twin-pairs included in these calculations being ($n_1 = 100$) and ($n_2 = 46$), respectively [7]. Using an exact binomial test [21], and using the definitions provided in Table 2, the *CIs* for these observations are:

$$y'_1 = P(MS \,|\, F, MZ_{MS})_2 = 0.34 CI = (0.24 - 0.44)$$

$$y'_2 = P(MS \,|\, M, MZ_{MS})_2 = 0.065 CI = (0.014 - 0.18)$$

The 95% *CI* for the difference in *MZ*-twin concordance between *men* and *women* is calculated as:

$$CI = (y'_1 - y'_2) \pm 1.96 \sqrt{\frac{y'_1 (1 - y'_1)}{n_1} + \frac{y'_2 (1 - y'_2)}{n_2}}$$

In which case:

$$y'_1 - y'_2 = 0.275 CI = (0.16 - 0.39)$$

This large and significant difference in the *current relative penetrance* values between susceptible *women* and *men* for their *MZ*-twin concordance rates, strongly suggests that, *currently*, the same *relative penetrance* also pertains to the (*F*, *G*) and (*M*, *G*) subsets (*see Section 2c in* S1 File). Therefore, we assume that:

$$Zw_2 = P(MS \,|\, F, G)_2 > P(MS \,|\, M, G)_2 = Zm_2$$

Previously, we used three independent methods (*based on observation*) to estimate the value of $P(MS)_2$ [3]. The first method relied on measures of the population prevalence of MS in North America together with the observed age-distribution for MS-onset, the second method considered the age-specific prevalence of MS in the age-band of 45–54 years, and the third method considered a *population-based* multiple-cause-of-death study from British Columbia, which reported the proportion of death certificates that mention MS. The parameter-value range supported, collectively, by these different methods was: $0.0025 \leq P(MS)_2 \leq 0.0046$ [3]. Nevertheless, for the purposes of the present analysis, we expanded the "plausible"

range for this parameter to include:

$$0.001 \leq P(MS)_2 \leq 0.006$$

**B. Non-observed parameter values.** In addition to the observed parameter values (*above*), and using the definitions in (Table 2), we determined acceptable values for 12 additional parameters: $P(G)$; $p = P(F | G)$; $x = P(MS | G)_2$; $x' = P(MS | IG_{MS})_2$; $x_1 = P(MS | F, G)_2$; $x_1' = P(MS | F, IG_{MS})_2$; $x_2 = P(MS | M, G)_2$; $x_2' = P(MS | M, IG_{MS})_2$ and the ratios: $(s_a = P(MS | DZ_{MS})_2)/P(MS | S_{MS})_2$; $C = P(MS)_1/P(MS)_2$; $r = x_1'/x_1$; and: $s = x_2'/x_2$.

Most of these parameters vary depending upon the level of exposure–i.e., all except $P(G)$ and $P(F | G)$. Therefore, the acceptable ranges were estimated for the "*current*" *Time Period #2*. In several cases, there are constraints on the values that these non-observed parameters can take. For example, $P(MS)$ has been observed to be increasing, especially (but not only) among *women*, in many parts of the world between the two *Time Periods* [6, 22–30]. Therefore, the parameter ($C$) is constrained in three ways. First, it must be that:

$$1.\, C = P(MS)_1/P(MS)_2 < 1$$

Second and third, on theoretical grounds (*see Section 7a in* S1 File), the value of ($C$) is also constrained such that:

$$C < P(M | MS)_2/P(M | MS)_1$$

and:

$$C < P(F | MS)_2/P(F | MS)_1$$

In this case, using the limits, provided earlier, for the proportion of *women* among MS patients during different *Time Periods*, the ratio ($C$) is at its maximum possible value when:

$$P(M | MS)_1 = 0.29 \text{ and } P(M | MS)_2 = 0.26$$

Therefore, on these theoretical grounds, the value of ($C$) is further constrained such that:

$$2.\, C < P(M | MS)_2/P(M | MS)_1 = 0.26/0.29 = 0.90$$

and:

$$3.\, C < P(F | MS)_2/P(F | MS)_1 = 0.74/0.71 = 1.04$$

Only *Constraint #2* (*above*) satisfies all three, so that the maximum upper bound for ($C$) is 0.90. Nevertheless, the actual upper bound for ($C$) will depend upon the values that $P(M | MS)_1$ and $P(M | MS)_2$ take for in any specific solution. Moreover, if ($C < 0.25$) then there must have been a greater than 4-fold increase in $P(MS)$ for Canada, which has taken place over a 35–40 year-interval. This seems to be an implausibly large increase based on the available data [6, 22–30]. Therefore, we conclude that:

$$0.25 \leq C < 0.9$$

Because MS develops in some individuals, the parameter $P(G)$ cannot be equal to 0. Also, because both *women* and *men* can develop MS, the parameter $P(F | G)$ cannot be equal to either

0 or 1. Therefore, the plausible ranges for these parameters are:

$$0 < P(G) \leq 1 \text{ and } : 0 < P(F \mid G) < 1$$

Furthermore, the *penetrance* of MS for the subsets $(F, G)$ and $(M, G)$ can be expressed (*see above*) such that:

$$x_1 = Zw = P(MS \mid F, G) = P(MS, E \mid F, G)$$

$$x_2 = Zm = P(MS \mid M, G) = P(MS, E \mid M, G)$$

Because everyone who develops MS must be a member of the $(G)$ subset, therefore, considering only these subsets, the ratio of *women* to *men*, during any *Time Period* can be expressed as:

$$(F : M) \text{ sex ratio} \} = P(F, MS)/P(M, MS) = P(F, MS, G)/P(M, MS, G)$$

or:

$$(F : M) \text{ sex ratio} \} = \{P(MS \mid F, G)/(MS \mid M, G)\} * \{P(F \mid G)/P(M \mid G)\}$$

Consequently, using the definitions provided *above* and in Table 2, the parameters $(x_1)$, $(x_2)$, $(x)$, $(p)$ and $(p')$, during any *Time Period* for the gender partition, are related such that:

$$(F : M) \text{ sex ratio} = \left(\frac{x_1}{x_2}\right) * \left\{\frac{p}{1-p}\right\} = \frac{p'}{1-p'} \tag{1a}$$

or, equivalently:

$$(F : M) \text{ sex ratio} = \left(\frac{Zw}{Zm}\right) * \left\{\frac{p}{1-p}\right\} = \frac{p'}{1-p'} \tag{1b}$$

$$x = px_1 + (1-p)x_2 \tag{1c}$$

$$p' = px_1/x \tag{1d}$$

These relationships require no assumptions and $(x_1)$, $(x_2)$, $(x)$, $(p)$ and $(p')$ must <u>*always*</u> satisfy Eqs 1a–1d during <u>*any*</u> *Time Period*, regardless of which *Model* is employed in the analysis [3].

Based on theoretical considerations (*see Section 1b in* S1 File) for the parameter $(s_a)$–*see below*–we demonstrate that: $(s_a \geq 1)$. Indeed, this relationship is confirmed observationally, where the recurrence risk of MS for a proband with a co-twin who is a member of the $(DZ_{MS})$ subset, is consistently reported to be greater than the recurrence risk of MS for a proband sibling with a co-sibling, who is a member of the $(S_{MS})$ subset [7, 31–37]. Therefore, from the definitions of $(MZ_{MS})$ and $(IG_{MS})$–*see Sections 1b & 7b in* S1 File–it must be the case that, during

our "*current*" *Time Period*:

$$x \leq x' \leq P(MS \mid MZ_{MS})_2$$

$$x_1 \leq x_1' \leq P(MS \mid F, MZ_{MS})_2$$

$$x_2 \leq x_2' \leq P(MS \mid M, MZ_{MS})_2$$

Using the constraints (*above*) on $P(MS \mid MZ_{MS})_2$, $P(MS \mid F, MZ_{MS})_2$ and $P(MS \mid M, MZ_{MS})_2$, therefore, the plausible ranges for $(x)$, $(x_1)$ & $(x_2)$ during the $2^{nd}$ *Time Period* are:

$$0.001 \leq x \leq x' \leq 0.33$$

$$0.001 \leq x_1 \leq x_1' \leq 0.44$$

and:

$$0.001 \leq x_2 \leq x_2' \leq 0.18$$

As noted above (Methods #1D), the observed *MZ*-twin concordance rate may be increased due to the fact that the proband disproportionately shares the $(E_{twn})$ and $(E_{sib})$ environments with a co-twin who has (or will develop) MS. Notably, however, any such impact (if it exists) must represent an environmental influence. Therefore, the maximum probability of developing MS for susceptible individuals under optimal environmental conditions–i.e., $P(MS \mid E)$– must be greater than the *currently* observed *MZ*-twin concordance rates (*see Section 7b in* S1 File). Consequently, we can use the Table 2 notations, and the definitions of (*c*) and (*d*)–*see* Methods #4A; *below*–to demonstrate that, because, *currently*, $(Zw_2 > Zm_2)$, and because both $P(MS)$ and $P(F \mid MS)$ are *currently* increasing, each of the following relationships must hold simultaneously:

$$P(MS \mid M, MZ_{MS})_2 < \boldsymbol{c} = P(MS \mid M, E, G) \leq \boldsymbol{d}$$

and:

$$P(MS \mid F, MZ_{MS})_2 < \boldsymbol{d} = P(MS \mid F, E, G) \leq 1$$

Notably, these relationships <u>include</u> the possibility that: $\boldsymbol{c} = \boldsymbol{d} = 1$

Finally, as discussed in (*Section 1c in* S1 File), we estimate the impact of the disproportionately shared $(E_{twn})$ and $(E_{sib})$ environments for *MZ*-twins as:

$$s_a = P(MS \mid DZ_{MS})_2 / P(MS \mid S_{MS})_2 \geq 1$$

and:

$$P(MS \mid IG_{MS})_2 = P(MS \mid MZ_{MS})_2 / s_a$$

In the Canadian data [7], the life-time probability of developing MS for the proband of a co-*DZ*-twin with MS (5.4%) was found to be greater than that for the proband of a non-twin co-sibling with MS (2.9%). From these observations, the point-estimate for $(s_a)$ becomes:

$$s_a = P(MS \mid DZ_{MS})_2 / P(MS \mid S_{MS})_2 = 0.054/0.029 = 1.86$$

This point estimate is approximately the same for both *men* and *women* (*see Section 1d in* S1 File). Thus, these observations from Canada suggest that sharing the $(E_{twn})$ environment

with a co-twin who develops MS markedly increases the likelihood of the proband twin developing MS for both *men* and *women* [3]. Nevertheless, it is possible that impact of these disproportionately shared environments may be over- or under-estimated by the Canadian data [7]. In any event, based on theoretical considerations (*see Section 1d in* S1 File), if we use the point-estimate from the Canadian data [7] that:

$$P(MS \,|\, IG_{MS})_2 \leq P(MS \,|\, MZ_{MS})_2 = 0.253$$

then:

$$P(MS \,|\, IG_{MS})_2 > P(MS \,|\, S_{MS})_2 = 0.029 = P(MS \,|\, MZ_{MS})_2 / 8.7$$

Therefore, for the purpose of both *Models*, we considered the plausible range for $(s_a)$ to be:

$$1 \leq s_a \leq 8.6$$

However, because, the point-estimate for $(s_a)$ from the Canadian data [7] is generally greater than that reported in other similar studies [31–37], we also considered, separately, the more restrictive circumstances, in which: $1 \leq s_a < 1.9$

## 3. Cross-sectional model

The *Cross-sectional Model* is developed in detail in (*Section 3a–3c in* S1 File). Because we are here considering only the "*current*" *Time Period #2*, the environmental designations relating to the conditions of the time–i.e., both the designation of $(E_T)$ and the use of the subscript (*2*)– have been eliminated from those parameter definitions that vary with the environmental conditions of the time (*see* Methods *#1A; above; see also* Table 2). Also, for simplicity of notation, we use the notation and definitions provided in *Methods #2B* (*above*) and in (Table 2); including the variance $(Var(X) = \sigma_X^2)$ of *penetrance* values for members of the $(G)$ subset.

We also make the following two assumptions.

### *Assumption #1*

Because *MZ*-twinning is generally thought to be non-hereditary [18–20], we assume that everyone in the population has the same *a priori* chance of having an *MZ*-twin and, thus, that:

$$\forall G_k \in Z : P(MZ \,|\, G_k) = P(MZ)$$

### *Assumption #2*

The *penetrance* of MS for a proband *MZ*-twin, whose co-twin is of unknown status, is assumed to be the same as if that genotype had occurred without having an *MZ* co-twin (i.e., the *penetrance* of MS for each genotype is independent of *MZ*-status). This assumption translates to assuming that the impact of experiencing any particular $(E_{twn})$ and $(E_{sib})$ environments together with an *MZ* co-twin is the same as the impact of experiencing the same $(E_{twn})$ and $(E_{sib})$ environments alone. Alternatively, it translates to the testable hypothesis that the mere fact of having an *MZ* co-twin does not alter the $(E_{twn})$ and $(E_{sib})$ environments in such a way that MS becomes more or less likely in both twins. Thus, we are here assuming that, for any *Time Period*:

$$\forall G_i \in G : P(MS \,|\, G_i, MZ) = P(MS \,|\, G_i)$$

Using these assumptions, we demonstrate in (*Section 3a–3c in* S1 File), that the following relationships hold:

$$P(IG_{MS}) = P(MZ_{MS}) = P(MS) \tag{2a}$$

$$x = (x'/2) \pm \sqrt{(x'/2)^2 - \sigma_X^2} \tag{2b}$$

$$x_1 = Zw = \frac{x + \sqrt{x^2 - \{1 + (r/s)(1 - p)/p\}\{x^2 - xx'(1 - p)/s\}}}{p + (r/s)(1 - p)} \tag{2c}$$

$$x_2 = Zm = \frac{x - \sqrt{x^2 - \{1 + (s/r)(p/(1 - p))\}\{x^2 - xx'\, p/r\}}}{(1 - p) + (s/r)p} \tag{2d}$$

We also demonstate that the *penetrance* variance $(\sigma_X^2)$ is restricted such that:

$$0 \leq \sigma_X^2 \leq (x'/2)^2 \tag{2e}$$

which is the same as the maximimum possible variance for any distribution [38] on the closed interval [0, *x'*]–*see Section 3a, Equation 2d in* S1 File.

*Quadratic solutions. Equation 2b* has two solutions–the so-called *Upper Solution* and the *Lower Solution*, depending upon the value of the (±) sign. The *Upper Solution* represents the gradual transition from a distribution, when $(\sigma_X^2 = 0)$, in which everyone has a *penetrance* of (*x'*) to a bimodal distribution, when $\{\sigma_X^2 = (x'/2)^2\}$, in which half of the (*G*) subset has a *penetrance* of (*x'*) and the other half has a *penetrance* of zero. Although, under *some* environmental conditions: $(\forall x_i \in X: x_i > 0)$, as noted previously (*see* Methods #*1A*), there may be certain environmental conditions, in which, for *some* individuals in the (*G*) subset:

$$P(MS \mid G_i, E_T) = 0$$

Therefore, the *Upper Solution*, during any particular *Time Period*, is constrained such that:

$$x'/2 \leq x \leq x'$$

The *Lower Solution* represents the gradual transition from the bimodal distribution described above to increasingly extreme and asymmetric distributions [3]. The *Lower Solution*, however, is further constrained by the requirement of *Equation 2e* that when: $(\sigma_X^2 = 0)$ then: (*x* = *x'*). Therefore, the *Lower Solution* is constrained such that:

$$0 < x \leq x'/2$$

For this analysis, we also assume that either the set (*G*) by itself or, considered separately, the sets (*F, G*) and (*M, G*), conform to the *Upper Solution*. In this circumstance, on theoretical grounds, from Eqs 2b & 2e (*above*), it must be the case that either:

$$1 \leq x'/x \leq 2$$

or both:

$$1 \leq r = x_1'/x_1 \leq 2;$$

and:

$$1 \leq s = x'_2/x_2 \leq 2$$

Using a "substitution" analysis, we wrote a computer program, which incorporated the acceptable ranges for the parameters $\{P(G); P(MS \mid MZ_{MS}); p = P(F \mid G); r; s; P(MS); P(F \mid MS);$ and $s_a\}$, into the *Summary Equations* (*below*) and determined those combinations (i.e., solutions) that fit within the acceptable ranges for both the observed and non-observed parameters (*see* Methods *#1E & #2*).

*Summary equations.* from: *Definitions* (Table 2); *Equations 1–e; above & Section 7a in* S1 File

$$P(MS \mid G) = x; P(MS \mid F, G) = x_1 = Zw; P(MS \mid M, G) = x_2 = Zm$$

$$P(MS \mid IG_{MS}) = x'; P(MS \mid F, IG_{MS}) = x'_1; P(MS \mid M, IG_{MS}) = x'_2$$

$$P(F \mid G) = p; P(F \mid MS) = p'; r = x'_1/x_1; s = x'_2/x_2$$

$$x = (x'/2) \pm \sqrt{(x'/2)^2 - \sigma_X^2}$$

$$x_1 = Zw = \frac{x + \sqrt{x^2 - \{1 + (r/s)(1-p)/p)\}\{x^2 - xx'(1-p)/s\}}}{p + (r/s)(1-p)}$$

$$x_2 = Zm = \frac{x - \sqrt{x^2 - \{1 + (s/r)(p/(1-p))\}\{x^2 - xx'p/r\}}}{(1-p) + (s/r)p}$$

$$x = px_1 + (1-p)x_2$$

$$p' = px_1/x$$

$$p'/(1-p') = (x_1/x_2) * \{p/(1-p)\}$$

$$x' = x + \sigma_X^2/x$$

## 4. Longitudinal model

**A. General considerations.**  The *Longitudinal Model* is developed in detail in *Sections 4a–c; 5a; & 6a–c in* S1 File. Following standard survival analysis methods [39], we define the cumulative survival $\{S(u)\}$ and failure $\{F(u)\}$ functions where: $F(u) = 1 - S(u)$. These functions are defined separately for *men* $\{S_m(u)$ and $F_m(u)\}$ and for *women* $\{S_w(u)$ and $F_w(u)\}$. In addition, we define the hazard-rate functions for developing MS at different exposure-levels $(u)$ in susceptible *men* and *women* {i.e., $h(u)$ and $k(u)$, respectively}. These hazard-rate functions for *women* and *men* may or may not be proportional to each other but, if they are proportional, then: $k(u) = R * h(u)$, where $(R > 0)$ represents the hazard proportionality factor. Furthermore, as defined previously (*see* Methods *#1B*), the term $P(E \mid G, E_T)$ represents the probability of the event that a member of the $(G)$ subset, selected at random, will experience an environmental exposure "*sufficient*" to cause MS, given their unique genotype and given the prevailing

environmental conditions of the time ($E_T$). We define the exposure ($u$) as the odds that the event ($E$) occurs during the *Time Period* ($E_T$) such that:

$$u = P(E \mid G, E_T)/[1 - P\left(E \mid G, E_T\right)]$$

We further define $H(a)$ to be the cumulative hazard function (for *men*) at an exposure-level of ($u = a$) such that:

$$H(a) = \int_0^a h(u)du$$

Similarly, we define $K(a)$ to be the cumulative hazard function (for *women*) at the same exposure-level of ($u = a$) such that:

$$K(a) = \int_0^a k(u)du$$

and, if the hazards are proportional, then:

$$K(a) = \int_0^a R*h(u)du = R*H(a)$$

In *Section 4a in* S1 File), we develop this *Longitudinal Model* and demonstrate there that these cumulative hazard functions are exponentially related to the cumulative survival. Thus:

$$F_m(a) = Zm = P(MS, E \mid M, G) = \boldsymbol{c}*[1 - e^{-H(a)}] \tag{3a}$$

and:

$$F_w(a) = Zw = P(MS, E \mid F, G) = \boldsymbol{d}*[1 - e^{-K(a)}] \tag{3b}$$

where:

$$\boldsymbol{c} = \lim_{a \to \infty}(Zm) = P(MS \mid M, G, E) \leq 1 \tag{3c}$$

and:

$$\boldsymbol{d} = \lim_{a \to \infty}(Zw) = P(MS \mid F, G, E) \leq 1 \tag{3d}$$

Notably, also, because the response curves for both *men* and *women* are exponential, any two points of observation on these curves will determine the entire curve. Designating the fixed (but unknown) exposure level ($a$) at *Time Period #1* as ($a_1$), and at *Time Period #2* as ($a_2$), we can then use the values of $Zw$ and $Zm$ during these two *Time Periods* to determine, and thus to plot, the entire response curve separately for susceptible *women* and susceptible *men*– see Section 4a, *Equations S6a & S6b* and *S7a & S7b in* S1 File.

{*NB: In this circumstance, we are using the cumulative hazard functions, H(a) and K(a), as measures of exposure for susceptible men and women, not as measures of either survival or failure. By contrast, failure, as defined here, is the event that a person develops MS over the course their life-time. As an example, for men, the term: Zm = P(MS, E $\mid$ M, G, E_T) represents the probability that, during the Time Period (E_T), this failure event occurs for a randomly selected man from the (M, G) subset of (Z). Notably, also, the exposures of H($\underline{a}$) and K(a) are being used in preference to the, perhaps, more intuitive measure of exposure (u = a) provided above. Nevertheless, when {P(E $\mid$ G, E_T) = 0}; both exposure measures are zero–i.e., {a = 0} and {H(a) = K(a) =*

0}. *Also, as the value of*: $\{P(E \mid G, E_{T)} \rightarrow 1\}$; *both exposure measures become infinite–i.e.,* $\{a \rightarrow \infty\}$ *and* $\{H(a)$ & $K(a) \rightarrow \infty\}$. *And, finally, all of these exposure measures increase monotonically with increasing* $P(E \mid G, E_T)$. *Therefore, the mapping of the* $(u = a)$ *measure to either the* $H(a)$ *and* $K(a)$ *measures is both one-to-one and onto. Consequently, all these measures of exposure are equivalent and the use of any of these exposure scales is appropriate. Although the relationship of both the* $H(a)$ *and* $K(a)$ *scales to* $P(E \mid G, E_T)$ *is less obvious than it is for the* $(u = a)$ *scale where*: $P(E \mid G, E_T) = a/(a + 1)$, *the* $H(a)$ *and* $K(a)$ *scales, nonetheless, have the advantage that the probability of failure for each is an exponential function of exposure as measured by* $H(a)$ *or* $K(a)$ *and, thus, these scales are more mathematically tractable.}*

*Environmental exposure levels during different time periods.* As developed in *Section 4a–4c* in S1 File, and because both $P(MS)$ and $P(F \mid MS)$ are increasing with time in both *women* and *men* [6, 22–30], we can define the change in the fixed (but unknown) exposure level that has taken place between the two *Time Periods* in *men* and *women* {i.e., $(q_m)$ and $(q_w)$, respectively} such that:

$$H(a_2) - H(a_1) = q_m > 0$$

and:

$$K(a_2) - K(a_1) = q_w > 0$$

Previously, we assigned the value of these arbitrary units as $(q_m = 1)$ and $(q_w = 1)$ in these *Equations* [3], although such an assignation may be inappropriate. Thus, these units (whatever they are) still depend upon the actual (but unknown) level of environmental change that has taken place for *men* and *women* between the two chosen *Time Periods*. From Eqs 3a and 3b (*above*), these exposure levels depend upon the values of ($c$) and ($d$), which can range over the intervals of: $(1 \geq c > Zm_2)$ and $(1 \geq d > Zw_2)$; the exposure level for each gender being at its *minimum* value (i.e., $q_m^{min}$ and $q_w^{min}$) when ($c = 1$) for *men* and when ($d = 1$) for *women–see Section 4b* in S1 File.

However, these *minimum* exposure level changes, $(q_m^{min})$ and $(q_w^{min})$, may not accurately characterize the *actual* (but unknown) level of environmental change, which has taken place for susceptible *men* and for susceptible *women* between the two *Time Periods*. Therefore, we will refer to $(q_m)$ and $(q_w)$ as the *"actual"* exposure-level changes, which may be different from these *minimum* exposure-level changes such that:

$$q_m \geq q_m^{min}$$

and:

$$q_w \geq q_w^{min}$$

*Relationship of failure to true survival.* Unlike true survival (where everyone dies given sufficient time), the probability of developing MS, either for the subset of susceptible *women* {$Zw = P(MS, E \mid F, G)$} or for the subset of susceptible *men* {$Zm = P(MS, E \mid M, G)$}, may not approach 100% as the probability of exposure {$P(E \mid G, E_T)$} approaches unity (*see Section 4a–4f in* S1 File). Moreover, the limiting value for the probability of developing MS in susceptible *men* ($c$) need not be the same as that in susceptible *women* ($d$). Also, even though the values of the ($c$) and ($d$) parameters are unknown, they are, nonetheless, constants for any disease process, which requires environmental factors as an essential component of disease pathogenesis, and they are independent of whether the hazards are proportional. Finally, the threshold environmental exposure (at which MS becomes possible) must occur at: $P(E \mid G, E_T) = 0$; for one (or both) of these two subsets, provided that this exposure level is possible [3]. If the hazards are

proportional, a difference in threshold ($\lambda$) can be defined as the difference between the threshold in susceptible *women* ($\lambda_w$) and the threshold in susceptible *men* ($\lambda_m$)–i.e., ($\lambda = \lambda_w - \lambda_m$). Thus, if the threshold in *women* is greater than the threshold in *men*, ($\lambda$) will be positive and ($\lambda_m = 0$); if the threshold in *men* is greater than the threshold in *women*, ($\lambda$) will be negative and ($\lambda_w = 0$).

Also, in true survival, both the clock and the risk of death begin immediately at time-zero and continue indefinitely into the future, so that the cumulative probability of death always increases with time. By contrast, here, it may be that the prevailing environmental conditions during some *Time Period* ($E_T$) are such that: $P(E \mid G, E_T) = 0$; even for quite an extended period (e.g., centuries or millennia). In addition, unlike the cumulative probability of death, here, exposure can vary in any direction with time depending upon the specific environmental conditions during ($E_T$). Therefore, although the cumulative probability of failure (i.e., developing MS) increases monotonically with increasing exposure, it can increase, decrease, or stay constant with time.

*Relationship of the (F:M) sex ratio to exposure.* Finally, (*see Section 4d in* S1 File) regardless of ($\lambda$), and regardless of any proportionality, during any *Time Period*, the ratio of the failure probability in susceptible *women* to that in susceptible *men* ($Zw/Zm$) can be expressed as:

$$Zm/Zw = \{P(E \mid G, M, E_T)/P(E \mid G, F, E_T)\} * \{c/d\} \tag{4}$$

Consequently, any observed disparity between ($Zw$) and ($Zm$), during any *Time Period*, must be due to a difference between *men* and *women* in the likelihood of their experiencing a "*sufficient*" environmental exposure, to a difference between ($c$) and ($d$), or to a difference in both. Therefore, by assuming that: ($c = d \leq 1$), we are also assuming that any difference in disease expression between susceptible *women* and *men* is due entirely to a difference between susceptible *men* and *women* in the probability of their experiencing a "*sufficient*" environmental exposure, despite the fact that, for every ($i$), the exposures $\{E_i\}$ and $\{E_{iw}\}$ are both *population-wide* and fixed during any *Time Period* ($E_T$). Because these exposures are "*available*" to everyone, therefore, if the level of "*sufficient*" exposure differs between genders, one possibility might be that this is due to a systematic difference in behavior between susceptible *women* and *men*–i.e., to an increased exposure to, or avoidance of, susceptible environments by one or the other gender (perhaps consciously or unconsciously; or perhaps due to differing gender-roles, differing occupations, differing recreational activities, etc.). However, the fact that *most women* behave differently from *men* does not mean that *all women* do so. Notably, if the circumstance of ($\lambda \neq 0$) were explained by a systematic difference in behavior, then the observation of ($\lambda > 0$) suggests that the behavior of *men* leads to a greater exposure than the behavior of *women*. However, any general conclusion regarding such a difference in behavior between susceptible *women* and *men* cannot be rationalized with the observation that, *currently*: ($Zw_2 > Zm_2$).

Another possible explanation for ($\lambda > 0$), which does not pose this difficulty, is that the distributions of the "critical exposure *intensity*" levels (thresholds) differ between *men* and *women* (*see Section 6g in* S1 File). In this case, although the same exposure "*intensity*" may be experienced equally by the two genders, this "*intensity*" might be "*sufficient*" for a disproportionate number of *women* or *men*. This possibility is considered in detail elsewhere (*see Sections 6g & 8a, 8b in* S1 File).

Also, regardless of whether the hazards are proportional, and because proportion of *women* among susceptible individuals ($p$) is a constant (*see* Table 2), therefore, for any solution, the ratio ($Zw/Zm$), during any *Time Period* ($E_T$), will be proportional to the observed (F:M) *sex ratio* during that period (*see Equation 1b; above*).

*The response curves to increasing exposure.* As noted *above*, any two points of observation on these exponential response curves will define the entire curve (e.g., the values of $Zw$ and $Zm$ during *Time Period #1 and Time Period #2*). Moreover, if these two curves can be plotted on the same *x-axis* (i.e., if *men* and *women* are responding to the same environmental events), the hazards will always be proportional, in which case the values of $(R = q_w/q_m)$ and $(\lambda)$ are determined by a combination of the observed values for $(Zw_2)$, $(Zm_2)$, and the $(F{:}M)$ *sex ratio* change and the fixed (but unknown) values of $(c)$, $(d)$, and $(C)$–*see Sections 4b (Equations S6e & S7c), 6a (Equation S11c, S11d) & 7a in* S1 File–*see also Summary Equations (below)*, Moreover, the values of: $(c)$ and $(d)$ are independent of whether <u>some</u> susceptible individuals can <u>only</u> develop MS in response to extreme (and improbable) environmental conditions (*see Section #1B, above*). This is because these values are determined exclusively by the values of: $(Zw)$, $(Zm)$, $P(MS)$ and the $(F{:}M)$ *sex ratio*, at the two observation points. Nevertheless, if <u>all</u> susceptible individuals can develop MS in response to such extreme conditions, then: $(c = d = 1)$.

**B. Non-proportional hazard.** If the hazard functions for MS in *men* and *women* are not proportional (*see Sections 5a & 6g, 6h in* S1 File), it is always possible that the "*actual*" exposure level changes for *men* and *women* are each at their "*minimum*" values–i.e., $(q_m^{min})$ and $(q_w^{min})$–in which case: $(c = d = 1)$–*see* Methods *#4C (below)*. However, in this circumstance, the MS in *women* must be considered to be a separate disease from MS in *men* (*see Section 6g, 6h in* S1 File). Also, we should note that the condition of: $(c = d = 1)$ is required, unless "true" randomness is a component of disease pathogenesis (*see* Discussion *Section*).

Moreover, in this non-proportional circumstance, the various observed and non-observed epidemiological parameter values still limit possible solutions. However, in this case, although $(c)$ and $(d)$ will still be constants, no information can be learned about them or about their relationship to each other from changes in the $(F{:}M)$ *sex ratio* and $P(MS)$ over time. The observed changes in these parameter values over time could all simply be due to the different environmental circumstances of different times and different places. In this case, also, although *men* and *women* will still each have environmental thresholds, the parameter $(\lambda)$–which relates these thresholds to each other–is meaningless, and there is no hazard proportionality factor $(R)$.

Nevertheless, even with non-proportional hazard, the ratio $(Zw/Zm)$, during any *Time Period*, must still be proportional to the observed $(F{:}M)$ *sex ratio* during that *Time Period* (*see Equation 1b*) and, if: $c = d \leq 1$, then any observed disparity between $(Zw)$ and $(Zm)$, must still be due entirely to a difference between *women* and *men* in the likelihood of their experiencing a "*sufficient*" environmental exposure $(E)$ during that *Time Period* (*see* Eq 4; *above*).

**C. Proportional hazard.** By contrast, if the hazards for *women* and *men* are proportional with the proportionality factor $(R)$, the situation is altered. First, because $(R > 0)$, those changes, which take place for the subsets $P(F, MS)$ and $P(M, MS)$ over time, must have the same directionality. Indeed, this circumstance is in accordance with our epidemiological observations where, over the past several decades, the prevalence of MS has been noted to be increasing for both *women* and *men* [6, 22–30]. Second, including a possible difference in threshold between the genders, the proportionate hazard *Model* can be represented by those circumstance for which:

$$\forall H(a) > \lambda : K(a) = R*\{H(a) - \lambda\} > 0 \tag{5a}$$

so that, from the *Sections 4a & 6a in* S1 File, for *men* and *women*, at the *1ˢᵗ* and *2ⁿᵈ Time*

*Periods*, can be re-written as:

$$F_m(a_1) = Zm_1 = P(MS, E \mid M, G)_1 = \boldsymbol{c}*[1 - e^{-H(a_1)}] \tag{5b}$$

$$F_m(a_2) = Zm_2 = P(MS, E \mid M, G)_2 = \boldsymbol{c}*[1 - e^{-\{H(a_1) + q_m\}}] \tag{5c}$$

$$F_w(a_1) = Zw_1 = P(MS, E \mid F, G)_1 = \boldsymbol{d}*[1 - e^{-R*\{H(a_1) - \lambda\}}] \tag{5d}$$

and:

$$F_w(a_2) = Zw_2 = P(MS, E \mid F, G)_2 = \boldsymbol{d}*[1 - e^{-R*\{H(a_1) + q_m - \lambda\}}] \tag{5e}$$

In this circumstance, as demonstrated in *Section 6a in* S1 File, during any *Time Period*, the parameters ($\lambda$) and ($R$) are determined such that:

$$\lambda = \{\ln[1 - Zw/\boldsymbol{d}] - \ln[1 - Zm/\boldsymbol{c}]\}/R + [(R - 1)/R]*H(\boldsymbol{a}) \tag{6a}$$

and:

$$R = q_w/q_m \tag{6b}$$

When: ($R = 1$) these *Equations* simplify to:

$$\lambda = \ln[1 - Zw/\boldsymbol{d}] - \ln[1 - Zm/\boldsymbol{c}] \tag{6c}$$

and:

$$q_w = q_m \tag{6d}$$

For any specific exposure level $\{H(a) > \lambda\}$, the quantities ($Zw$) and ($Zm$) are unknown. However, considering any disease for which a proportionate hazard *Model* is appropriate, the parameters ($\boldsymbol{c}$, $\boldsymbol{d}$, $R$, & $\lambda$) are fixed (but unknown) constants, so that, from *Equation S11a, S11b, Section 6a in* S1 File, the values of ($Zm$) and ($Zw$) are also fixed at any specific exposure level $\{H(a)\}$.

*Defining an "Apparent" proportionality factor.* We can also define a so-called "*apparent*" value of the hazard proportionality factor ($R^{app}$) such that: $R^{app} = (q_w^{min}/(q_m^{min})$. This "*apparent*" value incorporates, potentially, two fundamentally different processes. First, it may capture the increased level of "*sufficient*" exposure experienced by one group compared to the other. Indeed, from Eq 4, this is the only interpretation possible for circumstances in which: ($\boldsymbol{c} = \boldsymbol{d} \leq 1$). Second, however, if we admit the possibility that: ($\boldsymbol{c} < \boldsymbol{d} \leq 1$), then as shown in the *Section 6b in* S1 File some of ($R^{app}$) will be accounted for by the difference of ($\boldsymbol{c}$) from unity. For example, when ($\boldsymbol{d} = 1$), it will be the case that:

$$R^{app} \geq R = q_w^{min} q_m$$

In this manner, if $(q_m > q_m^{min})$, a portion of the "*apparent*" value ($R^{app}$) will be accounted for by a reduction in value of ($\boldsymbol{c}$) from unity, if such a reduction is possible. Moreover, if such a reduction is possible for susceptible *men*, then, clearly, it is also possible that the value of ($\boldsymbol{d}$) is also reduced from unity in susceptible *women*, in which case: ($\boldsymbol{c} \leq \boldsymbol{d} < 1$), where the "*actual*" exposure level in *women* ($q_w$) would be greater than its minimum value ($q_w^{min}$) such that:

$$R = q_w/q_m > q_w^{min}/q_m$$

Consequently, in each of these circumstances, the *"actual"* value of ($R$) may be different from its *"apparent"* value ($R^{app}$). Nevertheless, from Eqs 6a and 6b (*above*), and from the *Sections 4b & 6a, 6b in* S1 File, under <u>all</u> circumstances, for which: ($c = d \leq 1$), then:

$$R = q_w/q_m = q_w^{min}/q_m^{min} = R^{app}$$

Consequently, under those conditions, for which: ($c = d = 1$), this requires that both:

$$(q_m = q_m^{min}) \, \& \, (q_w = q_w^{min})$$

or, equivalently:

$$(q_m/q_m^{min}) = (q_w/q_w^{min}) = 1$$

By contrast, under those conditions, for which: ($c = d < 1$), this requires that both:

$$(q_m > q_m^{min}) \, \& \, (q_w > q_w^{min})$$

or, equivalently:

$$(q_m/q_m^{min}) = (q_w/q_w^{min}) > 1$$

*Implications that the* ($R$) *value has for the values of* ($\lambda$), ($c$) *and* ($d$). As demonstrated in *Section 6c in* S1 File, and based solely on the observations of an increasing $P(MS)$ and an increasing ($F:M$) *sex ratio* with time [6, 22–30]–a circumstance which is true considering the *"current"* Time Period #2 together with any of the reported previous 5-year epochs as *Time Period #1* [6]–we can conclude, based on purely theoretical grounds, that, if the hazards are proportional, then:

$$\forall (R \geq 1) : \lambda > 0$$

and also:

$$\forall (R \leq 1) : c < d \leq 1$$

**D. Strictly proportional hazard: ($\lambda = 0$).** As demonstrated in *Section 6c, 6d in* S1 File, if ($R \geq 1$) and ($\lambda = 0$), the observed ($F:M$) *sex ratio* either decreases or remains constant with increasing exposure (*see* Methods #4 *Equations 11a–11c*), regardless of the parameter values for ($c$) and ($d$)–e.g., Fig 1A & 1B. Consequently, the only "strictly" proportional circumstances, which are possible, are those in which *men* have a greater hazard than *women* and where ($c < d \leq 1$)–e.g., Fig 1D.

{*NB*: *In these and subsequent Figures, all response curves exemplifying the conditions in which* ($c = d \leq 1$), *are depicted for the condition* ($c = d = 1$). *Nevertheless, for all those conditions where* ($c = d < 1$), *the response curves differ from the curves depicted in the Figures only in so far as the y-axis has a different scale. Therefore, the response curves, depicted at:* ($c = d = 1$), *are representative of all curves for which* ($c = d$)–*see note in Section 6b in* S1 File.}

**E. Intermediate proportional hazard: ($\lambda < 0$).** We can also consider another possible *Model*, which is intermediate between the "strictly" proportional and non-proportional hazard *Models* as described *above*. In this intermediate *Model*, the hazards are still held to be proportional although the onset of the response curves are offset from each other by an amount ($\lambda \neq 0$). As noted earlier: $\forall$ ($R \geq 1$): $\lambda > 0$ and, therefore, for those circumstances in which ($\lambda < 0$), the hazard in *men* must be greater than the hazard in *women* (*see Section 6e in* S1 File). Otherwise, the ($F:M$) *sex ratio* will decrease with increasing exposure, which is contrary to the

evidence [6]–e.g., Fig 2A. Moreover, under those conditions, for which ($c = d \leq 1$) & ($R < 1$) & ($\lambda < 0$), the (*F:M*) *sex ratio* will decrease with increasing exposure until the two response curves have intersected (e.g., Fig 2A & 2B), reaching a level below $\{p/(1 - p)\}$. Following this, the (*F:M*) *sex ratio* steadily increases to ultimately reach the level of $\{p/(1 - p)\}$. However, after the response curve in *men* intersects that in *women* (i.e., after this nadir), this circumstance requires that (*Zm > Zw*) throughout the entire remaining response curve until an (*F:M*) *sex ratio* of: $\{p/(1 - p)\}$ is reached (e.g., Fig 2A & 2B). Thus, the only circumstance in which the Model of ($c = d \leq 1$) is possible is one in which ($p$) is at least as large as the "*current*" value of ($p'$)–i.e., ($p'_2 \geq 0.74$)–*see* Fig 2B; *see also Section 2c in* S1 File. Each of these possibilities is contrary to evidence where *currently* ($Zw_2 > Zm_2$) and, thus, where: $\{(F\!:\!M)$ *sex ratio* $> p/(1 - p)\}$–*see Equation 1b* (*above*). Thus, the condition of: ($\lambda < 0$) is only possible, in circumstances where: ($c < d$)–e.g., Fig 2D.

**F. Intermediate proportional hazard: ($\lambda > 0$).** *Exposure "Intensity*. In considering the notion of exposure "*intensity*", three conclusions that seem well established. First, for every proportional hazard solution that we identified (*see Results Section*), we found that: ($R^{app} > 1$). Moreover, as demonstrated on theoretical grounds in *Section 6b, 6c in* S1 File), and as depicted in Figs 4 and 5 & *S1 Fig in* S1 File, in these circumstances, it must be that:

$$\forall(R \leq 1) : c < d$$

Second, as demonstrated in *Section 6c in* S1 File, under those circumstances, in which both $P(MS)$ and $P(F|MS)$ are increasing with time [6], then:

$$\forall(R \geq 1) : \lambda > 0$$

Third, from the Canadian data [6], it seems inescapable that, as the probability of a "*sufficient*" exposure for susceptible individuals has increased over the past several decades, the probability of developing MS for susceptible *women* has increased at a faster rate than it has for susceptible *men*. Consequently, if the hazards in *men* and *women* are proportional, this faster rate of increase in susceptible *women* implies that one of the following two conditions must hold. Thus, either:

1) $R \leq 1$ in which case: $c < d$

or: 2) $R > 1$ in which case: $\lambda > 0$

Clearly, the first of these conditions excludes the possibility that: $c = d = 1$

In considering the second condition, it should be noted that both of our measures of exposure–i.e., ($a$) and $H(a)$–relate directly back to the parameter $P(E|G)$, which represents the probability of the event that a randomly selected susceptible individual (either a *man* or a *women*) experiences an environmental exposure "*sufficient*" to cause MS in them. Therefore, this second condition–i.e. that: $\lambda > 0$ –indicates that, as the probability of a "*sufficient*" exposure decreases, there comes a point {i.e., $H(a) = \lambda$} where only susceptible *men* can develop MS. This implies that, at (or below) this point: ($R \approx 0$).). Consequently, the requirement that ($R > 1$) creates a paradox in that, for the second condition to be true, susceptible *women* must be more likely than *men* to experience a "*sufficient*" exposure when the probability $\{P(E|G)\}$ is high and, yet, susceptible *men* must be much more likely than *women* to experience a "*sufficient*" exposure when this probability is low.

There are two obvious ways to avoid this potential paradox. The first is to conclude that the hazards are not proportional. Nevertheless, despite this possibility, such a conclusion also presents problems of its own (*see* Discussion *Section*). For example, because *women* and *men* of the

same "*i-type*" necessarily have proportional hazards (*see Section 6h in* S1 File), in this case, we would also have to conclude that susceptible *women* and *men* can never be in the same "*i-type*" group and, thus, that each gender requires distinct sets of environmental conditions to develop MS. Also, we would have to further conclude that MS in *women* must represent a disease distinct from MS in *men*. Alternatively, if susceptible *women* and *men* could both be members of certain "*i-type*" groups but not others, we would have to conclude MS represents three distinct diseases (one in *women*, one in *men*, and a third in both). Any such conclusion seems to be at substantial variance with both the genetic and the epidemiological evidence (*see* Discussion *Section*).

The second way to avoid the paradox, is to conclude that the first of the two possible proportional hazard conditions is true–i.e., that both ($R \leq 1$) and: (*c* < *d*). Notably, the condition of: ($R \leq 1$) is compatible with any value of ($\lambda$). However, if ($\lambda > 0$), the simultaneous condition of: ($R \leq 1$), offers, at least, a consistent interpretation of the existing data because, under these conditions, at every population exposure level (*a*), the probability of the event that a randomly selected susceptible *man* will experience a "*sufficient*" environmental exposure to cause MS in them is as great, or greater, than the same probability for a susceptible *woman* (*see S1 Fig in* S1 File, Figs 4 & 5). Thus, the notion of a "critical exposure *intensity*" (*discussed below*), although it may be necessary to rationalize a threshold difference, it is not necessary to resolve a paradox. Nevertheless, accepting this conclusion, does require also accepting the fact that some susceptible *men* will never develop MS, even when the correct genetic background occurs together with an environmental exposure "*sufficient*" to cause MS in that individual.

Nevertheless, despite the paradox created by the possibility that: ($\lambda > 0$) & ($R > 1$), there are potential rationales for resolving it. For example, one way to explain it might be if *some* susceptible *men* and *women* had "*purely genetic*" MS [3]–i.e., that these individuals can develop MS under *any* environmental circumstance. If the proportion of "*purely genetic*" MS were equal in *women* and *men*, such individuals could, effectively, raise the *x-axis* for each of the response curves {i.e., increase the level of (0) on the *y-axis*} to the point where the response curves for *men* and *women* intersect, in which case, there would be no lag when considering "*environmental*" MS, alone [3]–*see* Fig 4. However, after this intersection (i.e., after the onset of these "effective" response curves), the conditions would be identical to those described for ($\lambda = 0$)–*see Section 6c, 6d in* S1 File–in which case, when (*c* = *d* = 1), the *F:M sex ratio* will decline with increasing exposure–a possibility that is counter factual [6, 22–30]. Naturally, the proportion of "*purely genetic*" MS might not be equal in *women* and *men* but, in such a circumstance, the paradox would remain unresolved [3]. Consequently, a "*purely genetic*" rationale is not possible when: (*c* = *d* = 1).

However, other potential rationales can also be envisioned (*see Section 6g, in* S1 File). To do this, we introduce the notion of a "critical exposure *intensity*" level, as the exposure level, at (or above) which, the exposure becomes "*sufficient*" for each susceptible individual (*see* Fig 5). If this notion is appropriate, it could help to rationalize the paradox of having both: ($\lambda > 0$) & ($R > 1$)–*see Sections 6g & 8a, 8b in* S1 File. However, to accommodate the condition of: (*c* = *d* = 1), the required circumstances are extreme–e.g., *S2, S3 Figs in* S1 File–and don't match well with the response curves presented in Fig 3. By contrast, in all circumstances, those conditions for which (*c* < *d*) are much simpler, don't require any extreme circumstances, fit with any value of (*R*), and match well with the response curves depicted in Fig 4 –e.g., *S1–S3 Figs in* S1 File.

*Exposure "Intensity" in susceptible women.* Any condition for which ($\lambda > 0$) indicates that there must be some environmental conditions in which only susceptible *men* can experience a "*sufficient*" exposure. This circumstance requires that the threshold difference for, at least, some "*i-types*" ($\lambda_i$), is such that: ($\lambda_i > 0$). We will define the family of exposures $\{E_{iw}\}$ to be the

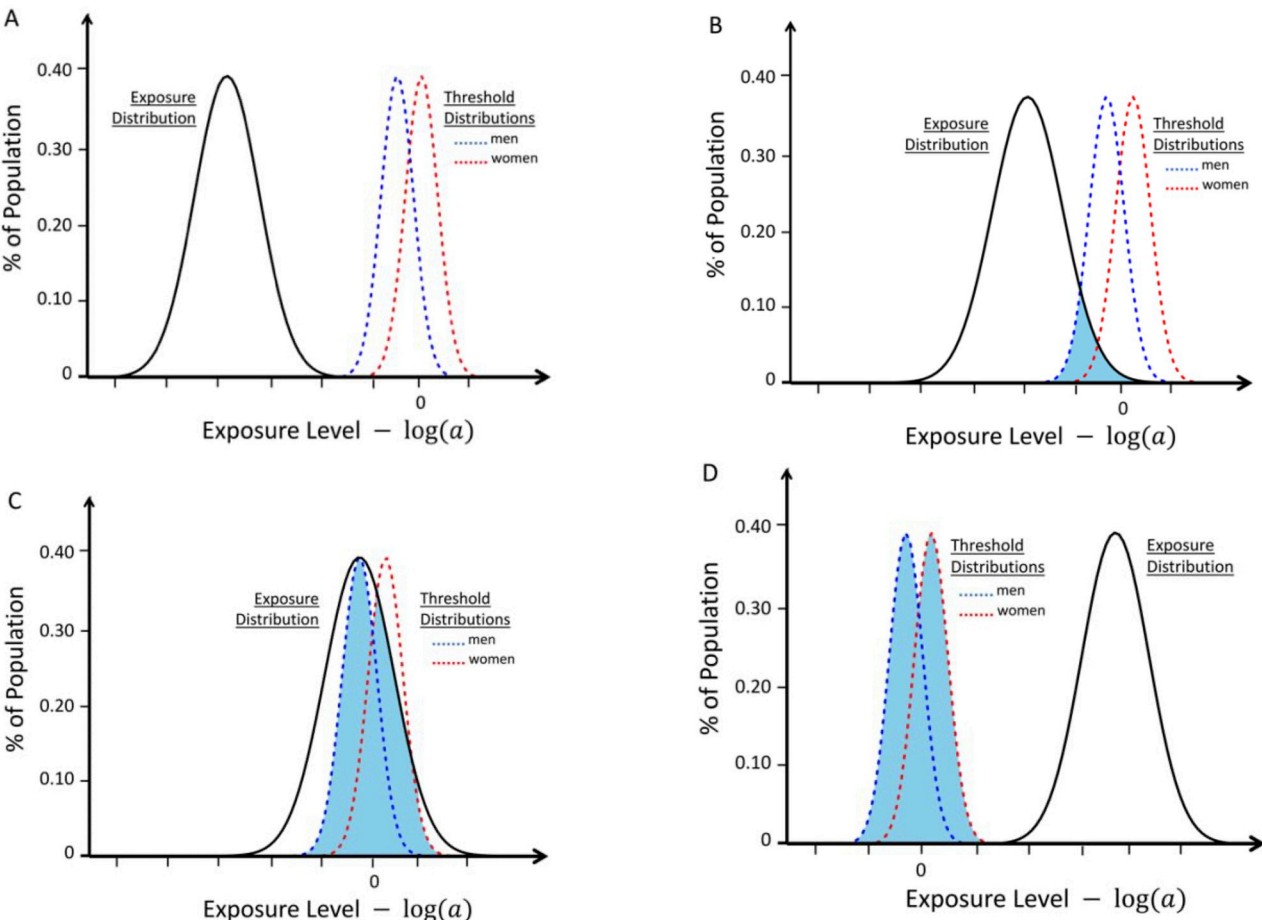

**Fig 5. Hypothetical relationship between exposure "*intensity*" and disease expression (*see Sections 6g & 8a, 8b in* S1 File).** Plotted on the *x-axis* is the level (or "*intensity*") of exposure in units of the *log-transformed* exposure–*log(a)*. Plotted on the *y-axis* is the proportion of the susceptible population (*G*) who experience an exposure "*sufficient*" to cause MS in them. The solid black lines represent the distribution of "*actual*" level of exposure experienced by the susceptible population. The dotted lines (red for *women* and blue for *men*) represent the distributions of these "critical exposure intensity" (or "threshold") levels for susceptible *men* and *women*. These "threshold" levels for each individual are defined as that exposure level, at (or above) which, the exposure becomes "*sufficient*" to cause MS in that person. These threshold distributions have been plotted, arbitrarily, for conditions of (*p* = 0.5). Because (*a*) is the odds of exposure, the distribution of these "threshold" levels are expressed in units *log(a)*, because this transformation will generally normalize the variance [39]–*see also Section 8a, 8b in* S1 File. In these *Figures*, the exposure level of: {*log(a)* = 0}, has been chosen as the point where the average odds of a "critical exposure *intensity*" level is equal to (1). No other units are provided because these are undefined other than as they relate to the variance of these "threshold" distributions in susceptible *men* and *women* ($\sigma_w^2$ and $\sigma_m^2$), respectively. The circumstances depicted are those, in which *men* and *women* have the same variance but *men* have a lower mean compared to *women* (i.e., $\mu_w - \mu_m = 2 * \sigma_w^2$). In any case, however, because ($\lambda > 0$), *men* must disproportionately (or exclusively) experience a "*sufficient*" exposure at low exposure "*intensities*". In these examples, the blue shading represents those individuals who receive a "*sufficient*" exposure as the level of population exposure increases progressively–i.e., *Fig 5A* depicts the circumstance, in which the population exposure is such that no one experiences a "*sufficient*" exposure; *Fig 5B and 5C* depict circumstances, in which some (but not all) individuals experience a "*sufficient*" exposure; and *Fig 5D* depicts the circumstance where the population exposure has increased to the point where it exceeds the "critical exposure *intensity*" level for everyone.

subset of exposures, within the {$E_i$} family, that are "*sufficient*" for susceptible "*i-type*" *women* such that:

$$\forall (iw = 1, 2, \ldots, m_{it}) : \{E_{iw}\} \subset \{E_i\}$$

where, for at least one (*i*), it must be the case that:

$$P(\{E_{iw}\}) < P(\{E_i\})$$

In turn, as for our earlier definition of (*E*)–*see* Methods *#1B* –we define the event (*E<sub>w</sub>*) to represent the union of the (*m<sub>it</sub>*) disjoint events, which exhibit the pairing of susceptible "*i-type*" *women* with "*sufficient*" environments, where:

$$(E_w) = (\{E_{1w}\}, G_{1t}, F) \cup (\{E_{2w}\}, G_{2t}, F) \ldots \cup (\{E_{m_{it}w}\}, G_{m_{it}t}, F)$$

and:

$$P\left(E \,\middle|\, G, E_T\right) > P(E_w \,\middle|\, G, E_T)$$

*Exposure variability* (*i.e., for* $R_i$ & $\lambda_i$) *in "i-type" individuals.* If both *men* and *women* are (or potentially could be) members of any specific "*i-type*" group, by definition, these *men* and *women* each have a non-zero probability of developing MS in response to every one of the ($v_i$) "*sufficient*" sets of exposures within the$\{E_i\}$ family for this group. In these circumstances, these specific *i-types*, considered separately, can be plotted on the same *x-axis* and, thus, will necessarily exhibit proportional hazards for the two genders–*see Sections 4f & 6h in* S1 File. Moreover, as demonstrated in *Section 6h in* S1 File, the condition of proportionality for the entire susceptible population will still be present, regardless of whether different *i-types* have different proportionality constants, and regardless of whether susceptible individuals of different *i-types* have different threshold differences between *men* and *women*.

*Contrasting the possibilities that*: (*c* = *d*) *or* (*c* < *d*). When ($\lambda > 0$), to account for an increase in the (*F:M*) *sex ratio*, as shown in Fig 3, although it is possible for: (*c* = *d* ≤ 1), this circumstance, nevertheless, seems unlikely. First, to achieve an *F:M sex ratio*, which reaches its *current* level (*i.e., p* ≈ 0.76), requires either that both the value of ($\lambda$) is small and the value of (*R*) is large (e.g. Fig 3B–3D), or that the value of (*p*) is large (e.g. Fig 3A). And second, the ascending portion of the response curve is very steep, which indicates that any change in the (*F:M*) *sex ratio* is quite large in response to small changes in exposure. Thus, the window of possible changes in environmental exposure necessary to explain the Canadian data is quite narrow [6]. Also, if notions of a "critical exposure *intensity*" (*see above; see also Section 6g in* S1 File) are correct, then neither of these conditions fit well with a transition from a *male predominant* MS to a *female predominant* MS, which takes place relatively late in the response curve, even under extreme conditions–*see Section 8a, 8b in* S1 File. Moreover, following this narrow window, and for most of these response curves, the (*F:M*) *sex ratio* is declining–a circumstance, which is contrary to evidence [6, 22–30]. Also, finally, the increase in failure rate (i.e., the increase in the *penetrance* of MS for the population), which was observed in Canada between the two *Time Periods*, was large (>32%) and especially prominent among *women* (*see Section 7a in* S1 File). Thus, although compatible with the condition of: (*c* = *d* = 1), each of the required circumstances seem to be at odds with the Canadian data, which demonstrates that the (*F:M*) *sex ratio* has been steadily, and gradually, increasing over many decades [6] and where, *currently*, the proportion of *women* among MS patients is quite high. By contrast, the circumstances of Figs 1D, 2D & 4A–4D *& S1–S3 Figs in* S1 File–i.e., where *c* < *d* ≤ 1)–result in a continuously increasing (*F:M*) *sex ratio* with increasing exposure over most (or all) of the response curves, they easily account for the magnitudes of the observed (*F:M*) *sex ratios*, they don't invoke extreme circumstances, and, as in Figs 1D & 4A–4D, they could also account for the observation that, at an earlier *Time Point* in the history of MS [40], in both Europe and the United States, the proportion of *men* among individuals with MS seemed to substantially

exceed that of *women* such that:

$$0.58 = P(M \mid MS, E_T) > P(F \mid MS, E_T) = 0.42$$

In conclusion, therefore, as indicated in *Methods #4C*, the condition of: ($c < d \leq 1$) is necessarily true for all circumstances, in which ($R \leq 1$) and also, as discussed *above*, seems likely to be true for those circumstances, in which: ($R > 1$).

*Summary equations*. For each of the two proportional hazard *Models*, we can use both the observed parameter values, the change in the ($F$:$M$) *sex-ratio*, and the change in $P(MS)$ for Canada between any two *Time Periods* [6], and, thereby, construct each of these response curves in their entirety [3]. The values for: $Zw_2$, $Zm_2$, $Zw_1$, $Zm_1$, I, ($d$), $P(E \mid G, F)$, $P(E \mid G, M)$, ($C$), and ($\lambda$) can then be determined [3] as:

$$Zm_2 = P(MS, E \mid G, M)_2 = P(MS \mid G, M)_2 = P(M, MS)_2 / P(G, M)$$

$$Zw_2 = P(MS, E \mid G, F)_2 = P(MS \mid G, F)_2 = P(F, MS)_2 / P(G, F)$$

$$Zm_1 = P(MS, E \mid G, M)_1 = \{P(M, MS)_1 / P(M, MS)_2\} * Zm_2$$

$$Zw_1 = P(MS, E \mid G, F)_1 = \{P(F, MS)_1 / P(F, MS)_2\} * Zw_2$$

$$c = (Zm_2) * \{e^{q_m} - [P(M, MS)_1 / P(M, MS)_2]\} / (e^{q_m} - 1)$$

$$d = (Zw_2) * \{e^{q_w} - [P(F, MS)_1 / P(F, MS)_2]\} / (e^{q_w} - 1)$$

$$P(E \mid M, G)_2 = Zm_2 / c$$

$$P(E \mid F, G)_2 = Zw_2 / d$$

$$H(a_2) = \ln[1 - Zm_2 / c]$$

$$K(a_2) = R * (H(a_2) - \lambda) = \ln[1 - Zw_2 / d]$$

$$C < P(M \mid MS)_2 / P(M \mid MS)_1$$

and:

$$\lambda = \{\ln[1 - Zw_2 / d] - \ln[1 - Zm_2 / c]\} / R + [(R - 1)/R] * H(a_2)$$

For the non-proportional *Model*, those parameters, which include ($c$) or ($d$), cannot be estimated from the observed changes in $P(MS \mid F)$ and $P(MS \mid M)$ over time. Notably, the values for $P(F \mid MS)_1$, $P(F \mid MS)_2$, and $P(MS)_2$ have been directly or indirectly observed [3, 6]. Also, the values of ($Zw_1$), ($Zw_2$), ($Zm_1$), ($Zm_2$), $P(E \mid G, F)_2$, $P(E \mid G, M)_2$, ($c$) and ($d$) are, not surprisingly, only related to the circumstances of either *men* and *women*, considered separately. Using a "substitution" analysis, we wrote a computer program, which incorporated the acceptable parameter ranges (*see* Methods #2; *above*) for the parameters {$P(G)$; $P(MS \mid MZ_{MS})_2$; $p = P(F \mid G)$; $P(MS)_2$; $P(MS \mid F, G)_2$; $P(F \mid MS)_1$; $r$; $s$; $s_a$ and $C$}, into the governing equations (*above*) and determined those combinations (i.e., solutions) that fit within the acceptable ranges for

both the observed and non-observed parameters (*see* Methods *#2*). For this analysis, unlike for our *Cross-sectional Model*, we loosened the constraints on the values of (*r*) and (*s*) such that:

$$1 \leq r = x_1'/x_1 \leq 30; \text{ and} : 1 \leq s = x_2'/x_2 \leq 30$$

## Results

### 1. Cross-sectional model

Assuming that the subset (*G*) conforms to the <u>*Upper Solution*</u> of the *Cross-sectional Model*, and using *Assertion C* (a*bove*) the range of values for the parameters $P(G)$ and $P(MS|G)_2$ were:

$$0.003 \leq P(G) < 0.55$$

$$0.01 < P\left(MS \middle| G\right)_2 \leq 0.33$$

If we consider the more restricted range of ($s_a < 1.9$) for the impact of sharing the ($E_{twn}$) environment with an *MZ*-twin, then:

$$0.003 \leq P(G) < 0.3$$

$$0.05 < P\left(MS \middle| G\right)_2 \leq 0.3$$

Assuming that the subset (*G*) does not conform to the <u>*Upper Solution*</u>, but that, considered separately, each of the subsets (*F*, *G*) and (*M*, *G*) do, then:

$$0.005 < P(G) \leq 0.82$$

$$0.004 \leq P(G|F) \leq 0.66$$

$$0.008 \leq P(G|M) \leq 0.99$$

If we again consider the more restricted range of ($s_a < 1.9$) for the impact of sharing the ($E_{twn}$) environment with an *MZ*-twin, then:

$$0.005 < P(G) \leq 0.55$$

$$0.006 \leq P(G|F) \leq 0.14$$

$$0.008 \leq P(G|M) \leq 0.99$$

We previously concluded that it was possible (or even probable) that men might be disproportionately represented in the subset (*G*), although any marked disparity in this regard seemed implausible [3]. Therefore, if the restriction of: $\{P(M|G) \leq 0.75\}$ is included with the restrictions such that: ($s_a < 1.9$), ($r \leq 2$) & ($s \leq 2$), and also using the "*current*" *sex-ratio* data

from Canada [6] such that: $P(F\,|\,MS)_2 = 0.74 - 0.78$; (*see* Methods *#2*), then these estimates are:

$$0.005 < P(G) < 0.3$$

$$0.004 \leq P(G\,|\,F) \leq 0.14$$

$$0.01 \leq P(G\,|\,M) < 0.28$$

$$0.004 \leq x = P\left(MS\,|\,G\right)_2 \leq 0.12$$

$$0.02 \leq x' = P\left(MS\,|\,IG_{MS}\right)_2 < 0.34$$

$$0.07 \leq x_1 = P\left(MS\,|\,F,G\right)_2 \leq 0.31$$

$$0.001 \leq x_2 = P\left(MS\,|\,M,G\right)_2 \leq 0.08$$

$$0.14 \leq x'_1 = P\left(MS\,|\,F,IG_{MS}\right)_2 < 0.43$$

$$0.01 \leq x'_2 = P\left(MS\,|\,M,IG_{MS}\right)_2 \leq 0.14$$

$$0.025 < P(F\,|\,G) < 0.66$$

$$1.9 \leq P\left(MS\,|\,F,G\right)/P(MS\,|\,M,G) = x_1/x_2 \leq 8.3$$

$$2.0 \leq P(MS\,|\,F,IG_{MS})/P(MS\,|\,M,IG_{MS}) = x'_1/x'_2 \leq 16$$

## 2. Longitudinal model

Using the *Longitudinal Model*, assuming non-proportional hazards, the possible ranges for these various parameters were:

$$0.001 < P(G) \leq 0.52$$

$$0.001 < P(G|F) < 0.32$$

$$0.001 < P(G|M) \leq 0.94$$

$$0.10 \leq P(F|G) \leq 0.71$$

$$0.004 \leq x = P\left(MS|G\right)_2 \leq 0.20$$

$$0.02 \leq x' = P\left(MS|G, IG_{MS}\right)_2 \leq 0.34$$

$$0.001 \leq P\left(MS|G\right)_1 \leq 0.15$$

$$0.03 \leq x_1 = P\left(MS|F, G\right)_2 \leq 0.44$$

$$0.001 < x_2 = P\left(MS|M, G\right)_2 \leq 0.125$$

$$0.03 \leq x_1' = P\left(MS|F, G, IG_{MS}\right)_2 \leq 0.44$$

$$0.002 \leq x_2' = P\left(MS|M, G, IG_{MS}\right)_2 \leq 0.175$$

$$1.2 \leq x_1/x_2 = P\left(MS|F, G\right)/P(MS|M, G) \leq 32$$

$$2.0 \leq x_1'/x_2' = P(MS|F, IG_{MS})/P(MS|M, IG_{MS}) < 35$$

In addition, we found that the solution space for both ($r$) & ($s$) was restricted: ($r < 20$) and: ($s < 30$). Restricting the ranges such that: ($s_a < 1.9$), ($r \leq 2$) & ($s \leq 2$) changes the above

estimations such that:

$$0.001 < P(G) < 0.30$$

$$0.001 < P(G|F) \leq 0.09$$

$$0.004 < P(G|M) \leq 0.54$$

$$0.10 \leq P(F|G) \leq 0.65$$

$$0.02 \leq x = P\big(MS|G\big)_2 \leq 0.20$$

$$0.10 \leq x' = P\big(MS|G, IG_{MS}\big)_2 \leq 0.34$$

$$0.005 \leq P\big(MS|G\big)_1 \leq 0.15$$

$$0.10 \leq x_1 = P\big(MS|F, G\big)_2 \leq 0.30$$

$$0.006 < x_2 = P\big(MS|M, G\big)_2 \leq 0.125$$

$$0.14 \leq x'_1 = P\big(MS|F, G, IG_{MS}\big)_2 \leq 0.42$$

$$0.008 \leq x'_2 = P\big(MS|M, G, IG_{MS}\big)_2 \leq 0.175$$

$$1.7 \leq x_1/x_2 = P\big(MS|F, G\big)/P\big(MS|M, G\big) \leq 27$$

$$2.0 \leq x'_1/x'_2 = P(MS|F, IG_{MS})/P(MS|M, IG_{MS}) < 27$$

Using the *Longitudinal Model*, assuming ($c = d = 1$) and, thus, with ($R = R^{app} = q_w^{min}/q^{min}$), the possible ranges for these parameters are unchanged from the unrestricted non-

proportional *Model* except for the additional estimations of:

$$0.0005 \leq \lambda \leq 0.13$$

$$1.3 \leq R^{app} \leq 1177$$

$$0.03 \leq P\left(E \mid F, G\right) \leq 0.44$$

$$0.001 \leq P\left(E \mid M, G\right) \leq 0.125$$

$$1.2 \leq P\left(E \mid F, G\right)/P\left(E \mid M, G\right) \leq 32$$

If the solution space were restricted: such that: ($s_a < 1.9$), ($r \leq 2$) & ($s \leq 2$), the above estimates are unchanged except that:

$$1.9 \leq R^{app} \leq 516$$

$$0.10 \leq P\left(E \mid F, G\right) \leq 0.30$$

$$0.006 \leq P\left(E \mid M, G\right) \leq 0.125$$

$$1.9 \leq P\left(E \mid F, G\right)/P\left(E \mid M, G\right) \leq 27$$

Considering the circumstances where ($R = 1$) and ($\boldsymbol{d} = 1$), these estimates are the unchanged from the non-restricted values above except:

$$0.002 < \lambda < 2.4$$

$$0.002 \leq \boldsymbol{c} \leq 0.786$$

$$1.3 < \boldsymbol{d}/\boldsymbol{c} < 493$$

$$0.03 \leq P\left(E \mid F, G\right) \leq 0.3$$

$$0.03 \leq P\left(E \mid M, G\right) < 0.94$$

$$0.04 \leq P\left(E \mid F, G\right)/P\left(E \mid M, G\right) \leq 0.95$$

As in our analysis of the *Cross-sectional Model* (above), if the restriction of: $\{P(M \mid G) \leq 0.75\}$ is included with the above restrictions such that: ($s_a < 1.9$), ($r \leq 2$) & ($s \leq 2$) then, for

$(R = 1)$, these estimates become:

$$0.001 < P(G) < 0.10$$

$$0.001 < P(G|F) \leq 0.09$$

$$0.001 < PG|M) \leq 0.08$$

$$0.08 \leq x = P(MS|G)_2 \leq 0.20$$

$$0.10 \leq x' = P(MS|G, IG_{MS})_2 \leq 0.34$$

$$0.02 \leq P(MS|G)_1 \leq 0.15$$

$$0.10 \leq x_1 = P(MS|F, G)_2 \leq 0.30$$

$$0.03 < x_2 = P(MS|M, G)_2 \leq 0.125$$

$$0.14 \leq x_1' = P(MS|F, G, IG_{MS})_2 \leq 0.42$$

$$0.03 \leq x_2' = P(MS|M, G, IG_{MS})_2 \leq 0.175$$

$$1.7 \leq x_1/x_2 = P(MS|F, G)/P(MS|M, G) \leq 4.5$$

$$2.0 \leq x_1'/x_2' = P(MS|F, IG_{MS})/P(MS|M, IG_{MS}) < 8.6$$

$$0.007 < \lambda < 2.4$$

$$0.04 \leq c \leq 0.55$$

$$1.8 < d/c < 26$$

$$0.1 \leq P(E|F, G) \leq 0.3$$

$$0.11 \leq P(E|M, G) \leq 0.93$$

$$0.18 \leq P(E|F, G)/P(E|M, G) \leq 0.95$$

## Discussion

The present analysis provides considerable insight to the nature of susceptibility to MS. To begin, there are two statements, which are necessarily true regarding the role of genetics and the environment in MS pathogenesis. First, if we include "any genotype" as one of the possible "*susceptible*" genotypes, then every person who develops MS must have a "*susceptible*" genotype. Second, if we include "any environmental experience" as a possible "*sufficient*" environment, then every person who develops MS must have experienced a *sufficient*" environment.

Because these general statements *must* be true, we have defined "*susceptible*" genotypes and "*sufficient*" environments very broadly to encompass any possibility. Thus, we define the subset of "*susceptible*" individuals (*G*) to consist of every person (genotype) in the population who has *any* non-zero chance of developing MS under *some* environmental conditions (Methods #1A). Similarly, we define a "*sufficient*" environment as *any* set of environmental conditions that are "*sufficient*" to cause MS in *some* member of the (*G*) subset (Methods #1B). Notably, this definition includes *every* environmental condition or experience (known, suspected, or unknown), which is required (i.e., necessary) for such "*sufficiency*".

Moreover, we define the probability {*P(G)*} as the probability of the event that a randomly selected member of general population (*Z*) is also a member of the (*G*) subset. Also, as the likelihood of a "*sufficient*" exposure for the entire (*G*) subset increases to unity, we define the constants (***c***) and (***d***) to represent the limiting probability of developing MS for *male* and *female* members of the (*G*) subset, respectively. In this case, the two principal conclusions, which can be drawn from our analysis, can be stated as:

1. $P(G) < 1$

and: 2. $c < d \leq 1$

The first of these conclusions seems inescapable, based both on the data from Canada [6, 7] and on the data about *MZ*-twin concordance rates, reported from other locations around the world [3, 7, 55–62]. Thus, both of our *Models*, and the intersection of all our analyses, substantially support each other. For example, regardless of the whether the *Cross-sectional* or the *Longitudinal Model* was used, regardless of the whether the hazards are proportional and, if proportional, regardless of the proportional *Model* assumed, the consistently supported range for *P(G)* is:

$$0.003 < P(G) \leq 0.52$$

Thus, under any circumstance, a large percentage of the general population ($\geq$ 48%), and likely the majority, must be impervious to getting MS, regardless of their environmental experiences. Consequently, if a person doesn't have the appropriate genotype, they can't get the disease. This conclusion is particularly evident for *women*, where:

$$0.001 < P\left(G \middle| F\right) < 0.32$$

Thus, much of the population and most *women* lack this essential component of MS pathogenesis. Notably, *any* conclusion that: {*P(G)* < 1} excludes the possibility that MS can ever occur in persons who lack a genetic predisposition for the disease. In this sense, fundamentally, MS must be a genetic disorder although its genetic basis is quite complex (*see below*).

Nevertheless, fundamentally, MS is also an environmental disease. Thus, over the last several decades, both the prevalence (and, thus, the *penetrance*) of MS and the *F:M sex ratio* have increased in many parts of the world [6, 22–30]. Because genetic factors do not change this quickly, these facts implicate an environmental factor (or factors) as also critical to disease

pathogenesis [3, 9]. Moreover, this conclusion is also indicated by the fact that ($E_{twn}$) environment significantly impacts the likelihood that an individual either has, or will subsequently develop, MS [7, 31–37]. And finally, this is supported by the fact that the recurrence risk for *MZ*-twins (with identical genomes), as discussed *below*, is generally reported to be less than ~30%–an observation, which indicates that genetics plays only a minor role in determining who does, and who does not, develop disease.

Our second principal conclusion (*above*) relates to these environmental events and, if correct, this conclusion indicates that "true" randomness plays a role in disease pathogenesis. However, the evidence for this conclusion is not as compelling as it is for our first. Thus, there are potential scenarios that can be envisioned (and require consideration), under which the condition of ($c = d = 1$) might be possible and, thus, in which "true" randomness might not play a role in disease development. Principal among these scenarios is the possibility that the hazard functions for developing MS in the two genders are not proportional (*see Section 5a in* S1 File). In this view, each gender develops disease in response to different sets of environmental conditions and, thus, MS in *women* and MS in *men* represent two or three fundamentally different diseases–*see Section 6g, 6h in* S1 File. Moreover, in this non-proportional view, the environmental changes, which have taken place in Canada between the two *Time Periods* of 1941–1945 & 1975–1980 (whatever these are), would be interpreted as involving those events that impact MS development in susceptible *women* to a considerably greater extent than they do those events that impact MS development in susceptible *men*. However, even in this case, the limits derived for the parameters $\{(PG), P\left(G \mid F\right), C, x, x', x_1, x'_1, x_2, x'_2, x_1/x_2, \text{ and } x'_1/x'_2\}$ would still apply.

Currently, many (most) authorities believe that *men* and *women* with MS have the same disease and, therefore, would likely find the notion that MS in *men* and MS in *women* represent fundamentally different diseases, involving different environmental events, to be implausible. Nevertheless, because this possibility is the most obvious and most compelling counterargument to our second conclusion (*above*), it is important to consider the epidemiological evidence against this notion in some detail. Also, in this regard, it is important to appreciate two general features regarding "*i-type*" groups as defined here (Methods *#1A*). First, if *men* and *women* are (or potentially could be) members of the any particular "*i-type*" group, the hazards must be proportional within that group and, second, if both *men* and *women* are (or potentially could be) members of every "*i-type*" group, the hazards must be proportional within the population (*see Section 6f & 6h in* S1 File). Moreover, the "proportional hazard" view does not depend upon every "*i-type*" group having either the same proportionality constant or the same the threshold difference between susceptible *women* and susceptible *men* (*see Section 6g, 6h in* S1 File). Rather, it depends only upon the same environmental events having a non-zero probability of impacting the development of MS in both susceptible *women* and *men* (*see Sections 4f & 6g, 6h in* S1 File).

It is noteworthy, therefore, that both genders seem to share very similar mechanisms of disease pathogenesis. Indeed, there have been several epidemiological observations that link MS, unequivocally, to environmental factors, to genetic factors, or to both and when these have been explored systematically, these factors seem to impact both *men* and *women* in a similar manner. For example, a *month-of-birth* effect has been reported in MS whereby, in the northern hemisphere, the risk of subsequently developing MS is greatest for babies born in May and least for babies born in November compared to other months during the year [41]. This *month-of-birth* effect was predicted to be inverted in the southern hemisphere [41] and, in fact, a subsequent *population-based* study from Australia found the peak risk to be for babies born in November-December and the nadir to be for babies born in May-June [42]. Although this

*month-of-birth* effect is somewhat controversial [43], it has been widely (and reproducibly) reported by many authors and the effect is apparent in both *men* and *women* [41, 42, 44, 45]. Thus, MS-risk seems to cycle throughout the year and this observation, if correct, clearly, implicates an environmental factor (or factors)–affecting both *men* and *women* alike–which is (are) linked to the solar cycle and occur(s) during the intrauterine or early post-natal period [9]. Second, the recurrence risk of MS is generally found to be greater in a co-twin of a *DZ*-twin proband with MS compared to a non-twin co-sibling of a sibling proband with MS [7, 31–37]. This effect also implicates an environmental factor (or factors) that occur(s) in proximity to the birth and this effect is apparent in both *men* and *women* [3, 9]. Third, it is widely reported that MS becomes increasingly prevalent in those geographic regions, which lie farther (either north or south) from the equator [9, 46, 47]. This observation could implicate either environmental or genetic factors although the fact that a similar latitude gradient is also evident for *MZ*-twin concordance rates [3] suggests that its basis is environmental. Regardless, however, this gradient is apparent in both *women* and *men* [46, 47]. Fourth, evidence of a prior *EBV* infection is found in essentially all MS patients compared to ~95% in controls [9, 48]. Indeed, if, in fact, a prior *EBV* infection is present in 100% of MS patients, then an *EBV* infection must be a necessary factor in the causal pathway leading to MS for all susceptible *women* and *men* [9]. Stated alternatively, in the context of the *Models* considered in this manuscript, an *EBV* infection must be a necessary factor for <u>*every*</u> set of "*sufficient*" exposures for <u>*every*</u> susceptible individual. Such a conclusion, by itself, strongly suggests that the pathogenic mechanisms are very similar among <u>*all*</u> susceptible individuals. It also suggests that the number of different sets of "*sufficient*" exposures, for each susceptible individual, is quite limited.

{*NB*: *This conclusion ignores those potential sets of environmental exposure (discussed in Section #1B), under which MS can only be provoked in some individuals by extremely unlikely circumstances (e.g., being inoculated with myelin basic protein together with complete Freund's adjuvant). Nevertheless, even including such possibilities will not affect our estimates for* (***c***) *and* (***d***) *because our estimates for these constants are derived from those failure probabilities,* (*Zw*), (*Zm*), *P*(*MS*) *and the (F:M) sex-ratio, which we actually (or potentially) observe–see* Methods *#4A; above.*}

Fifth, a vitamin D deficiency has been implicated as being an environmental factor in MS pathogenesis [9, 49–53] and this factor is related to MS in both *men* and *women* [49–53]. And lastly, smoking tobacco has been implicated as being environmental factor associated with MS pathogenesis [9, 54] and, again, this factor is associated with MS in both *women* and *men* [54].

Also, the genetic basis of MS seems to be very similar in both *women* and *men*. Thus, the strongest genetic associations with MS are for certain haplotypes within the *HLA*-region on the short arm of Chromosome 6 [3, 7, 55–61] and, in the predominantly Caucasian Wellcome Trust Case Control Consortium (*WTCCC*) dataset [60, 61], the most strongly MS-associated haplotypes in this region are similarly associated with MS in both *women* and *men* (Tables 3 & 4). Moreover, all but one the 233 genetic loci, which have been identified as being "MS-associated", are located on autosomal chromosomes and even the X-chromosome risk variant (identified by this study) was found to be present in both *men* and *women* with MS [60]–i.e., an individual's status at these different genetic loci is unlikely to differ systematically between genders (*see Section 6f in* S1 File). Also, studies of familial MS underscore the common genetic basis for MS in susceptible *women* and *men* [7, 62–64]. Thus, the risk of MS is increased for both twin and non-twin co-siblings (either *male* or *female*) of a proband with MS, regardless of the proband's gender [7, 62, 64]. Similarly, both *male* and *female* offspring of conjugal MS couples (i.e., where both parents have MS) have an increased risk of developing MS, which approaches that found for *MZ*-twins [62, 63]. Also, *male* and *female* half-siblings (i.e., who share one biological patent) are both at increased risk of MS, regardless of whether they share

**Table 3. MS associations for Class I and Class II *HLA*-haplotypes in *men* and *women***.**

| Haplotype | OR[†] (Women) | p | OR[†] (Men) | p |
|---|---|---|---|---|
| *DRB1*15:01~DQB1*06:02–1 copy* [‡] | 3.1 | < E-182 | 2.7 | < E-82 |
| *DRB1*15:01~DQB1*06:02–2 copies* [‡] | 6.5 | < E-110 | 6.2 | < E-70 |
| *DRB1*03:01~ DQB1*02:01–1 copy* [‡] | 1.1 | < 0.05 | 1.2 | < 0.01 |
| *DRB1*03:01~ DQB1*02:01–2 copies* [‡] | 2.9 | < E-21 | 2.2 | < E-7 |
| *A*02:01~C*05:01~B*44:02–1 copy* | 0.6 | < E-7 | 0.6 | < E-3 |
| *DRB1*15:01~DQB1*06:02–(AA)* [‡‡] | 1.8 | < E-3 | 3.1 | < E-3 |
| *DRB1*03:01~ DQB1*02:01–(AA)* [‡‡] | 1.5 | 0.005 | 1.4 | ns |
| *A*02:01~C*05:01~B*44:02–(AA)* [‡‡] | 0.5 | ns | 0.4 | ns |

\* Class I or Class II haplotypes from the WTCCC cohort [60, 61] within the human leukocyte antigen (*HLA*) region on the short arm of Chromosome 6. These haplotypes include either *HLA* Class I (*A, C & B*) or *HLA* Class II (*DRB1 & DQB1*) alleles. The WTCCC cohort consisted of 11, 376 MS patients from Europe and America. The large majority (71.8%) of this cohort of predominantly Caucasian MS patients were *women* [60, 61].

[†] Odds ratio (*OR*) for MS in *men* and *women* having either 1 or 2 copies of each haplotype except for the *A*02:01~C*05:01~B*44:02* haplotype, which had too few observations of the *2-copy* data. Each was compared to individuals having no copies of the other 2 *haplotypes* (95% *CI* range in parenthesis). The *p*-values are expressed in scientific notation as powers of 10 (E).

[‡] The difference in *OR* between possessing *1* or *2 copies* of the *DRB1*15:01~DQB1*06:02 01* haplotype was significant for both *women* ($p < 10^{-13}$) and *men* ($p < 10^{-10}$). Similarly, the difference in *OR* between possessing *1* or *2 copies* of the *DRB1*03:01~DQB1*02:01* haplotype was significant for both *women* ($p < 10^{-14}$) and *men* ($p < 0.001$).

[‡‡] The same Class I or Class II haplotypes in our African American (*AA*) population [65]. As in the WTCCC and in other Caucasian populations [3, 60, 61, 94], the large majority (79.6%) of this cohort of 1, 306 African American MS patients were *women* [65]. Because of the much smaller numbers of case and controls compared to the WTCCC, only single copies of these haplotypes are compared between *AA women* and *men*. Each of these haplotypes represent the Caucasian haplotypes admixed into the African genome [65].

**Table 4. MS associations for conserved extended *HLA*-haplotypes in *men* and *women*.**

| CEH Name* | CEHs** A~C~B~DRB1~DQB1~SNP | OR[†] (Women) | p | OR[†] (Men) | p |
|---|---|---|---|---|---|
| *c1* | *01:01~07:01~08:01~03:01~02:01~a6* | 3.5 | < E-7 | 1.8 | 0.02 |
| *c2* [‡] | *03:01~07:02~07:02~15:01~06:02~a1* | 2.7 | < E-78 | 2.5 | < E-36 |
| *c3* [‡] | *02:01~07:02~07:02~15:01~06:02~a1* | 1.9 | < E-17 | 1.9 | < E-9 |
| *c5* | *02:01~05:01~44:02~04:01~03:01~a3* | 0.5 | < E-9 | 0.5 | < E-6 |
| *c1 (AA)* [‡‡] | *01:01~07:01~08:01~03:01~02:01~a6* | 1.9 | 0.02 | 1.9 | ns |
| *c2 (AA)* [‡‡] | *03:01~07:02~07:02~15:01~06:02~a1* | 3.0 | 0.002 | 4.5 | 0.004 |

\* Arbitrary names for the four most common MS-associated conserved extended haplotypes (*CEHs*) in the WTCCC within the human leukocyte antigen (*HLA*) region on the short arm of Chromosome 6 [61].

\*\* These *CEHs* include both *HLA* Class I (*A, C & B*) and *HLA* Class II (*DRB1 & DQB1*) alleles in addition to the haplotypes of 11 single nucleotide polymorphisms (*SNPs*) spanning the *HLA* Class II chromosomal region [60].

[†] Odds ratio (*OR*) for MS in *men* and *women* having 2 copies of the (*c1*) CEH (only homozygotes were *MS*-associated) or any number of copies of the (*c2*), (*c3*), or (*c5*) *CEHs* (both homozygotes and heterozygotes were *MS*-associated) compared to having no copies of the other three *CEHs* (95% *CI* range in parenthesis). The *p*-values (if less than 0.001) are expressed in scientific notation as powers of 10 (E).

[‡] The difference in *OR* between the (*c2*) and (*c3*) *CEHs* was significant for both *women* (*p = 0.0001*) and *men* (*p = 0.03*).

[‡‡] The same *CEHs* in our African American (*AA*) population [65]. Because of the much smaller numbers of cases and controls (especially among men), only single copies of the *CEHs* (*c1*) ad (*c2*) are compared between the *AA women* and *men*. Each of these *CEHs* represent the Caucasian haplotypes admixed into the African genome [65].

the mother or father [62]. Collectively, these observations provide compelling evidence that a common genetic risk is affecting susceptible *men* and *women* alike. And finally, in our study of MS in African Americans [65], we found that when these risk-haplotypes (predominantly Caucasian in origin) were admixed with the African genome, they were associated with a risk of MS (in both *men* and *women*) similar to that found for these haplotypes in the predominantly Caucasian *WTCCC* population (Tables 3 & 4). Thus, even when these risk haplotypes are added to a different genetic background, the genetic basis MS is still quite similar for both *women* and *men* [65]. Each of these observations, is strongly supportive of the notion that susceptible *men* and *women* share a very similar, if not the same, genetic basis (whatever this is) and, therefore, that both can, potentially, be members of any "*i-type*" group (*see Section 6f, 6h in* S1 File).

Nevertheless, although it seems quite similar for both *women* and *men*, the genetics of MS is quite complex. First, the strongest MS-associated genetic trait is the *DRB1\*15:01~DQB1\*06:02* haplotype, located in the *Class II HLA* region on the short arm of chromosome 6 (*6p21*). For heterozygotes, this haplotype has an odds ratio (*OR*) of ($OR \approx 3$) and an *OR* of ($OR \approx 6$) for homozygotes [3, 7, 55–61]. The other genetic associations are quite weak with a median ($OR = 1.158$), and with an interquartile range of ($1.080 - 1.414$) [60]. Second, despite the *HLA-DRB1\*15:01~DQB1\*06:02* haplotype having the strongest MS-association of any, this haplotype, of all the haplotypes in this region, is (by far) the most highly "*selected*" among Caucasians, being carried by 24% of the Caucasian population [3, 7, 55–61]. Third, even considering just the strongest 103 associations among the 233 MS-associated loci, <u>everyone</u> among the 30, 248 individuals in the *WTCCC* population has a unique genotype [3, 60]. Moreover, only a small fraction of this population shares even four risk alleles with other individuals–almost all in different combinations from each other [3]. And, lastly, the fact that the *MZ*-twin concordance rates (from around the world), have always been reported to be less than 50% and have, generally, been reported to be less than ~30%, indicates that genetics plays only a minor role in determining who develops MS [3, 7, 55–62].

In addition, it is helpful to consider further our notion of exposure "*intensity*". For example, we consider (*see* Methods *#1A, #4F &* Fig 5*; see also Sections 6g, 6h & 8a, 8b in* S1 File) the possibility that each susceptible individual, potentially, might require a different family of "*sufficient*" sets of environmental exposure {$E_i$}; that each family, potentially, might have a different threshold difference ($\lambda_i$); that each family, potentially, might have a different proportionality constant ($R_i$); and that each family, potentially, might have many different "*sufficient*" sets within it (*see* Methods *#2A–B; see also Section 6h in* S1 File). Moreover, to explain ($\lambda > 0$), we concluded that this exposure, at least for low "*intensities*" measured on the (*a*) scale, needs to be more "*intense*" in <u>every</u> susceptible woman than the *minimum* exposure necessary considering <u>all</u> susceptible *men*. Considering each of these circumstances together, it seems rather surprising, if this marked variability described above truly existed, that this could possibly lead to a circumstance in which <u>all</u> susceptible *women* required a more "*intense*" exposure compared to <u>some</u> susceptible *men* (e.g., Fig 5*; see also Sections 6g, 6h; & 8a, 8b in* S1 File). Alternatively, if everyone required the same (or a very similar) set of environmental factors or events for the exposure to be "*sufficient*", it might be easier to rationalize any differences (between "*i-type*" groups) in the "*intensity*" of their required exposures (*see Section 6g, 6h in* S1 File). In addition, this might also make it easier to rationalize the fact that those environmental factors, which have been consistently identified as MS-associated, have been linked to MS, generally, but not to any subgroup [9, 41, 42, 44–54].

As noted earlier (*see Equations 4 & 7;* Methods *#4A & 4F*), by assuming that: ($c = d \leq 1$), we are also assuming that the difference in disease expression between *men* and *women* is due entirely to a difference in the likelihood of their experiencing a "*sufficient*" environmental

exposure. Therefore, because *currently* ($Zw_2 > Zm_2$) and because the (*F:M*) *sex ratio* is increasing–*see* Methods *#2C* –those conditions in which ($c = d \leq 1$) would necessarily lead to the conclusion that, *currently*, susceptible *women* are more likely to experience a "*sufficient*" environment compared to susceptible *men* despite the fact that the probabilities of exposure to each family of environmental events, $P(\{E_i\} | E_T)$ and $P(\{E_{iw}\} | E_T)$, are fixed constants during any ($E_T$).

Moreover, there are also several additional lines of evidence, which, taken together, also suggest that the circumstance of ($c = d = 1$) is unlikely. First, on theoretical grounds, any circumstance, in which ($\lambda \leq 0$), are also those, in which the condition of ($c = d \leq 1$) is not possible (*see* Methods *#4D & 4E; see also Section 6c in* S1 File). Second, considering only those conditions, in which ($\lambda > 0$)–*see* Methods *#4F; see also Section 6c in* S1 File–there are only two possibilities:

1) ($R \leq 1$); in which case: $\forall(\lambda)$: ($c < d \leq 1$)

and: 2) ($R > 1$); in which case: ($\lambda > 0$)

As discussed earlier (*Methods #4F; above*), the possibility that: ($\lambda > 0$) & ($R > 1$) creates a paradox because these two conditions indicate that, at low "*intensity*" exposures–i.e., {$H(a) \leq \lambda$}–susceptible *men* are much more "*responsive*" compared to susceptible *women* (i.e., $R \approx 0$) and, yet, at higher "*intensity*" exposures–i.e., {$H(a) > \lambda$}–somehow, susceptible *women* become more "*responsive*" compared to susceptible *men* (i.e., $R > 1$)–*see Sections 6g, 6h & 8a, 8b in* S1 File. To rationalize this paradox, we introduced the notion of a "critical exposure *intensity*" (or threshold) level of exposure necessary for disease to occur in each susceptible individual (*see* Fig 5; *see also S1–S3 Figs & Sections 6g & 8a; in* S1 File). However, even if this notion of an "*intensity*" threshold is appropriate, to accommodate the condition of: ($c = d = 1$), requires extreme conditions, which don't match well with the response curves presented in Fig 3 (e.g., *S2, S3 Figs in* S1 File). By contrast, in all circumstances, those conditions, for which: ($c < d$) are much simpler, don't require extreme conditions, fit with any value of ($R$), and match much better with the response curves depicted in Fig 4 (e.g., *S1–S3 Figs in* S1 File).

Third, as discussed earlier (*see* Methods *#4F*), the response curves required for conditions where: ($c = d = 1$) & ($R > 1$) have very steep ascending portions {generally due to large values of ($R$), very small values of ($\lambda$), or both} and, thus, present only a narrow window of opportunity to explain the Canadian data [6] regarding the changes in the (*F:M*) *sex ratio* and its magnitude over time (*see* Methods *#4F; see also* Fig 3). Also, for these response curves, following this narrow window, and contrary to the evidence [6], the (*F:M*) *sex ratio* decreases with increasing exposure (Fig 3). By contrast, the Canadian data suggests that there has been a gradual and sustained increase in the (*F:M*) *sex ratio* over the past several decades [6]. Moreover, if the notion of a "critical exposure *intensity*" is correct, the switch in the *F:M sex ratio* from *predominantly male* to *predominantly female* generally occurs too late in the response curves to match well with the Fig 3 requirements (*see S1–S3 Figs, Section 8a, 8b in* S1 File).

Fourth, as noted in *Methods #1D*, there seems to be little impact of the ($E_{sib}$) environment on the development of MS. However, when ($c = d$) the only explanation for ($R > 1$) is a disproportionate likelihood of exposure to "*sufficient*" environments experienced by *women* (*see above*). Thus, proband siblings and their non-twin co-siblings (both *men* and *women*), despite sharing common genes and a common childhood environment, still depend upon (and differ in) only their ($E_{pop}$) exposures to develop their MS.

Finally, and most importantly, for each of the known (or suspected) environmental factors related to MS pathogenesis, there is no evidence to suggest that *women* are disproportionately experiencing them compared to *men*. Thus, the *month-of-birth* effect is equally evident for

*men* and *women* [41, 42, 44, 45]; the latitude gradient is the same for both genders [9, 46, 47]; the impact of the ($E_{twn}$) environment is of the same magnitude for *men* and *women* (*Section 1d in* S1 File*)*; By young adulthood (i.e., 20–25 years), the likelihood of an *EBV* infection (a factor, almost certainly, in the causal chain leading to MS), is about equal (~95%) for both genders. Nevertheless, infection likely occurs earlier among *women* [9, 66, 67] although infectious mononucleosis (at least illness requiring hospitalization) seems to be more common among *men* [67–70]; vitamin D levels are the same in both genders [49–53]; and, in fact, smoking tobacco is more common among *men* [9, 54]. Taken together, these epidemiological observations suggest both that susceptible *women* and *men* require the same environmental events to cause their MS, and that, *currently*, they are each experiencing these events in an approximately equivalent manner. Therefore, these observations suggest that:

$$P(E \mid G, F, E_T)_2 \approx P(E \mid G, M, E_T)_2 \approx P(E \mid G, E_T)_2 \qquad (8a)$$

In this context, the possibility that ($c = d \leq 1$), ($\lambda > 0$) & ($R > 1$)–which are depicted in Fig 3–seems remote, especially given the facts that the relevant exposures are *population-wide* and that the difference between ($Zw$) and ($Zm$) can only be explained by a disproportionate exposure to "*sufficient*" environments by susceptible *women* (*see* Eq 4)–a circumstance for which there is decidedly no evidence (*see above*).

Moreover, because, the population experiences the same level of exposure ($u = a$) during any ($E_T$), therefore, from *Equations 4 & 7*, if this approximate equivalence is correct, then this indicates that any observed disparity between ($Zw_2$) and ($Zm_2$) must be due a disparity between ($c$) and ($d$), in which case, both: ($c < d$) and: ($R \approx 1$). Such a configuration easily explains an increasing ($F:M$) *sex ratio* and its magnitude throughout most (or all) of the response curves (*see* Fig 4A–4D; *see also S1 Fig and Section 8a, 8b in* S1 File), it accounts for a time in MS history where the disease may have been more prevalent in *men* [40]–e.g., Fig 4C & 4D–and, even though susceptible *men* and *women* have the same *population-wide* exposure, $\{P(\{E_i\} \mid E_T)\}$, it does not present us with the paradox that both: 1) susceptible *women* have an increased exposure compared to *men* when the exposure "*intensity*" ($a$) is high; and 2) only susceptible *men* are exposed when the exposure "*intensity*" ($a$) is low (*see above*).

Nevertheless, any condition, for which ($c < d$), does require that some susceptible *men* will never develop MS, even when the correct genetic background occurs together with an environmental exposure "*sufficient*" to cause MS in those individuals. Indeed, if, as suggested: ($R = 1$), then, from *Section 6c in* S1 File, it is necessarily the case that: ($c < d$) and indeed, both in theory and in practice (*see Results*), our findings indicate that, in this circumstance, such *men* (i.e., who never develop MS) comprise 21–99% of the susceptible *male* subset ($M, G$). Naturally, in this circumstance, it seems likely that the proportion of *women* who ultimately develop MS, given the same conditions, will also be less than unity (e.g., Fig 4B & 4D). However, because, for the purposes of our analysis, we needed to assume that: ($d = 1$), this possibility cannot be addressed using the Canadian data.

Some of the individuals who don't develop "*clinical*" MS despite having an environmental exposure "*sufficient*" to cause MS, no doubt, will have subclinical disease. Indeed, as suggested by several autopsy studies, the prevalence of "asymptomatic" MS in the population ($Z$) may be as high as ~0.1% [71–74]. Moreover, such a figure is generally supported by several magnetic resonance imaging (*MRI*) studies of asymptomatic individuals [75, 76]. Nevertheless, although these considerations suggest that some proportion of MS can be asymptomatic, this fact seems unlikely to account for any difference either ($c$) from ($d$), or of ($c$) from the expected 100% occurrence of MS in *men* who are both genetically susceptible and, in addition, experience an environment "*sufficient*" to cause MS given their specific genotype. Thus, if asymptomatic

disease did account for (*c*) being less than (*d*), then *men* should account for a disproportionately large percentage of these asymptomatic individuals. However, this is decidedly not the case. Rather, *men* account for only 16% of the asymptomatic individuals detected by *MRI* [75, 76]–a percentage well below their *current* proportion of symptomatic cases [3]. Consequently, if (*c* < *d*), as the Canadian data [6] seems to indicate, then chance must play a role in disease pathogenesis.

Alternatively, however, perhaps our definition of exposure "*intensity*" does not account properly for certain other potential aspects of exposure "*intensity*", which might play an important role in disease pathogenesis. As a concrete example of this notion, suppose that one (or more) of the "*sufficient*" sets of exposures for the $i^{th}$ susceptible individual includes both a deficiency of vitamin D and a prior *EBV* infection [3, 9], each occurring during or after some critical age of the person's life (not necessarily the same age). Furthermore, suppose that, with all other necessary factors being equal in the $i^{th}$ susceptible individual, a mild vitamin D deficiency for a short period during the critical time, together with an asymptomatic *EBV* infection at age 10, causes MS to develop 10% of the time, whereas a more prolonged, and more marked, vitamin D deficiency during the critical period, together with a symptomatic *EBV* infection (mononucleosis) at age 15, causes MS to develop 75% of the time. Notably, each of these posited conditions is "*sufficient*", by itself, to cause MS; the only difference is in the likelihood of this outcome, given the different levels (i.e., "*intensity*") of exposure.

Although this notion of "*intensity*" differs from our previous definition and can't be easily quantified, presumably, there will be a positive correlation between an increasing "*intensity*" of this exposure (whatever this means operationally) and an increasing risk of MS for each susceptible individual. Moreover, each susceptible individual must reach a *maximum* likelihood of developing MS as the "*intensity*" of their exposure increases. This maximum may be at 100% or it may be at something less than this but, whatever it is, there must be a maximum for each person. In addition, unlike our previous definition of $P\{E_i\}$, where only one "*sufficient*" set of exposures was necessary, here, an individual for whom two or more of their "*sufficient*" sets of exposure occur, may experience a greater "*intensity*" of exposure than if only one set occurs. Nevertheless, none of these circumstances alters the fact that each susceptible person will still have their "*maximum*" likelihood of developing MS under optimal environmental conditions. We can then define the "*intensity*" of exposure–$P(E \mid G, E_T)$–as the average (or expected) "*intensity*" of exposure (however this is measured) experienced by members of the (*G*) subset, given the environmental conditions of the time ($E_T$). When no "*sufficient*" exposure occurs for any member of (*G*): $P(E \mid G, E_T) = 0$. When the "*intensity*" has increased to the point where every member of (*G*) has reached their maximum likelihood of developing MS then: $P(E \mid G, E_T) = 1$. And, again, we can define (*u*), as the odds of a susceptible person experiencing a "*sufficient*" exposure:

$$u = P(E \mid G, E_T)/[1 - P(E \mid G, E_T)].$$

Although, clearly, this conceptualization of exposure *intensity* is different (and perhaps more realistic) than the "*sufficient*" exposures considered earlier, two of its features are particularly noteworthy. First, randomness is integral to this notion of exposure "*intensity*". Thus, disease expression at low "*intensity*" exposures, by definition, incorporates an element of chance because the likelihood of developing MS under these conditions must be less than the *maximum*, for at least some susceptible individuals. If not, then this "*intensity*" of exposure would have no impact on anyone, and this *Model* becomes equivalent to (i.e., reverts to) the "*sufficient*" exposures *Model* considered earlier. Second, despite exposure being measured differently, and despite the hazard functions likely being different, all the equations and

transformations presented in *Methods #4A–F* (*above*) as well as the calculated response curves are unchanged by measuring exposure as *"intensity"* in this manner rather than as *"sufficiency"*. Indeed, this conclusion applies to any measure of exposure, which incorporates the notion of *"sufficient"* sets of environmental exposure defined earlier (*see* Methods *#1B*).

Thus, by any measure, the Canadian data [6] seem to indicate that there is a "truly" random factor (i.e., an element of chance) in MS pathogenesis, at least for *men*, which determines, in part, who gets the disease and who does not. Such a conclusion might be viewed as surprising because, in the universe envisioned by many physicists, events are (or seem to be) deterministic [77, 78]. For example, imagine a rock thrown at a window. If the rock has a mass, a velocity, and an angle of impact sufficient to break the window, given the physical state of the window at the moment of impact, then we expect the window to break 100% of the time. If the window only breaks some of the time, likely, we would conclude that we hadn't adequately specified the sufficient (i.e., initial) conditions. If the *population-based* observations in over 29,000 Canadian MS patients are to be believed, however, this is not so for the development of MS. Even for an individual with a susceptible genotype and an environmental experience "*sufficient*" to cause disease given their specific genotype, they still may or may not develop the illness. This result cannot be ascribed to contributions from other, unidentified, environmental factors because each set of environmental circumstances considered here is defined to be "*sufficient*", by itself, to cause MS in that specific susceptible individual. If other environmental conditions were needed to cause MS reliably in that individual, these conditions would already be necessary components of these "*sufficient*" environments (*see* Methods *#1B*). Even altering the definition of exposure to include the importance of different meanings of exposure "*intensity*" doesn't alter this conclusion. Certainly, the invocation of a "truly" random processes in disease pathogenesis requires replication, both in MS and in other disease states, before being accepted as fact. Nevertheless, if replicated, such a result would imply that there is a fundamental randomness to the behavior of some complex physical systems (e.g., organisms).

Notably, other authors, have also invoked random mechanisms as being involved in MS disease pathogenesis [79–84]. One group has used different methods of advanced computer network modeling, both to reproduce the known dynamics of the MS disease process and to reproduce the biological diversity that exists within actual patient populations [81–83]. These models, which are extended to predict the effects of given treatments [81–83], incorporate randomness into the complex interactions of immune system cells including those of B-cells, T-regulatory cells, T-helper cells, cytotoxic T-cells, and natural killer cells, together with their response to, or productions of, certain immune-related cytokines (e.g., IFN$\gamma$, IL-2, IL-10 and IL-17). The use of such stochastic modeling seems quite promising both in characterizing the known dynamics of MS and in predicting the response of MS patients to different therapies using simulated patient populations [81–83].

Another group has developed a model that combines deterministic factors together with a so-called "stochastic forcing" factor, by which these authors mean by an external stimulus that contributes to MS disease expression (i.e., relapses and remissions). The behavior of this factor is taken to be intrinsically random and can only be characterized probabilistically [79, 80, 84]. This random event is envisioned to exert its effect through a non-linear contribution to "gene expression noise" (i.e., the random variability in gene expression) possibly modulated by the so-called "transient transcriptome", which includes several very short-lived RNA species (e.g., enhancer, short intergenic non-coding, and antisense) that are known to impact gene expression [79, 84]. Thus, these authors hypothesize that this "stochastic noise in gene expression, through its pervasive effects on virtually all biological processes, may be the factor that amplifies and reshapes the deterministic effects of genetic and environmental risk factors" and,

indeed, some of the solutions to their model predict well the apparently random pattern of relapse and remission observed in MS [79].

Nevertheless, for each of these cases, as well as for other such modeling approaches [82], randomness is incorporated (*a priori*) into the model to make the model more representative of the "actual" disease process and, thereby, make the predicted responses to therapy more accurate. Unfortunately, however, the fact that including randomness improves the performance of these models does not serve as a *test* of whether "true" randomness ever occurs. For example, the outcome of a coin-flip or measuring the kinetic energy of a gas particle in a gas at thermodynamic equilibrium may be most accurately "modeled" by treating the coin-flip outcome, or the kinetic energy of a particle, as a random variable taken from sample spaces with defined probability distributions. Nevertheless, question remains as to whether these probability distributions represent a *complete* description of these processes, or whether these distributions are merely a convenience for us to compensate for a deficiency in our detailed knowledge about the underlying conditions (e.g., in the case of a coin flip performed with respect to the inertial frame of the Earth: the initial orientation of the coin; the direction, location, and magnitude of the forces exerted on the coin at the time of the flip; and the forces acting on the coin as it travels through the air and ultimately hits the ground). Indeed, the issue of whether such processes (which we model as random) represent "truly" random events, has been debated ever since the notion of determinism was first introduced by the French polymath Laplace in the early 17th century [78, 85, 86]. For example, in 1908, the mathematician and physicist, Henri Poincaré, argued that: "every phenomenon, however trifling it be, has a cause, and a mind infinitely powerful and infinitely well-informed concerning the laws of nature could have foreseen it from the beginning of the ages. If a being with such a mind existed, we could play no game of chance with him; we should always lose." [77]. A similar deterministic viewpoint is still current among many authorities today [78, 85, 86].

It is, therefore, of note that some contemporary authorities have argued from fundamental physical principles that "true" randomness (i.e., thermodynamic equilibrium or maximum entropy) was a primordial property of our universe in the earliest tiny fraction of a second of the big bang and that this inherent randomness is reflected by a *currently* observable randomness for both microscopic (i.e., quantum uncertainty) and macroscopic descriptions of the universe [85]. By contrast, the deterministic hypothesis envisions that earliest state of the universe was one of minimum entropy and asserts that, when we perceive certain macroscopic events as being due to chance, this perception is illusory and merely a reflection of our ignorance regarding the relevant initial conditions [77, 78, 85]. This is certainly the viewpoint expressed by Poincaré in 1908 (quoted *above*), and even with the subsequent development of quantum theory and an understanding of quantum uncertainty, many contemporary authorities still subscribe to a substantially similar view [78, 85, 86]. For example, one contemporary author has expressed this deterministic worldview succinctly by noting that, while "the quantum equations lay out many possible futures, . . . they deterministically chisel the likelihood of each in mathematical stone" [79]. By contrast, the contemporary physicist and mathematician Stephen Hawking, while agreeing that the wave equations of quantum physics are deterministic and that the entropy of the early universe was minimal, still argues, at a theoretical level, that the existence of black hole emissions implies that "the loss of particles and information down black holes [means] that the particles that [come] out [are] random. One [can] calculate probabilities, but one [can] not make any definite predictions. Thus, the future of the universe is not completely determined by the laws of science" [86]. Other authorities disagree that the existence of black hole emissions have any such implications [78]. Obviously, the question of which, if either, of these alternative views of the universe represents reality has far-reaching implications [77, 78, 85, 86].

Perhaps the best contemporary evidence for macroscopic randomness, cited by proponents of the non-deterministic worldview, is the case of biological evolution by means of natural selection [86]. Thus, natural selection is envisioned to be a non-sentient process, which depends upon the occurrence of apparently random events and, using these events, permits living species to respond both continuously and adaptively to the varying environmental conditions of different times, or different places, or both. Moreover, the direction, in which any new species evolves, is seemingly not predictable but, rather, depends upon the nature of the specific random events, which take place.

Placed into a broader context, this biological evolution, which has been so clearly documented on Earth, is probably best viewed as a part of (or as a continuation of) the process of chemical evolution–a process that began only a few minutes after the onset of the big bang, and at a time when the universe was composed of ~75% hydrogen, ~25% helium, a few of their isotopes, and a small admixture of lithium [78, 87, 88]. The chemistry of this early universe was extremely rudimentary. Helium (*He*) is the lightest of the nobel gases and reacts with almost nothing. Hydrogen (*H*) and lithium (*Li*) combine to form only a few simple chemical compounds such as lithium hydride (*LiH*) and molecular hydrogen ($H_2$). A more complex chemistry (and, in particular, the chemistry necessary both to create and sustain life and to permit biological evolution) only evolved later with synthesis of the heavier atomic elements–a synthesis that, following these first few minutes of the big bang, only occurred with the collapse and/or explosion of massive stars at the end of their life cycle [78, 87, 88]. This synthesis, and the subsequent build-up of heavier elements in the universe, was gradual and took time.

Also, this process of chemical evolution continues to this day, not only with the ongoing synthesis of heavier elements inside contemporary stars and the interactions of these elements with each other throughout the universe, but also with the synthesis of a multitude of novel chemical compounds, created by living organisms. Moreover, each step of this evolutionary sequence seems to require the occurrence of random events–i.e., which nuclei happen to collide, whether they fuse, whether (and when) they decay, where and when stars form, which stars become a supernova, where and when these supernovas occur, which life-forms evolve and under what circumstances, with what chemistries, in what places, and with what evolution over time, etc.

In this broader context, then, it is extremely hard to imagine that processes such as biological evolution or the function of the immune system are pre-determined outcomes and yet, for the macroscopic processes that produce them, to be so exquisitely adaptive to contemporary external events and, also, to be dependent upon apparently random occurrences. However, notwithstanding any deficiency we might have with our imagination, it is extremely difficult to *prove* that any macroscopic process (including this evolutionary sequence or the function of the immune system) is "truly" random.

Nevertheless, despite this difficulty, the hypothesis of determinism is quite fragile in the sense that, if the "true" randomness of even one macroscopic process or event could be established, the hypothesis of determinism would be undermined. However, to do this requires an experiment (i.e., a *test*) for which the outcome predicted by determinism differs from that predicted by non-determinism. Perhaps surprisingly, the epidemiological data collected about MS in Canada (or similar data that might be collected about other disease states and other populations in the future) presents us with the opportunity to apply just such *test*. Thus, the deterministic hypothesis *requires* the condition that: ($c = d = 1$) although the observation of this condition, by itself, could not establish determinism as true. By contrast, the observation that either: ($c < d = 1$) or: ($c \leq d < 1$) would indicate that "true" randomness is an integral part of the process of MS disease development and, thus, would undermine the notion that our universe is deterministic. Consequently, if replicated (in MS or in other disease states), the

Canadian data on MS [4–8], which strongly suggests that ($c < d$), provides empiric evidence in support of the non-deterministic worldview.

There are two features of the response curves in *men* and *women* that merit further comment. First, the plateaus for these curves (if, in fact, $c < d \leq 1$)–e.g., Figs 1D, 2D and 4A–4D–reflect this inherent randomness in the process of disease development. Indeed, in this circumstance, it would be this randomness, rather than the genetic and environmental determinants, which lies at the heart of the difference in disease expression between *men* and *women*. Thus, genetically susceptible *women*, who experience an environment "*sufficient*" to cause MS given their genotype, are more likely to develop disease compared to susceptible *men* in similar circumstances. Consequently, if ($c < d$), there must be something about "*female-ness*" that favors disease development in *women* over *men* although, whatever this is, it is not part of any causal chain of events leading to disease (i.e., in the sense that, if a truly random coin-flip determines, in part, an outcome, then this random event is not part of any _causal_ chain). As noted above, if either ($c < d = 1$) or: ($c \leq d < 1$), then disease development in the setting of a susceptible individual experiencing a "*sufficient* exposure must include a truly random event (at least for *men*). Moreover, if: ($c < d$), the fact that this random process favors disease development in *women* does not make it any less random. For example, the flip of a biased coin is no less random than the flip of a fair coin. The only difference is that, in the former circumstance, the two possible outcomes are not equally likely. In the context of MS, "*female-ness*" would then be envisioned to bias the coin differently than does "*male-ness*" (whatever these terms mean).

In this regard, a recent study of the "transcriptomic profile" of MS patients (in either relapse or remission) and of controls, found 174 genes whose transcription products were altered in both remission and relapse–a high proportion of which displayed a so-called "mirror pattern" such that they were upregulated in remission and downregulated in relapse or *vice versa* [89]. Moreover, using a co-expression analysis of these genes, these authors were able to demonstrate that these transcriptomes seemed to be organized into four modules–three *female*-specific and one *male*-specific [89]. With the caveat that this report concerns relapses and remissions (and not causation), these results suggest that, despite *men* and *women* sharing the same 174 genes, the physiology of their transcription differs between genders. Such physiological differences between *women* and *men*, potentially, might contribute to creating a bias such as that posited *above* for "*female-ness*".

Second, the thresholds reflect the minimum exposure at which disease expression begins and the response curves, with increasing exposure, that follow this onset [3], need to account for the changes in MS epidemiology that have been observed over the last several decades [6, 22–30, 40]. If the hazards in *men* and *women* are not proportional, as discussed *above*, little accounting is necessary. By contrast, if the hazards are proportional and if both: ($c < d \leq 1$) & ($\lambda > 0$), then these circumstances could account for all of the epidemiological observations–i.e., the increasing prevalence of MS [22–30], the continuously increasing proportion of *women* among MS patients [6, 22–30, 40], the magnitudes of the observed (*F:M*) *sex ratios* [3, 6], and a 1922 study [3, 40], in which MS prevalence in both the United States and Europe was reported to be substantially higher in *men* than in *women* (e.g., Figs 1D and 4A–4D *& S1–S3 Figs in* S1 File).

During the development of our *Longitudinal Model*, we observed that when the prevailing environmental conditions of a time ($E_T$) were such that: {$P(E \mid E_T) = 0$}, no member of ($G$) could develop MS. Previously, we considered the possibility that such an environment might not be possible to achieve because some susceptible individuals might be able to develop MS under _any_ environmental conditions–i.e., if these cases were "*purely genetic*" [3]. Upon further reflection, however, such a possibility seems remote. Most importantly, if an *EBV* infection is, in fact, a necessary factor for MS pathogenesis in _every_ susceptible individual who *currently* develops disease [9, 48], then this observation, alone, excludes the possibility of "*purely genetic*"

MS. Also, MS seems to be a disease of relatively recent onset. For example, MS (or any similar disease) does not seem to occur spontaneously in any other mammalian species (regardless of how closely they are related to us) and, *currently*, its occurrence is very infrequent among indigenous Africans (the continent where humans originally evolved). Each of these observations support the notion that disease MS was far less frequent in antiquity than it is today. Moreover, the first clinical description of MS was published in 1868 by Charcot, although earlier pathological descriptions predated this clinical description by ~30 years [89, 90]. Perhaps, the earliest described cases of MS were either that of a women named Halldora from Iceland (c.1193) or that of Saint Lidwina of Schiedam (c. 1396), although each of these case descriptions, especially that of Halldora, seem unconvincing [89–92]. The argument that Augustus d'Este (c. 1822) suffered from MS is more compelling [89, 90]. And even though many human afflictions were initially described during the advent of modern medicine in the 19th century, MS is a rather distinctive disorder, and it seems likely that, if MS existed, case descriptions (familiar to us) would have appeared in earlier eras. Moreover, with the onset of the industrial revolution in the late 18th or early 19th century, the environmental conditions began to change substantially (especially for humans). Therefore, both MS as a disease and permissive environmental conditions seem likely to be of relatively recent onset. More importantly, ever since its original description, MS seems to be changing in character–a fact that underscores the critical importance of environmental factors in MS pathogenesis. For example, although considered uncommon initially, ever since Charcot's initial characterization, MS has become increasingly recognized as a common neurological condition [90–94]. Also, in the 19th century Charcot's triad of limb ataxia, nystagmus (internuclear ophthalmoplegia), and scanning (cerebellar) speech was considered typical whereas, today, while this triad still occurs, such a syndrome is unusual [89–94]. Moreover, in the late 19th and early 20th centuries, the disease was thought to be more (or equally) prevalent in *men* compared to *women* [40, 81, 82], whereas, today, *women* account for 66–76% of the cases [3, 94]. Also, in many parts of the world, MS is increasing in frequency, particularly among *women* [6, 22–30]. Indeed, in Canada, *P*(*MS*) has increased by an estimated minimum of 32% over a span of 35–40 years (*see Section 7a in* S1 File)–a circumstance which has led to a 10% increase in the proportion of *women* among MS patients ($p < 10^{-6}$) over the same time-interval [6].

By contrast, those genetic markers, which are associated with MS, seem to have been present for far greater periods of time. For example, the best established (and strongest) genetic associations with MS are for certain haplotypes within the *HLA* region on the short arm of chromosome 6 (e.g., Table 3), including haplotypes such as *DRB1*15:01~DQB1*06:02; DRB1*03:01~ DQB1*02:01*; and *A*02:01~C*05:01~B*44:02* [55–61]. Each of these haplotypes, as well as each of the conserved extended haplotypes (*CEHs*) in the *HLA* region–*see* Table 4 –is well represented in diverse human populations around the globe [65, 94, 95] and, thus, both these haplotypes and these *CEHs* must be of ancient origin. Presumably, therefore, the absence of MS prior to the late 12th or 14th (and possibly the early 19th) century, together with the markedly changing nature of MS over the past 200 years, points to a change in environmental conditions as the basis for the recent occurrence of MS as a clinical entity and for the changes in MS epidemiology, which have taken place over the past two centuries. Consequently, it seems that $\{P(E \mid G, E_T) = 0\}$ is possible under those environmental conditions that existed prior to the late 12th or 14th century and, thus, that "*purely genetic*" MS does not exist.

## Conclusion

Our results, together with the implications that our different *Models* have for the nature of MS susceptibility, lead to important conclusions regarding the underlying mechanisms of disease

pathogenesis. Thus, the two principal findings of our study are that: $P(G) \leq 0.52)$} and: ($c < d \leq 1$). As a consequence of these conclusions, there must be three essential components to disease pathogenisis. First, for the development of MS to take place, this requires the individual has an appropriate (i.e., a susceptible) genotype. If an individual lacks this susceptible genotype, MS cannot develop. Moreover, much of the population (and most *women*) lack this essential component of MS pathogenesis. Second, for MS to develop in a susceptible individual, they must experience an environmental exposure "*sufficient*" to cause MS given their specific genotype. If a susceptible individual doesn't experience such an exposure, again, MS cannot develop. And third, even when the necessary genetic and environmental factors, required for MS pathogenesis, co-occur for an individual, this still seems to be insufficient for that person (at least for susceptible *men*) to develop MS. Thus, even in this circumstance, disease pathogenesis seems not to be deterministic but, rather, seems to involve an important element of chance (i.e., disease development is, in part, "truly" random). Finally, the conclusion that the macroscopic process of disease development includes this truly random element, if replicated (either in MS or in other complex diseases), provides empiric evidence in support for the notion that our universe is not deterministic.

## Supporting information

**S1 File.**
(PDF)

## Acknowledgments

We are especially indebted to John Petkau, PhD, Professor Emeritus, Department of Statistics, University of British Columbia, Canada. Dr. Petkau helped immeasurably with this project; devoting numerous hours of his time to critically reviewing early drafts of this manuscript and providing an invaluable contribution both to the clarity and to the logical development of the mathematical and statistical arguments presented herein.

## Author Contributions

**Conceptualization:** Douglas S. Goodin.

**Formal analysis:** Douglas S. Goodin.

**Methodology:** Douglas S. Goodin.

**Software:** Douglas S. Goodin.

**Writing – original draft:** Douglas S. Goodin.

**Writing – review & editing:** Douglas S. Goodin, Pouya Khankhanian, Pierre-Antoine Gourraud, Nicolas Vince.

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
