## [Decision Letter · Decision Letter 0]

2 Jan 2023

PONE-D-22-31636Multiple Sclerosis: Exploring the Limits of Genetic and Environmental SusceptibilityPLOS ONE

Dear Dr. Goodin,

Thank you for submitting your manuscript to PLOS ONE. Your manuscript is reviewed by one expert in the field. I have real difficulty to locate reviewers for your ms. I also read your manuscript quickly. We all agree that this is an interesting and potentially important work.  The Reviewer has some major concerns. The Reviewer’s comments are attached.  Please address the concerns in your revised version. After careful consideration, I feel that your manuscript has merit and will be reconsidered for publication in PLoS One after major revision. 

We look forward to receiving your revised manuscript.

Kind regards,

Luwen Zhang

Academic Editor

PLOS ONE

Journal Requirements:

Reviewers' comments:

Reviewer's Responses to Questions

**Comments to the Author**

1. Is the manuscript technically sound, and do the data support the conclusions?

Reviewer #1: Yes

2. Has the statistical analysis been performed appropriately and rigorously? 

Reviewer #1: Yes

3. Have the authors made all data underlying the findings in their manuscript fully available?

Reviewer #1: Yes

4. Is the manuscript presented in an intelligible fashion and written in standard English?

Reviewer #1: Yes

5. Review Comments to the Author

Reviewer #1: The authors present a very detailed and passionate statistical analysis of multiple sclerosis genetics and non-genetic risk factors and conditions for probable onset. The manuscript is well written, but I have two main concerns at this point:

1. The format is not attuned to the PlosOne format and readers. About 90% of the sections “Methods” and “Longitudinal model” should be moved to supplementary material. The Discussion section should be split: the math-heavy part should go into supplementary, and the more clinical part should be retained in the main text. The Conclusion section is missing – even if it is a half-page, I’d ask authors to please add it.

2. While the statistical part is extremely well detailed, other prior work on stochasticity in MS that came up in the past 10 years, have not been cited. At minimum, authors should incorporate these 8 references from this query

https://scholar.google.com/scholar?q=%22stochasticity%22+intitle%3A%22multiple+sclerosis%22&hl=en&as_sdt=0%2C22&as_ylo=2012&as_yhi=

i. Umeton, Renato, et al. "Multiple sclerosis genetic and non-genetic factors interact through the transient transcriptome." Scientific reports 12.1 (2022): 1-13.

ii. Bordi, Isabella, et al. "A mechanistic, stochastic model helps understand multiple sclerosis course and pathogenesis." International Journal of Genomics 2013 (2013).

iii. Mentis, A‐FA, et al. "Viruses and endogenous retroviruses in multiple sclerosis: From correlation to causation." Acta Neurologica Scandinavica 136.6 (2017): 606-616.

iv. Pernice, Simone, et al. "Computational modeling of the immune response in multiple sclerosis using epimod framework." BMC bioinformatics 21.17 (2020): 1-20.

v. Pernice, Simone, et al. "Exploiting Stochastic Petri Net formalism to capture the Relapsing Remitting Multiple Sclerosis variability under Daclizumab administration." 2019 IEEE International Conference on Bioinformatics and Biomedicine (BIBM). IEEE, 2019.

vi. Bordi, Isabella, et al. "Noise in multiple sclerosis: unwanted and necessary." Annals of Clinical and Translational Neurology 1.7 (2014): 502-511.

vii. Irizar, Haritz, et al. "Transcriptomic profile reveals gender-specific molecular mechanisms driving multiple sclerosis progression." PLoS One 9.2 (2014): e90482.

viii. Sips, Fianne LP, et al. "In silico clinical trials for relapsing-remitting multiple sclerosis with MS TreatSim." BMC Medical Informatics and Decision Making 22.6 (2022): 1-10.

I will be happy to evaluate the manuscript once these comments have been addressed: I believe there is indeed very insightful research in this manuscript.

6. PLOS authors have the option to publish the peer review history of their article (what does this mean?). If published, this will include your full peer review and any attached files.

Reviewer #1: No

---

## [Author Response · Author response to Decision Letter 0]

8 Feb 2023

We detail (below) the modifications that we have made. 

Reviewer #1

Comments

1. The format is not attuned to the PlosOne format and readers. About 90% of the sections “Methods” and “Longitudinal model” should be moved to supplementary material. The Discussion section should be split: the math-heavy part should go into supplementary, and the more clinical part should be retained in the main text. The Conclusion section is missing – even if it is a half-page, I’d ask authors to please add it.

Response:

 As requested, we have removed a large portion of the mathematical development from the Methods and Discussion sections of the Main Text and placed this material into a new Supplementary Material. We have also added a Conclusion section, as requested.

 2. While the statistical part is extremely well detailed, other prior work on stochasticity in MS that came up in the past 10 years, have not been cited. At minimum, authors should incorporate these 8 references from this query.

Response:

 As requested, we have incorporated a discussion of these additional papers into the Discussion section of the Main Text.

I hope with these modifications, you will now find the manuscript suitable for publication in PLoS One. Thank you very much for your help and your consideration of this matter. I look forward to hearing from you in due course.

---

## [Decision Letter · Decision Letter 1]

3 Apr 2023

PONE-D-22-31636R1Multiple Sclerosis: Exploring the Limits and Implications of Genetic and Environmental SusceptibilityPLOS ONE

Dear Dr. Goodin,

Thank you for submitting your manuscript to PLOS ONE. Your article is somewhat strange to many reviewers.    The previous reviewer refused to review the revision and I had a hard time to locate a second reviewer.  After careful consideration, we feel that your manuscript will likely be suitable for publication if it is revised to address the points raised by the second Reviewer. Therefore, my decision is "Minor Revision."

We look forward to receiving your revised manuscript.

Kind regards,

Luwen Zhang

Academic Editor

PLOS ONE

Journal Requirements:

Reviewers' comments:

Reviewer's Responses to Questions

**Comments to the Author**

1. If the authors have adequately addressed your comments raised in a previous round of review and you feel that this manuscript is now acceptable for publication, you may indicate that here to bypass the “Comments to the Author” section, enter your conflict of interest statement in the “Confidential to Editor” section, and submit your "Accept" recommendation.

Reviewer #2: (No Response)

2. Is the manuscript technically sound, and do the data support the conclusions?

Reviewer #2: Yes

3. Has the statistical analysis been performed appropriately and rigorously? 

Reviewer #2: Yes

4. Have the authors made all data underlying the findings in their manuscript fully available?

Reviewer #2: Yes

5. Is the manuscript presented in an intelligible fashion and written in standard English?

Reviewer #2: Yes

6. Review Comments to the Author

Reviewer #2: Report on Multiple Sclerosis: Exploring the Limits of Genetic and Environmental Susceptibility

The manuscript presents models to define the relative contribution of genetic and environmental determinants to the onset of multiple sclerosis (MS), addressing the comments of a previous reviewer, with whom we agree. In itself, the approach is interesting and potentially delivers valuable information the relative contribution of heritable and non heritable factors and chance in causing MS. However, several issues still limit the usefulness of the work.

The proposed model explores the interplay between genetic and environmental factors predisposing to MS. The model is based on the assumption that genetic predisposition and exposure to environmental factors are both required to develop MS. It has long been acknowledged that both heritable and non-heritable factors contribute to MS onset, and several alleles that increase susceptibility to MS have been identified. However, given the weak contribution of each genetic factor, the possibility that people lacking genetic predisposition develop MS has never been dismissed. Furthermore, protective factors and lifestyles are not taken into account explicitly. So, although the proposed model reasonably describes a large proportion of MS epidemiology, its inherent limits should be discussed.

The Discussion should be rewritten altogether. In its present version it is over 12 pages long and, being extremely verbose, it conveys the impression that authors want to cover every aspect of human knowledge. For instance, 4 pages are devoted to a discussion of determinism in science. This is fully irrelevant to the focus of the paper, even in light of the proposed casual occurrence of MS in the male population: "chance" might simply represent some factor which has eluded definition thus far (adoption of protective lifestyles?). An element of randomness in MS is also supported by other studies – quoted by the authors– and the deterministic nature of our universe has been questioned since the introduction of quantum physics, over 100 years ago. This paper is not an adequate forum for further discussion of the topic and statements on it must be removed from Abstract and Conclusions as well.

Several paragraphs summarise the description of MS in history. Again, this paper is not devoted to tracing the history of MS, or of medicine in general. Moreover, the lack of a reliable description of MS cases before early 19th century is no indication of the absence of cases: considering that symptoms spontaneously remit over long time periods and that they are not specific, the disease possibly was confused with others or overlooked altogether, in particular when the duration of human life was much shorter than today. By the way, a putative case possibly occurred in Iceland in the 12th century (Eur Neurol. 2006;55(1):57-8). However, the topic is only suitable for works devoted to the history of medicine.

On the other hand, the limits of the results are at best poorly discussed. For instance, in the discussion, Authors state: "Nevertheless, the notion that MS in men and MS in women are fundamentally different diseases, involving distinct environmental events, seems implausible", a perfectly acceptable statement. In the next 4 pages, lot of (somewhat unnecessary) data are brought around to support the statement, but not a single line is devoted to evaluate the implications of this implausible conclusion on the validity of the model.

In the same line, different scenarios about the hazard functions for MS in men and women are presented (non-proportional, proportional and intermediate). Apparently, the intermediate is the only one that predicts a F:M ratio compatible with reported data. However, I have been unable to find a clear commitment of the authors toward this scenario.

Overall, the paper would gain readability if re-written with a sharper focus on its main topic, a model explaining the observed epidemiological data on MS, eliminating all digressions about non pertinent issues, in particular in the Discussion.

Minor point: In Figures 1-4 the scale for the blue curve is not indicated.

7. PLOS authors have the option to publish the peer review history of their article (what does this mean?). If published, this will include your full peer review and any attached files.

Reviewer #2: No

---

## [Author Response · Author response to Decision Letter 1]

19 Apr 2023

Reviewer #1

Comments

1. The model is based on the assumption that genetic predisposition and exposure to environmental factors are both required to develop MS. It has long been acknowledged that both heritable and non-heritable factors contribute to MS onset, and several alleles that increase susceptibility to MS have been identified, given the weak contribution of each genetic factor, the possibility that people lacking genetic predisposition develop MS has never been dismissed.

Response:

 We agree with the reviewer that all the genetic factors so far identified (or likely to be identified in the future) are only weakly associated with disease. However, I’m not sure what this implies. The genetics of MS are complex. In fact, considering the genes that have already been associated with MS, everyone with MS has a unique combination [3] and the most strongly MS-associated haplotype in Caucasians (HLA-DRB1*15:01~DQB1*06:02) is also the most selected haplotype in the Caucasian population, being carried by 25% of individuals.

Nevertheless, for the purposes of our analysis, we define, very broadly, the set of “susceptible” genotypes (G) to include every individual (genotype) in the population who has any non-zero chance of developing MS under some environmental conditions. Thus, if: “people lacking genetic predisposition” (i.e., anyone) can develop MS, then, it must be the case that: {P(G)=1}. Notably, therefore, any conclusion that: {P(G)<1}, excludes the possibility that MS can ever occur in “people lacking genetic predisposition” for the disease. 

Therefore, our consistent finding, in all Models, that: {P(G)≤0.52}, excludes the possibility raised by the reviewer. We have now clarified this point in the Discussion.

 2. Furthermore, protective factors and lifestyles are not taken into account explicitly. So, although the proposed model reasonably describes a large proportion of MS epidemiology, its inherent limits should be discussed.

Response:

We define a “sufficient” environment as any set of environmental conditions that are “sufficient” to cause MS in some member of the (G) subset. Notably this definition includes every environmental condition or experience (known, suspected, or unknown), which is required (i.e., necessary) for such “sufficiency”, including any protective factors. Thus, although we don’t know what factors are involved or how they act, a “sufficient” set of exposure takes into account all factors, including protective factors and lifestyles. 

The only requirement for a set of environmental conditions to be considered a “sufficient” environment, is for that set to be “sufficient”, by itself, to cause MS in one or more members of (G). We have now clarified this both in the Methods and the Discussion.

3. The Discussion should be rewritten altogether. In its present version it is over 12 pages long and, being extremely verbose, it conveys the impression that authors want to cover every aspect of human knowledge. For instance, 4 pages are devoted to a discussion of determinism in science. This is fully irrelevant to the focus of the paper, even in light of the proposed casual occurrence of MS in the male population: "chance" might simply represent some factor which has eluded definition thus far (adoption of protective lifestyles?). An element of randomness in MS is also supported by other studies – quoted by the authors– and the deterministic nature of our universe has been questioned since the introduction of quantum physics, over 100 years ago. This paper is not an adequate forum for further discussion of the topic and statements on it must be removed from Abstract and Conclusions as well.

Response:

Regarding protective life styles, see our response to Point 2 (above)

There are two principal conclusions, which we draw from our analysis and these are stated succinctly as:

 1. P(G)<1

 and: 2. c< d ≤1

The first of these is fairly straight-forward and this conclusion seems inescapable, based both on the data from Canada [6,7] and on the data about MZ-twin concordance rates, reported from other locations around the world [3]. Simply put, only a fraction of the population (Z) has any chance of getting MS. Thus, “genetic susceptibility” is a pre-condition for the development of MS. See our response to Point 1 (above).

The second conclusion, however, is more complicated. First, there are potential scenarios (conditions) that can be envisioned (within the context of our Model), under which: (c=d=1) might be possible. Principal among these is the possibility that the hazard functions for developing MS in the two genders are not proportional. The Reviewer feels that our description of this scenario as implausible is a “perfectly acceptable statement”. Obviously, we feel the same way, as do many other authorities. Nevertheless, because this scenario is the most obvious and most compelling counterargument to our conclusion that: (c<d≤1), it is essential that we cite and evaluate the evidence upon which these “feelings” are based. Thus, we strongly disagree with the Reviewer that this discussion is “somewhat unnecessary”.

Furthermore, our discussion of randomness vs. determinism is not “fully irrelevant” to the focus of the paper. Rather it is central. Indeed, the 1st Reviewer of our paper asked us to expand substantially our discussion of “randomness” and to include a discussion of 8 additional papers on this topic. The current Reviewer wants it cut. Nevertheless, the very nature of our second principal conclusion (above) requires that the issue of “randomness” be addressed. Thus, if: (c<d≤1), then “randomness” must play a role in disease pathogenesis. Obviously, if we are going to conclude that this is the case, it is essential that we consider alternative possibilities in our Discussion.

This Reviewer seems to have mixed feelings on the topic. On the one hand, the Reviewer states that “the deterministic nature of our universe has been questioned since the introduction of quantum physics, over 100 years ago”. And on the other hand, the Reviewer states that “…even in light of the proposed casual occurrence of MS in the male population: "chance" might simply represent some factor which has eluded definition thus far (adoption of protective lifestyles?)”. 

Does the reviewer believe that the issue of “true” randomness in nature has been settled and that our conclusion that disease development involves “chance” is nothing new? (first statement). Or does the Reviewer believe that what we perceive as “chance” is really a reflection of our ignorance? (second statement). Many (most) modern physicists would agree with the second statement, whereas many modern biologists would agree with the first (including those who wrote the 8 papers the 1st Reviewer asked us to include).

However, as we point out in our Discussion, in many cases, when “randomness” is invoked, this is done for the purpose of making whatever Model is being considered more realistic and its predictions more accurate. By contrast, in our case, “randomness” enters the Model spontaneously as a finding or conclusion from our analysis of the Canadian MS data (i.e., c<d≤1). It is not introduced a priori [e.g., 79-84]. Moreover, in our case, there is a clear “deterministic” alternative to our second conclusion, which is that (c=d=1). As the Reviewer points out, and as we also note in our Discussion, the Canadian data cannot settle this issue and, clearly, the Canadian data requires replication (either in MS or in another disease state) before being accepted as fact. Nevertheless, if: (c<d≤1), disease development cannot be deterministic.

The 1st Reviewer thought that our discussion of “randomness” needed to be expanded. The 2nd Reviewer thinks it should be cut. We feel that, regardless of whether or not our second principal conclusion (above) is accepted, a discussion of “randomness” cannot be avoided. However, we are happy to modify this section further as you see fit. We leave this to your editorial discretion.

4. Several paragraphs summarise the description of MS in history. Again, this paper is not devoted to tracing the history of MS, or of medicine in general. Moreover, the lack of a reliable description of MS cases before early 19th century is no indication of the absence of cases: considering that symptoms spontaneously remit over long time periods and that they are not specific, the disease possibly was confused with others or overlooked altogether, in particular when the duration of human life was much shorter than today. By the way, a putative case possibly occurred in Iceland in the 12th century (Eur Neurol. 2006;55(1):57-8). However, the topic is only suitable for works devoted to the history of medicine.

Response:

As noted in our response to Point 1 (above), our analysis demonstrates that a genetic predisposition is a necessary pre-condition for MS to develop and that only a fraction of the population has such a predisposition. However, this circumstance leaves open the possibility that, in some cases, a genetic predisposition may be all that matters (i.e., in these individuals any environment is “sufficient”). Nevertheless, our Model generates response curves, which can be extrapolated back to: {P(MS│ E_T)=0}. The question is whether or not these environmental conditions can actually exist. Our Model cannot address this question directly. Nevertheless, both the apparent infrequency of this condition in antiquity, and the fact that MS has never been identified in other mammalian species, suggests that this condition can, in fact, exist. If this is correct, then MS can never be “purely genetic”. Thus, the purpose of our brief discussion about the history of MS is to make the case that a “sufficient” environment (but not any environment) is also a necessary precondition for MS to develop. However, we are happy to modify this section further as you see fit. Therefore, again, we leave this to your editorial discretion. 

5. Apparently, the intermediate is the only one that predicts a F:M ratio compatible with reported data. However, I have been unable to find a clear commitment of the authors toward this scenario.

Response:

 As shown in Figure 1D, the “strictly” proportional Model with (R<1) & (c<d≤1) also predicts an F:M ratio compatible with reported data. We have now clarified this point.

6. Minor point: In Figures 1-4 the scale for the blue curve is not indicated.

Response:

The scale of the blue curve is indicated in each Figure. We have now clarified this in the legends.

I hope with these modifications, you will now find the manuscript suitable for publication in PLoS One. Thank you very much for your help and your consideration of this matter. I look forward to hearing from you in due course.

---

## [Editor Report · Decision Letter 2]

27 Apr 2023

Multiple Sclerosis: Exploring the Limits and Implications of Genetic and Environmental Susceptibility

PONE-D-22-31636R2

Dear Dr. Goodin,

We’re pleased to inform you that your manuscript has been judged scientifically suitable for publication and will be formally accepted for publication once it meets all outstanding technical requirements.

Kind regards,

Luwen Zhang

Academic Editor

PLOS ONE
---

## [Editor Report · Acceptance letter]

2 May 2023

PONE-D-22-31636R2 

Multiple Sclerosis: Exploring the Limits and Implications of Genetic and Environmental Susceptibility 

Dear Dr. Goodin:

I'm pleased to inform you that your manuscript has been deemed suitable for publication in PLOS ONE. Congratulations! Your manuscript is now with our production department. 

Kind regards, 

on behalf of

Dr Luwen Zhang 

Academic Editor

PLOS ONE